# CLIMB: Data Foundations for Large Scale Multimodal Clinical Foundation Models

**Wei Dai**[1]  **Peilin Chen**[1]  **Malinda Lu**[1]  **Daniel Li**[1]  **Haowen Wei**[2 3]  **Hejie Cui**[4]  **Paul Pu Liang**[1]

## Abstract

Recent advances in clinical AI have enabled remarkable progress across many clinical domains. However, existing benchmarks and models are primarily limited to a small set of modalities and tasks, which hinders the development of large-scale multimodal methods that can make holistic assessments of patient health and well-being. To bridge this gap, we introduce Clinical Large-scale Integrative Multimodal Benchmark (CLIMB), a comprehensive clinical benchmark unifying diverse clinical data across imaging, language, temporal, and graph modalities. CLIMB comprises 4.51 million patient samples totaling 19.01 terabytes distributed across 2D imaging, 3D video, time series, graphs, and multimodal data. Through extensive empirical evaluation, we demonstrate that multitask pretraining significantly improves performance on understudied domains, achieving up to 29% improvement in ultrasound and 23% in ECG analysis over single-task learning. Pretraining on CLIMB also effectively improves models' generalization capability to new tasks, and strong unimodal encoder performance translates well to multimodal performance when paired with task-appropriate fusion strategies. Our findings provide a foundation for new architecture designs and pretraining strategies to advance clinical AI research. Code is released at this link.

## 1. Introduction

Advances in AI for clinical data have significantly helped doctors process and analyze complex clinical information for diagnosis, treatment planning, and decision support (Esteva et al., 2019; Liu et al., 2025; Rajpurkar et al., 2022).

These domains include analyzing clinical images (Johnson et al., 2019; Irvin et al., 2019a), processing clinical notes (Liu et al., 2025; Johnson et al., 2016), and predicting patient outcomes (Yan et al., 2024). Despite these achievements, most current approaches remain limited to a few modalities primarily in the image and text domain (Thakoor et al., 2019; Jing et al., 2023b; Sharma et al., 2022), failing to capture the interactions between many medical indicators that clinicians routinely combine to make holistic assessments on patient health and well-being (Liang et al., 2024b; Rajendran et al., 2023; Shaik et al., 2024).

To develop the next generation of holistic multimodal clinical foundation models, we introduce Clinical Large-scale Integrative Multi-modal Benchmark (CLIMB), a comprehensive multimodal clinical benchmark that unifies data across imaging, language, temporal, and genomic modalities. Our dataset comprises 4.51 million patient samples, totaling 19.01 terabytes of data, with a diverse spectrum of modalities: 707K 2D imaging data (including X-rays, dermoscopy images, fundus images, and pathology slides), 1.83M 3D or video samples (ultrasounds, CT scans, endoscopic images and MRI images), 871K 1D data (electronic health records, EEG, ECG, gait and genomic data), 69.3K graph data (brain networks, molecules) and 1.03M multimodal data combining multiple of the above modalities. We accomplish this through a novel data collection and preprocessing pipeline that standardizes diverse data formats from 33 different medical institutions while preserving the natural patterns of missing data. The dataset encompasses 96 different clinical conditions across 13 clinical domains, making it one of the largest and most diverse public clinical benchmarks to date.

Through extensive empirical evaluation on CLIMB, we establish comprehensive benchmarks and best practices for clinical multimodal learning. As illustrated in Fig. 3, our analysis yields three key insights:

1. **Multitask pretraining:** We evaluate single encoders trained jointly on all tasks within each modality in CLIMB. Our experiments show that multitask pretraining significantly improves performance across clinical tasks, achieving up to 32.54% AUC improvement in COVID ultrasound and other understudied areas. Fur-

[1]Massachusetts Institute of Technology [2]Athinoula A. Martinos Center for Biomedical Imaging [3]Harvard Medical School [4]Stanford University. Correspondence to: Wei Dai <dvdai@mit.edu>.

*Proceedings of the 42nd International Conference on Machine Learning*, Vancouver, Canada. PMLR 267, 2025. Copyright 2025 by the author(s).

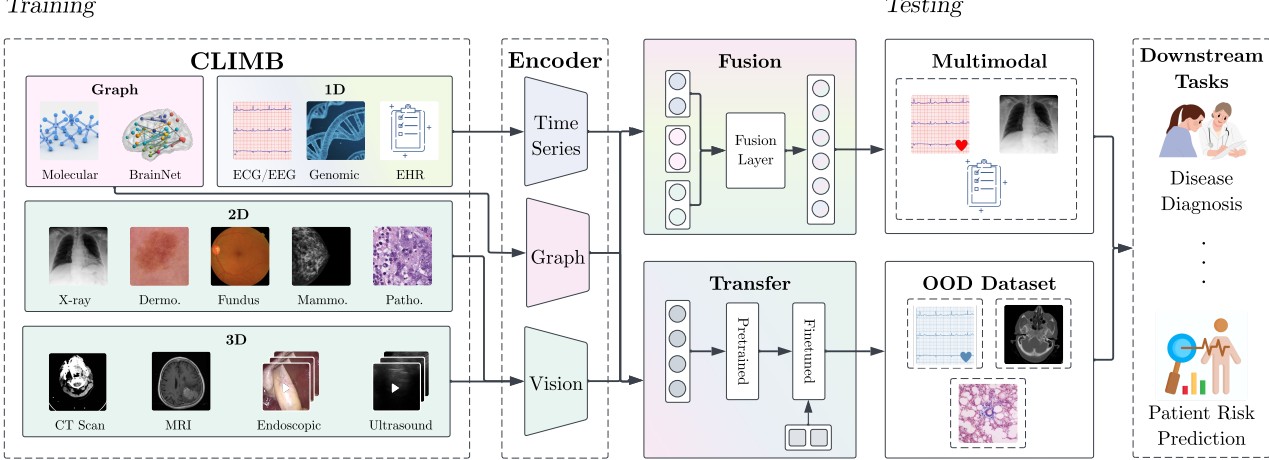

*Figure 1.* **Overview of the CLIMB framework for training and testing multimodal datasets.** The training phase incorporates diverse data modalities: graphs (molecular, BrainNet), 1D signals (ECG/EEG, genomics, EHR), 2D images (X-rays, dermoscopy, fundus, mammograms, pathology), and 3D scans (CT, MRI, endoscopy, ultrasound). Through multitask training on heterogeneous clinical data, our framework enhances model performance across individual tasks, particularly for understudied modalities defined in Fig. 2(b). This approach improves both generalization to novel tasks and multimodal understanding when combined with appropriate fusion strategies, ultimately advancing performance on critical clinical applications like disease diagnosis and patient risk prediction.

thermore, pretraining on CLIMB drastically improves model performance in novel and understudied tasks for general-domain encoders, specialized clinical encoders, and clinical large vision language models (LVLMs).

2. **Few-shot transfer:** We test how models pretrained on CLIMB generalize to new clinical tasks with limited labeled data. Models pretrained on CLIMB demonstrate significant improvements in few-shot learning scenarios, achieving up to 29% improvement in ultrasound and 23% in ECG tasks under few-shot settings compared to pretraining on existing datasets.

3. **Multimodal fusion:** Finally, we investigate different strategies for combining multimodal clinical data, including imaging, text, and time series on MIMIC-IV, a multimodal clinical benchmark. Our results show that single-modality pretraining on CLIMB enhances multimodal learning performance, leading to successful transfer to MIMIC-IV, and that complex fusion strategies perform better on challenging tasks.

In light of these findings, we release our vision, EEG, and ECG unimodal and multimodal models trained on CLIMB, which achieve state-of-the-art performance on multiple clinical tasks. We also provide detailed recommendations for model architecture selection and pretraining strategies across clinical modalities, establishing a practical framework for future clinical AI development. All code for data collection, training, evaluation, and pretrained weights is available at this link.

## 2. Related Work

We cover related work in unimodal and multimodal clinical benchmarks and models.

### 2.1. Unified Multimodal Clinical Benchmarks

The convergence of computational advances and large clinical datasets (Johnson et al., 2016; 2019; Irvin et al., 2019a) has enabled AI systems to match human performance across various medical tasks, from retinopathy detection to drug discovery (Tsiknakis et al., 2021; Sone et al., 2021; Rajkomar et al., 2019). While large-scale multimodal foundation models have shown promise in learning unified clinical representations (Liang et al., 2024a; Yang et al., 2024; Philiastides et al., 2021), current benchmarks typically focus on limited modalities like X-rays, pathology, or their combinations (Moses, 2021; Schneider et al., 2022; Nasir et al., 2023). Large benchmarks include BenchMD (Wantlin et al., 2023) and CARES (Xia et al., 2024), covering 7 clinical modalities (1D, 2D, and 3D) and 16 different 2D and 3D image modalities, respectively. As shown in Table 1, however, our dataset uniquely incorporates time series and graph data alongside traditional clinical imaging while maintaining the widest coverage across each of the modalities.

### 2.2. Multimodal Clinical Models

Recent clinical AI models broadly fall into two categories: LLM-based multimodal systems and specialized vision encoders. LLM-based approaches like Llava-Med (Li et al., 2023) and Med-Flamingo (Moor et al., 2023) combine frozen vision encoders with language models for clinical

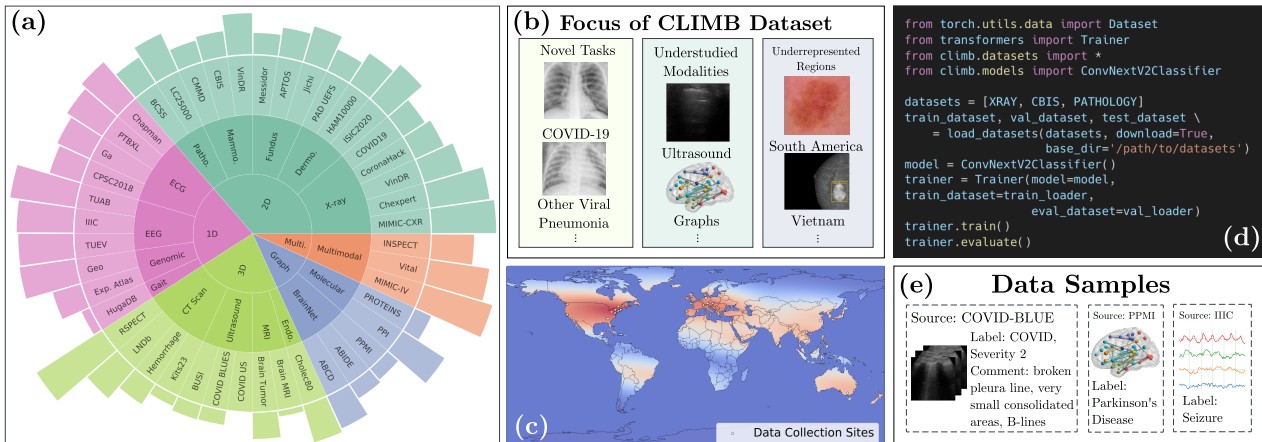

*Figure 2.* **Overview of CLIMB benchmark and code. (a) Visualization of CLIMB dataset composition.** The inner ring displays the primary data modalities (2D, 1D, Graph, Multimodal). The middle ring represents major clinical modalities within each modality. The outer ring shows the names of specific datasets within each category, with the outer bar plot representing the number of samples in that dataset. A detailed description of each modality and datasets are included in App. A. **(b) Focus of dataset collection.** During the collection of CLIMB, we aim to collect a diverse range of datasets, with a special focus on novel tasks, datasets from understudied modalities, and datasets from underrepresented regions. **(c) Distribution of data collection sites in CLIMB.** Red regions indicate areas where clinical datasets are commonly collected, whereas blue regions indicate places where clinical dataset collections are rare. **(d) Example code usage on CLIMB framework.** This code example loads a custom mixed subset of CLIMB spanning across three modalities, then trains a ConvNextv2 classifier on the dataset mixture with unified vocabulary. **(e) Sample data from CLIMB.** CLIMB preserves detailed labels, metadata and comments explaining the diagnosis.

tasks, but notably do not optimize the visual components. In contrast, vision-focused works like Swin-Unet (Cao et al., 2022) develop specialized encoders for clinical imaging, though typically for single modalities. M4oE (Jiang & Shen, 2024) represents a rare exception, using Swin Transformers (Liu et al., 2021) to create a mixture of experts model across both CT and MRI modalities, but they failed to expand it further into more modalities. While general vision architectures like ConvNeXt and EVA-2 (Liu et al., 2022; Fang et al., 2024) have demonstrated strong performance on natural images, efforts to develop clinical-specific encoders have largely focused on adapting older architectures, as seen in CLIP-based PMC-CLIP (Lin et al., 2023) and Vision Transformer-based MedViT (Manzari et al., 2023), leaving the potential of modern architectures for clinical tasks largely unexplored.

## 3. Dataset

In this section, we provide an overview of the CLIMB dataset, sourced from 44 public datasets spanning 15 modalities in 13 clinical domains. A detailed description of each modality and dataset is included in App. A, and a visual overview of the dataset is provided in Fig. 1. We first introduce the selection criteria of the CLIMB dataset, followed by the dataset information and statistics. In the end, we provide a simple code snippet demonstrating the use of CLIMB for model training and inference.

### 3.1. Dataset Selection Criteria

CLIMB unifies diverse public clinical datasets into a unified benchmark designed specifically for developing and evaluating multimodal medical AI systems. To maximize the diversity of the data, we established three key criteria to guide our dataset selection process. As illustrated in Figure 2(b), we prioritize datasets that address one or more of the following objectives:

1. **Novel tasks:** Recent emerging clinical challenges, such as COVID-19 diagnosis from chest imaging.
2. **Understudied modalities:** Data types traditionally underrepresented in clinical AI, including electroencephalograms (EEG), endoscopic videos, and graphs.
3. **Underrepresented regions:** Clinical data from geographic areas with limited representation in existing benchmarks, particularly South America and developing regions of South Asia.

A detailed description of our selection methodology and inclusion criteria is provided in App. A.2.

### 3.2. Dataset Construction

Based on the selection criteria above, CLIMB was eventually sourced from 44 public datasets. Fig. 2(e) shows three examples in CLIMB, with labels COVID-19, Parkinson's disease and Seizure, from ultrasound, brain network, and EEG, respectively (Wie, 2021; Cui et al., 2022; Jing et al., 2023a). Notably, additional metadata and explanation labels for the datasets are also preserved, as shown in the

Table 1. **Comparison of clinical benchmarks.** Abbreviations: BN = Brain Networks, Mol = Molecules, ECG = Electrocardiogram, EEG = Electroencephalogram, Genom = Genomics, Mammo = Mammography, Derm = Dermoscopy, Fund = Fundus, Path = Pathology, CT = Computed Tomography, MRI = Magnetic Resonance Imaging, US = Ultrasound, Endo = Endoscopy. CLIMB (ours) is the the most diverse, comprehensive, largest clinical public multimodal dataset up to date, which enables holistic studies on multiple modalities and provides data foundation for large-scale clinical pretraining across vision, language, time series and graphs. * For BenchMD, 5.1M out of 5.2M samples are EEGs, with only 0.1M samples from other modalities.

| Dataset | #Samples | Graph | | 1D | | | | | 2D | | | | | 3D | | | |
|---|---|---|---|---|---|---|---|---|---|---|---|---|---|---|---|---|---|
| | | BN | Mol | ECG | EEG | Genom | Gait | Text | X-ray | Mammo | Derm | Fund | Path | CT | MRI | US | Endo |
| BenchMD (Wantlin et al., 2023) | 5.2M* | ✗ | ✗ | ✓ | ✓ | ✗ | ✗ | ✗ | ✓ | ✓ | ✓ | ✓ | ✗ | ✓ | ✗ | ✗ | ✗ |
| PMC-VQA (Zhang et al., 2023) | 149K | ✗ | ✗ | ✗ | ✗ | ✗ | ✗ | ✓ | ✓ | ✗ | ✗ | ✗ | ✗ | ✓ | ✓ | ✓ | ✗ |
| GMAI-MMBench (Chen et al., 2024a) | 26K | ✗ | ✗ | ✗ | ✗ | ✗ | ✗ | ✓ | ✓ | ✗ | ✓ | ✓ | ✗ | ✓ | ✓ | ✓ | ✓ |
| CARES (Xia et al., 2024) | 18K | ✗ | ✗ | ✗ | ✗ | ✗ | ✗ | ✓ | ✓ | ✗ | ✓ | ✓ | ✓ | ✓ | ✓ | ✓ | ✓ |
| **CLIMB (ours)** | 4.51M | ✓ | ✓ | ✓ | ✓ | ✓ | ✓ | ✓ | ✓ | ✓ | ✓ | ✓ | ✓ | ✓ | ✓ | ✓ | ✓ |

Table 2. **Comparison of dataset statistics.** CLIMB-QA (ours) demonstrates the largest scale in terms of number of words, QA pairs, and overall dataset size.

| Dataset | #Words | #QA Pairs | Size of Dataset |
|---|---|---|---|
| PMC-VQA | 10.2M | 227K | - |
| GMAI-MMBench | 980K | 26K | 49 GB |
| CARES | 1.74M | 41K | 21.61 GB |
| **CLIMB-QA (ours)** | 129.1M | 4.51M | 19.01 TB |

COVID-19 ultrasound example.

To enable holistic training and benchmarking, we need to unify the input data loading and prediction tasks. To unify the label space, we consider two options. The first is to pose all tasks as multi-label classification given clinical data samples from different modalities. We combine the vocabularies in each dataset while merging semantically equivalent labels. Specifically, given a heterogeneous dataset collection $\mathcal{D} = \{D_1, ..., D_K\}$ with mixed annotation types (multi-label/multi-class), we define a unified label vocabulary $\mathcal{V} = \bigcup_{k=1}^{K} V_k$ where $V_k$ represents the label set of dataset $D_k$. To ensure consistency, we standardize terminology variations and combine similar concepts such as *Lung Opacity* from Chexpert and *Infiltration* from Vindr CXR to maintain a clean, unified vocabulary while preserving the original clinical meaning. A detailed description of how we standardize the taxonomy is included in App. C. The list of classes are included in App. Table 9.

The second option is to pose everything as QA. We also built a closed choice question-answering version of the dataset, namely CLIMB-QA, for comparable LVLM evaluation. In CLIMB-QA, each dataset is preprocessed with QA pairs containing close-ended multiple-choice questions. A detailed description of CLIMB-QA along with examples are included in App. B.

In addition, we preserve the metadata, demographic information, segmentation masks, and associated clinical reports from the original dataset and link them to every sample where applicable. To ensure comparability across model architectures, this information is not exposed to the model in the experiments, although we hope future works could utilize it to develop more robust and fair methods.

### 3.3. Dataset Statistics
CLIMB contains 4.51 million samples totaling 19.01 terabytes, with the following composition: 871K (19.31%) 1D time series and text data (including electronic health records, EEG, ECG, gait and genomic data), 707K (15.68%) 2D images (including X-rays, dermoscopy images, fundus images, and pathology slides), 1.83M (40.56%) 3D or video data (including ultrasounds, CT scans, endoscopic images and MRI images), 69.3K (1.54%) graph data (including brain networks, molecules), and 1.03M (22.90%) multi-modality data combining multiple of the above modalities.

As shown in Table 1, CLIMB has the widest range of modalities while incorporating time series and graph data, which distinguishes it from existing multimodal benchmarks in the field that typically only include images and text. Table 2 provides a quantitative comparison with other clinical datasets, demonstrating that CLIMB significantly exceeds existing benchmarks in scale, containing 129.1M words and 4.51M QA pairs across 19.01 TB of data, substantially larger than other multimodal clinical QA datasets.

Figure 2(a) shows the distribution across primary modalities and the size of individual datasets. We carefully balance the dataset such that each modality contains 3-5 datasets, providing multiple data sources per modality while maintaining diversity within each category. The geographic distribution of data sources is shown in Figure 2(c). The dataset includes data from 37 medical institutions across 18 countries, including contributions from Vietnam, Iraq, India, and Brazil, expanding the representation beyond traditionally well-represented regions.

### 3.4. Dataset Interface and Code Usage
We present a unified interface to download and process our dataset into a unified format for mixed large-scale pretraining, given that the user has provided agreement consents

on individual dataset websites. Figure 2(d) shows the code for a standard workflow of loading multiple medical imaging datasets and training a classification model with our CLIMB framework. The entire training and evaluation script can be completed in under ten lines of code while maintaining the flexibility for any custom models or training loops. We also provide a standardized training pipeline that is easily reproducible and parallelizable across multiple machines and instances.

# 4. Experiments

We run extensive experiments to investigate the core technical challenges for developing clinical foundation models with CLIMB. Specifically, we ask the following questions:

1. **RQ1:** Can multitask pretrained clinical models work across multiple tasks consistently, especially for understudied tasks?
2. **RQ2:** How well do multitask pretrained clinical models transfer to new tasks within the same clinical modality, especially tasks with limited data?
3. **RQ3:** Can multitask pretrained unimodal models be fused effectively to tackle multimodal clinical tasks?

## 4.1. Experimental setup

To answer the above research questions, we design our experiments as follows:

**RQ1: Multitask pretraining.** We investigate whether multitask learning can enable robust universal encoders for clinical tasks. For each input modality (vision 2D/3D, graph, EEG, ECG), we train a single encoder jointly on all related tasks in CLIMB. Each encoder is combined with a classification head that predicts task-specific labels from an aggregated vocabulary $\mathcal{V}$, which encompasses diagnostic terms across all tasks within that modality. For each modality, we evaluate both specialized medical models and general-domain architectures. In vision, we compare medical-specific encoders (MedViT (Manzari et al., 2023), PMC-CLIP (Lin et al., 2023), RAD-DINO (Pérez-García et al., 2025)) against general vision models (ConvNeXTv2 (Liu et al., 2022), SBB2 (Radford et al., 2021), Swin Transformer (Liu et al., 2021), EVA-2 (Fang et al., 2024), InternViT (Chen et al., 2024b)). For time-series data, we evaluate ECG-specific model, ECG JEPA (Kim, 2024), against general time-series architectures, UniTS (Gao et al., 2024). In the EEG domain, we test specialized architectures including SPARCNet (Jing et al., 2023b), CNNTransformer (Peh et al., 2022), FFCL (Li et al., 2022), ContraWR (Yang et al., 2023), STTransformer (Song et al., 2021), and BIOT (Yang et al., 2024). We also evaluate SoTA clinical VLM, LLaVa-Med (Li et al., 2023), on CLIMB-QA, a question-answering version of CLIMB designed for comparability across large VLMs and traditional encoders. Details on construction are included in App. B.

**RQ2: Few-shot transfer.** We investigate how well models pretrained on CLIMB can generalize to novel clinical tasks with limited labeled data. To evaluate few-shot generalization, we test on out-of-distribution (OOD) datasets $\mathcal{D}_{ood} \not\subset \mathcal{D}_{train}$. The OOD datasets are selected to reflect either a new task or a different data source within the same modality, simulating real-world scenarios where models must adapt to novel diagnostic tasks with limited labeled examples (1, 8, and 32 samples). We curated a diverse set of 10 datasets spanning 9 modalities, as detailed in App. D.7.1. A quantitative analysis of the OOD datasets is included in App. F. For each modality, we use the best-performing model from our RQ1 experiments: ConvNextv2 for vision tasks, ECG JEPA for ECG analysis, and BIOT for EEG processing. We compare two scenarios: (1) models initialized with publicly released pretrained weights, and (2) models pretrained on CLIMB. Both are then fine-tuned using few-shot samples from the target OOD dataset.

**RQ3: Unimodal pretraining to multimodal fusion.** We investigate how to effectively combine information from different clinical modalities (imaging, text, and time series) to improve patient outcome prediction. This addresses the practical clinical scenario where multiple types of medical data are available for diagnosis and prognosis. Our experiments pretrain models on CLIMB and transfer them to MIMIC-IV (Johnson et al., 2023), a large-scale multimodal clinical dataset. We evaluate three fusion strategies with increasing levels of cross-modal interaction: Late fusion, MLP fusion, and cross-attention fusion. Detailed architectural specifications are provided in App. D.2.3. We fix the encoder architectures across all experiments: ConvNextv2 for visual inputs, ClinicalBERT(Liu et al., 2025) for text, and ECG-JEPA for time series data. In addition, to evaluate how large scale pretraining helps with multimodal tasks across different encoders, we compare models initialized with our CLIMB pretrained weights against those using publicly available pretrained weights. We evaluate on two common clinical prediction tasks: length of stay (LOS) prediction and 48-hour in-hospital mortality prediction (48 IHM). These tasks are clinically significant and require integrating information across modalities.

**Evaluation metrics.** For consistency, we evaluate all classification tasks with balanced AUC, sensitivity, and specificity. Regression tasks (e.g., length of stay prediction) are evaluated with mean absolute error (MAE). Task specifications, metrics considerations and experimental setups are provided in App. D.

## 4.2. On RQ1: Multitask Pretraining Performance

**Importance of multitask pretraining.** Our experiments demonstrate that multitask pretraining yields substantial and consistent improvements across diverse clinical scenarios, as shown in Fig. 4. Notably, the most significant gains

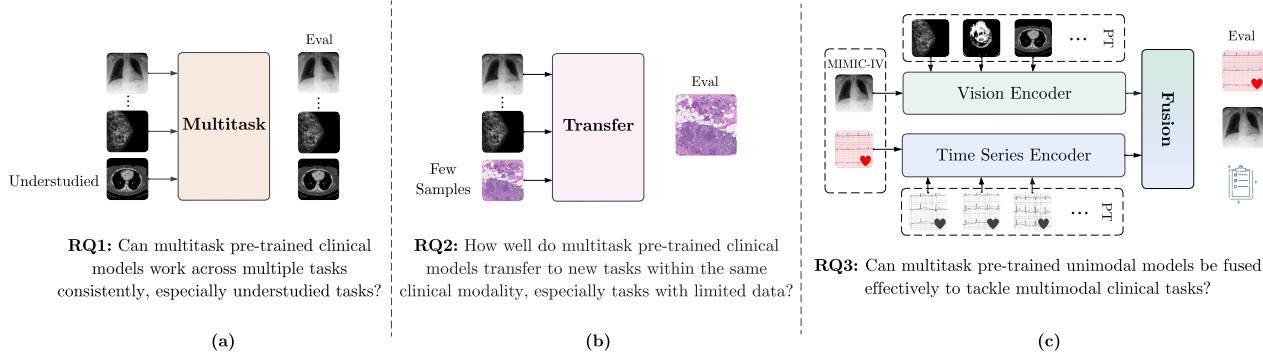

(a)         (b)         (c)

*Figure 3.* **Experimental setup for evaluating (a) multitask, (b) transfer, and (c) fusion learning strategies, addressing RQ1, 2, 3 respectively.** We show inputs to the model on the left and display evaluation on the right. (a) trains a single model on multiple clinical tasks across different medical modalities and evaluate on each individual dataset to see if model can generate across diverse tasks, including understudied ones. We compare the model's performance across different achitectures, as well as between the performance of single task learning versus multitask learning. (b) explores how well multitask pretrained clinical models transfer to new tasks, despite having few samples in the same modality. In this experiment, each model is trained on a few samples of a target out-of-distribution (OOD) dataset, as well as other tasks across different clinical modalities. (c) experiments on whether multitask pretrained unimodal models can be fused effectively to tackle multimodal clinical tasks. The source encoders are first separately trained with diverse data in their on modality, before being fused and evaluated on a new multimodal dataset (MIMIC-IV).

*Table 3.* **Performance comparison of trained multitask models across various medical imaging modalities.** Sen: Sensitivity, Spe: Specificity, AUC: Area under the receiver operating characteristic curve. CLIP-L2B: CLIP ViT-Laion 2B. The best performance for each metric is bolded. Detailed breakdowns of each dataset are attached in App. E.1. The experiment shows that general domain SoTA encoders perform better than specialized medical encoders, with ConvNeXTv2 offering the best performance-compute tradeoff.

| Model | Chest X-Ray | | | Mammography | | | Dermoscopy | | | CT Scan | | | Fundus | | | Ultrasound | | | Overall | | |
|---|---|---|---|---|---|---|---|---|---|---|---|---|---|---|---|---|---|---|---|---|---|
| | AUC | Sen | Spe | AUC | Sen | Spe | AUC | Sen | Spe | AUC | Sen | Spe | AUC | Sen | Spe | AUC | Sen | Spe | AUC | Sen | Spe |
| *Clinical Encoders* | | | | | | | | | | | | | | | | | | | | | |
| MedViT | .670 | .253 | .833 | **.627** | **.417** | .583 | .522 | .361 | .639 | .604 | .382 | .616 | .320 | .317 | .688 | .452 | .500 | .583 | .579 | .364 | .690 |
| PMC-CLIP | .725 | .251 | .883 | .614 | .312 | .710 | .674 | .325 | .706 | .619 | .407 | .593 | .508 | .220 | .785 | .521 | .384 | .609 | .635 | .341 | .724 |
| RAD-DINO | .818 | .406 | .928 | .566 | .314 | .701 | .717 | .348 | .715 | .653 | .408 | .594 | .606 | .221 | .786 | .639 | .431 | .619 | .681 | .368 | .729 |
| *General Domain Encoders* | | | | | | | | | | | | | | | | | | | | | |
| SBB2 | .791 | .401 | .922 | .538 | .262 | .663 | .784 | .362 | .724 | .691 | .403 | .590 | .732 | .293 | .821 | .711 | .495 | .689 | .730 | .420 | .754 |
| Swin Transformer | .795 | .389 | .926 | .513 | .200 | .599 | .815 | .435 | .747 | .685 | .429 | .615 | .770 | .327 | .838 | .705 | .545 | .712 | .765 | .436 | .775 |
| EVA-2 | **.863** | .382 | .929 | .516 | .320 | .699 | .716 | .353 | .724 | .531 | **.496** | .496 | .780 | .295 | .822 | .462 | .340 | .659 | .685 | .372 | .737 |
| InternViT | .815 | .413 | .930 | .532 | .340 | **.713** | .868 | .543 | .770 | **.706** | .469 | **.652** | .839 | .431 | .851 | .735 | .549 | .718 | .772 | .492 | .789 |
| ConvNeXTv2 | .817 | **.436** | **.939** | .558 | .330 | .706 | **.901** | **.568** | **.777** | .671 | .466 | .641 | **.873** | **.563** | **.888** | **.774** | **.641** | **.770** | **.787** | **.537** | **.806** |

*Table 4.* **Performance comparison of graph neural networks across brain networks and protein structures.** The best performance of each model is bolded. Graph transformer offers the best performance in terms of AUC and sensitivity-specificity balance.

| Model | BrainNet | | | Molecular | | | Overall | | |
|---|---|---|---|---|---|---|---|---|---|
| | AUC | Sen | Spe | AUC | Sen | Spe | AUC | Sen | Spe |
| GCN | .804 | .696 | .800 | .763 | .532 | .760 | .783 | .614 | .780 |
| GAT | .705 | **.916** | .404 | **.823** | **.551** | .801 | .764 | **.733** | .602 |
| Graph Transformer | **.852** | .810 | **.826** | .789 | .381 | **.920** | **.820** | .595 | **.873** |

are observed in the three understudied categories: novel tasks, understudied modalities, and datasets from under-represented regions, as defined in Sec. 3.1. Datasets that intersect multiple categories show the highest performance improvements, as exemplified by COVID-US with an AUC gain of 0.3254. In temporal modalities, particularly ECG analysis, the Ga dataset demonstrates this trend with an

absolute AUC improvement of 23 percentage points (from 0.474 to 0.704) when comparing single-task to multitask pretraining approaches, as shown in App. Table 30. These results suggest that multitask learning is particularly effective for scenarios where data or research attention has been historically limited.

**Comparison of encoder models.** Our analysis reveals counterintuitive patterns in encoder effectiveness across different domains. In the visual domain, image-based models consistently outperform video models across both image and video tasks, which we attribute to the substantially larger pretraining datasets available for image models. In addition, general-purpose architectures like ConvNextv2 significantly outperform clinical-specific encoders such as MedViT, achieving a 35.9% performance improvement. We hypothesize this superiority stems from the more diverse pretraining distribution encountered by general-domain en-

*Table 5.* **EEG Model Performance on Different EEG Datasets.** In general, pretrained models (starting with *BIOT-pretrain*) achieve better performances than models trained from scratch. However, there's no clear relationship between the number of pretrained datasets and the performance, indicating that quality and relevance of pretraining datasets may play a bigger role than the quantity of data.

| Model Name | IIIC | | | | TUEV | | | | TUAB | | | | Overall | | | |
|---|---|---|---|---|---|---|---|---|---|---|---|---|---|---|---|---|
| | AUC | Sens | Spe | F1 | AUC | Sens | Spe | F1 | AUC | Sens | Spe | F1 | AUC | Sens | Spe | F1 |
| SPARCNet | .846 | .525 | .905 | **.589** | .801 | .495 | .877 | .286 | .868 | .796 | .796 | .797 | .838 | .605 | .859 | .557 |
| CNNTransformer | .809 | .435 | .890 | .399 | .874 | .473 | .913 | .386 | .879 | .797 | .797 | .799 | .854 | .569 | .867 | .528 |
| FFCL | .841 | .458 | .805 | .448 | .801 | .443 | .906 | .347 | .874 | .787 | .787 | .789 | .839 | .562 | .833 | .528 |
| ContraWR | .832 | .446 | .892 | .410 | .847 | .439 | .898 | .370 | .872 | .782 | .782 | .781 | .850 | .556 | .858 | .520 |
| STTransformer | .785 | .412 | .884 | .407 | .701 | .371 | .874 | .262 | .864 | .785 | .785 | .787 | .783 | .522 | .848 | .485 |
| BIOT | .854 | .510 | .905 | .499 | .856 | .466 | .908 | .371 | .879 | .798 | .798 | .799 | .863 | .591 | .870 | .556 |
| BIOT-pretrain-PREST | .844 | .496 | .902 | .486 | **.898** | .580 | **.918** | .373 | .878 | .797 | .797 | .799 | **.873** | .624 | .872 | .552 |
| BIOT-pretrain-SHHS+PREST | .848 | .523 | .906 | .507 | .880 | **.586** | .914 | **.415** | **.882** | **.806** | **.806** | **.808** | .870 | **.639** | **.875** | **.577** |
| BIOT-pretrain-six-datasets | **.862** | **.546** | **.911** | .531 | .878 | .549 | .917 | .397 | .869 | .794 | .794 | .795 | .870 | .630 | .874 | .574 |

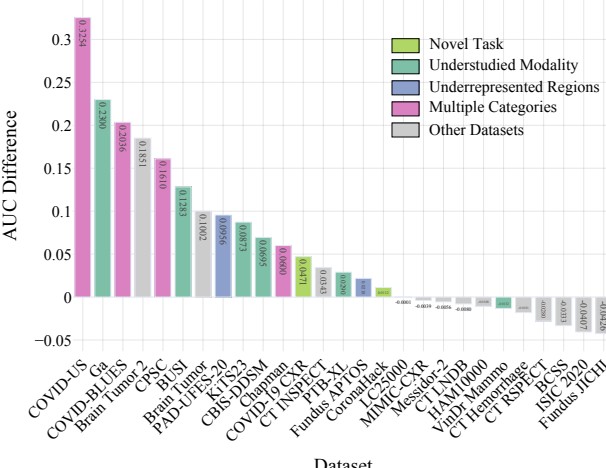

*Figure 4.* **Difference in AUC achieved by the multitask model compared to single-task training.** Novel Task represent emerging clinical challenges like COVID-19; Underrepresented Regions indicates datasets from underrepresented regions in developing countries such as Brazil and China; Understudied Modalities includes less common imaging types such as ultrasound and CT scans. Datasets belonging to multiple categories are highlighted in pink. In general, multitask learning helps the model to reach a better performance, with the greatest improvement observed in understudied tasks, regions, and/or modalities.

coders. In the EEG domain, we see a similar trend, with pretrained models achieving a better overall performance than all models trained from scratch, but models trained on more datasets (*BIOT-pretrain-six-datasets*) does not guarantee a better performance than models trained on less datasets. This indicates that in the EEG domain, the quality and relevance of the pretraining datasets is of importance. On the other hand, in the ECG domain, specialized architectures demonstrate clear advantages over general models, with ECG JEPA outperforming the general-purpose time series model UniTS by 36.8%. This dichotomy suggests that the optimal choice of architecture depends heavily on both the

*Table 6.* **Performance comparison of zero-shot and fine-tuned LLaVa-Med.** LLaVa-Med is the current SoTA LVLM model for clinical QA tasks. Both CLIMB-ConvNextv2 and LLaVa-Med are trained and evaluated on CLIMB in a comparable, close-ended manner. While directly fine-tuning the LLM on CLIMB-QA improves performance over zero-shot cases, they still lag behind SoTA multitask encoders, namely CLIMB-ConvNextv2, the SoTA multitask encoder trained on CLIMB.

| Dataset | Zero-Shot | | | Fine-Tuned | | |
|---|---|---|---|---|---|---|
| | Acc | Sens | Spe | Acc | Sens | Spe |
| Chest X-ray | .088 | .192 | .808 | .309 | .207 | .795 |
| MRI | .363 | .473 | .650 | .480 | .375 | .625 |
| Ultrasound | .448 | .427 | .640 | .579 | .389 | .611 |
| Mammography | .049 | .203 | .800 | .741 | .300 | .700 |
| Dermoscopy | .466 | .296 | .700 | .673 | .245 | .756 |
| Fundus | .434 | .202 | .794 | .578 | .217 | .783 |
| CT | .448 | .424 | .575 | .788 | .435 | .585 |
| Endoscopic | .000 | 1.00 | .000 | .296 | .143 | .857 |
| Average | .287 | .402 | .621 | .555 | .289 | .714 |
| **CLIMB-ConvNextv2** | - | - | - | **.877** | **.806** | **.787** |

modality and the availability of domain-specific pretraining data. In the graph domain, graph transformers work the best with the highest score across all metrics, as shown in Tab. 4.

**Comparing Vision Encoders with Large VLMs.** We evaluated LLaVA-Med on CLIMB-QA to assess current clinical VLMs' capabilities in multimodal understanding. While fine-tuning on CLIMB-QA improves performance over zero-shot inference by 28.7 percentage points, these results significantly lag behind CLIMB-ConvNeXtv2, our new SoTA vision encoder trained on CLIMB, by 32.2 percentage points in accuracy. The gap is particularly evident in modalities requiring fine-grained visual understanding, such as chest X-rays (30.9% accuracy) and endoscopic images (29.6% accuracy), where the model struggles to maintain balanced sensitivity and specificity. These results suggest

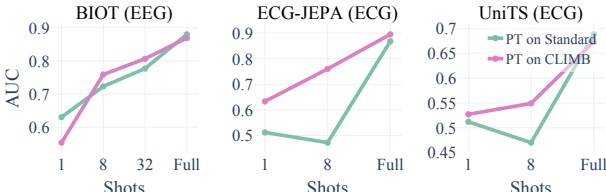

*Figure 5.* **Few-shot learning performance comparison across different time series domains.** We evaluate the few-shot performance (1-shot, 8-shot, and full dataset) of three representative models: BIOT for EEGs, ECG-JEPA and UniTS for ECGs. PT on Standard shows the performance when pretrained on their datasets from the original paper, while PT on CLIMB shows the performance when pretrained on our CLIMB dataset. Models pretrained on CLIMB demonstrate consistent improvements over the original ECG domain-specific models.

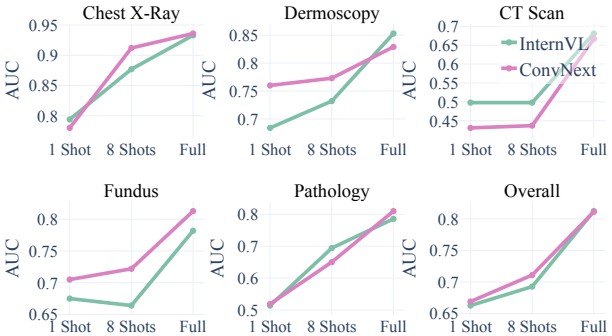

*Figure 6.* **Few-shot performance on out-of-distribution datasets.** InternVL represents the ViT-based model, while ConvNext represents the convolution-based model. In general, ViT-based models and ConvNext-based models exhibit similar overall performances across all shots, with ViT excelling at CT Scan tasks while ConvNext performs better at diagnosing fundus images.

that CLIMB is a rich data source for clinical AI that has a substantially different distribution than the pretraining dataset of current LLMs, and that training on CLIMB-QA can substantially improve existing models. However, the fine-tuned results also make it evident that current vision-language models require fundamental architectural innovations and novel training paradigms for the model to match the performance of fine-tuned dedicated vision encoders.

### 4.3. On RQ2: Few-shot Transfer Performance

**Strength of few-shot transfer.** As illustrated in Figure 7, our large-scale pretraining dataset enables efficient learning of novel tasks with limited samples, demonstrating consistent performance improvements across all modalities. The impact is particularly significant in traditionally challenging modalities such as CT scans and ultrasound imaging, where models achieved substantial gains in AUC of 28.7% and 29.1%, respectively. In time series domains such as ECG and EEG, our model outperformed SoTA weights specif-

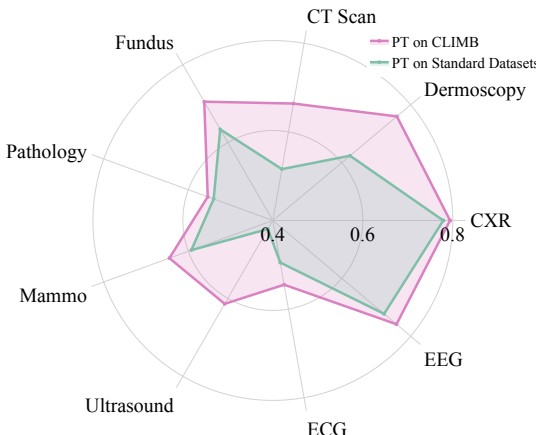

*Figure 7.* **Few-shot performance of models across different pre-training (PT) datasets.** CXR: Chest X-ray. We use ConvNextv2 for vision, ECG JEPA for ECG data, and BIOT for EEGs. PT on Standard Datasets is the performance of training one shot example on top of their own pretrained weights, while PT on CLIMB is the performance of training one shot example on top of our CLIMB dataset. For EEGs, 8 shot results are used instead due to the inherent complexity of the task.

ically trained on this domain, which again illustrates the benefits of our large-scale pertaining data.

**Architecture comparisons.** As shown in Figure 6, in the vision domain, we again found both ConvNextv2 and ViT-based models perform similarly under both full and few shot settings, as shown in Figure 6. ConvNext exhibit better performance in fundus and dermoscopy images, while ViT performs better for CT scans. We also found specialized ECG models like ECG-JEPA transfer better than universal time series models like UniTS, as demonstrated in Figure 5.

### 4.4. RQ3: Unimodal Pretraining to Multimodal Fusion

**Pretraining results.** Experimental results in Table 7 demonstrate that encoders pretrained on CLIMB consistently outperform those pretrained on other datasets across all evaluation settings. This performance advantage is maintained across diverse tasks, including length of stay prediction and in-hospital mortality prediction, both in full-data and few-shot scenarios. Notably, our pretrained encoders achieve 5.78% lower MAE (2.61 vs 2.77) in LOS prediction and 15.9% higher AUC in 8-shot mortality prediction, suggesting that the diverse modalities in CLIMB enable more robust feature representations.

**Fusion comparisons.** Our analysis reveals that the effectiveness of different fusion strategies varies with task complexity. For complex regression tasks like length of stay prediction, cross-attention mechanisms demonstrate superior performance, achieving the lowest MAE of 2.61. In contrast, for binary classification tasks such as 48-hour in-

Table 7. **Performance comparison of different multimodal fusion approaches on length of stay (LOS) prediction and 48-hour in-hospital mortality (48 IHM) prediction tasks.** CrossAtt: Cross Attention. Unimodal encoders trained on CLIMB transfers to better multimodal performance, given proper fusion strategies are used. Complex tasks like LOS require complex fusions like cross-attention, while simple tasks like 48 IHM work well under simple MLP fusion. FT = Fine Tuning.

| Enc. | Fusion | LOS | 48 IHM (Full FT) | | | 48 IHM (8-Shots) | | |
|---|---|---|---|---|---|---|---|---|
| | | MAE | AUC | Sens | Spec | AUC | Sens | Spec |
| **SoTA** | **Late** | 4.78 | 0.689 | 0.495 | 0.760 | 0.524 | 0.001 | **0.994** |
| | **MLP** | 2.98 | 0.957 | 0.806 | 0.979 | 0.556 | **0.536** | 0.538 |
| | **CrossAtt** | 2.77 | 0.786 | 0.628 | 0.814 | 0.580 | 0.286 | 0.766 |
| **Ours** | **Late** | 4.71 | 0.859 | 0.017 | **0.983** | 0.628 | 0.022 | 0.993 |
| | **MLP** | 2.84 | **0.961** | **0.824** | 0.975 | **0.672** | 0.295 | 0.858 |
| | **CrossAtt** | **2.61** | 0.796 | 0.822 | 0.590 | 0.570 | 0.294 | 0.753 |

hospital-mortality (48 IHM) prediction, MLP-based concatenation proves more effective, achieving the highest AUC while maintaining balanced sensitivity (0.824) and specificity (0.975). While late fusion appears to achieve higher specificity in some cases, its near-zero sensitivity indicates that it effectively defaults to predicting the majority class without any meaningful information. This pattern persists in the challenging 8-shot setting, where MLP fusion maintains the best performance while preserving a reasonable balance between sensitivity and specificity.

### 4.5. Comparison with Dataset-Specific SoTAs

To contextualize the performance of our pretrained unimodal and multimodal models, we compare them to prior reported results from state-of-the-art models specifically optimized for individual datasets in App. Table 36. While dataset-specific architectures sometimes outperform pretrained models through specialized optimizations, they often struggle to adapt to new tasks, even within the same modality it was trained on (see Sec. 4.2). The highly specialized nature of their architectures, like the multi-stage category-wise fine-tuning utilized in CFT (Chong et al., 2023), makes them difficult to adapt or retrain for different clinical tasks. Therefore, there is much value in training generalizable unimodal and multimodal models that can effectively adapt across diverse clinical modalities, tasks, and scenarios.

## 5. Conclusion

We present CLIMB, a comprehensive multimodal clinical benchmark unifying 4.51M samples across 44 datasets, 15 modalities, and 13 clinical domains. Our extensive empirical evaluation revealed several key insights for clinical AI development. First, multitask pretraining significantly improves performance on understudied modalities and novel tasks. Second, general-domain architectures outperform clinical-specific ones in multitask settings. Third, models pretrained on CLIMB demonstrate substantial im-

provements in few-shot scenarios across modalities. Finally, unimodal pretraining on CLIMB consistently enhances performance on downstream multimodal tasks. Based on these findings, we recommend leveraging general-domain architectures for visual tasks and emphasize the importance of multitask pretraining, especially for understudied domains. For multimodal applications, we suggest matching fusion complexity to task requirements and utilizing large-scale unimodal pretraining before multimodal integration.

Looking ahead, our findings point to several emerging research directions: developing novel architectures that better balance general and domain-specific features, finding new ways to combine unexplored modality combinations, and creating fusion mechanisms that adjust to task complexity. While dataset-specific models currently achieve higher performance through specialized optimizations, we encourage future research to develop general multitask encoders that can effectively adapt across diverse modalities and tasks. By releasing our dataset, code, and models, we hope to accelerate progress in these directions and advance the development of holistic clinical AI systems.

## Impact Statement

This paper presents empirical benchmarking, analysis, and development of multimodal clinical datasets and models. Multimodal AI can help clinicians analyze large-scale longitudinal data, make predictions, and investigate interventions. Furthermore, increasingly many indicators are no longer taken in the doctor's office, but daily, such as physiological sensors that track sleep, mood, stress, diet, exercise, and social interactions. Our findings can have a broad impact on developing holistic AI models of human health and wellness.

At the same time, data privacy and model fairness are critical qualities. There may be privacy risks associated with collecting and making predictions from multimodal clinical data. We have taken appropriate steps to only access data that participants have consented to public release, and to the best of our knowledge, all data was anonymized and stripped of all personal (e.g., personally identifiable information) and protected attributes (e.g., race, gender). There is also the risk that clinical AI models capture spurious features from race, gender, and other demographic variables, especially as more clinical data is provided. To deploy these algorithms at scale in the real world, it is also important to keep data and features secure without public sharing.

Overall, CLIMB offers opportunities to study the promises of multimodal AI while mitigating potential risks at scale across clinical modalities, tasks, and domains. We will continue expanding CLIMB to rigorously test for these social impacts and improve the safety and reliability of multimodal clinical models.

# Acknowledgments

We gratefully acknowledge NVIDIA for their generous support through GPU computing credits. We also extend our sincere thanks to Dr. Farzan Vahedifard (Neuroscientist, Athinoula A. Martinos Center for Biomedical Imaging, Harvard Medical School) for his insightful feedback on the clinical terminology used in our classification tasks, which significantly strengthened the rigor and clinical relevance of this work.

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

## A. Individual Dataset Details

*Table 8.* **Breakdown of CLIMB dataset by Modalities**

| Dataset | # Samples | Clinical Domain | Modality | Task | Fine-grained |
|---|---|---|---|---|---|
| PTB-XL | 21K | Cardiology | ECG | Diagnostics, Attribute Recognition | / |
| Chapman-Shaoxing | 40K | Cardiology | ECG | Diagnostics | / |
| Georgia | 20K | Cardiology | ECG | Diagnostics | / |
| CPSC | 6K | Sleep Cardiology | ECG | Abnormality Detection | / |
| IIIC | 134.5k | Neurological Disorders | EEG | Diagnostics | / |
| TUAB | 409.5k | Neurological Disorders | EEG | Abnormality Detection | / |
| TUEV | 111.9k | Neurological Disorders | EEG | Diagnostics | / |
| MIMIC-CXR | 356K | Radiology | Chest X-ray | Diagnostics, Classification | No |
| CheXpert | 212K | Radiology | Chest X-ray | Diagnostics, Classification | No |
| VinDr-CXR | 18K | Radiology | Chest X-ray | Diagnostics, Abnormality Detection | Both |
| COVID-19 | 2.9K | Radiology | Chest X-ray | Diagnostics | No |
| CoronaHack | 5.9K | Radiology | Chest X-ray | Diagnostics | No |
| VinDr-Mammo | 20K | Radiology, Oncology | Mammography | Diagnostics, Abnormality Detection | Both |
| CBIS-DDSM | 2.8K | Radiology, Oncology | Mammography | Diagnostics, Abnormality Detection, Classification | Both |
| CMMD | 1.8K | Radiology, Oncology | Mammography | Diagnostics, Segmentation, Abnormality Detection | Both |
| ISIC-2020 | 33K | Dermatology, Oncology | Dermoscopy | Diagnostics, Classification | No |
| HAM10000 | 10K | Dermatology, Oncology | Dermoscopy | Diagnostics, Segmentation | Both |
| PAD-UFES-20 | 2.3K | Dermatology, Oncology | Dermoscopy | Diagnostics, Classification | No |
| Messidor-2 | 1.7K | Ophthalmology | Fundus | Diagnostics | No |
| APTOS 2019 | 3.6K | Ophthalmology | Fundus | Diagnostics | No |
| Jichi | 9.9K | Ophthalmology | Fundus | Diagnostics, Prognostics, Severity Grading | No |
| LNDb | 5.6K | Radiology, Oncology | CT | Diagnostics, Abnormality Detection | Yes |
| INSPECT | 23K | Radiology | CT | Diagnostics, Prognostics | No |
| KiTS23 | 478 | Radiology, Oncology | CT | Segmentation | Yes |
| Hemorrhage | 2.5K | Radiology | CT | Diagnostics, Segmentation | Both |
| RSPECT | 1.79M | Radiology | CT | Diagnostics, Classification | Yes |
| EchoNet-Dynamic | 10K | Radiology | Ultrasound | Segmentation | Yes |
| BUSI | 780 | Radiology, Oncology | Ultrasound | Diagnostics, Segmentation | Both |
| COVID-BLUES | 362 | Radiology | Ultrasound | Diagnostics | No |
| COVID-US | 242 | Radiology | Ultrasound | Diagnostics, Severity Grading | No |
| Brain Tumor | 3.2K | Radiology, Oncology | MRI | Diagnostics | No |
| Brain MRI | 253 | Radiology, Oncology | MRI | Diagnostics | No |
| ABCD | 9.5K | Radiology | BrainNet | Classification | Yes |
| ABIDE | 1009 | Radiology | BrainNet | Classification | Yes |
| PPMI | 718 | Radiology | BrainNet | Classification | Yes |
| PROTEINS | 1113 | Molecular Biology | Molecule | Classification | No |
| PPI | 57K | Molecular Biology | Molecule | Classification | No |
| LC25000 | 18.7K | Pathology | Tissues | Classification | No |
| BCSS | 5.26K | Pathology | Tissues | Classification | No |
| Cholec80 | 14.4K | Surgery | Video | Workflow Analysis, Segmentation | Yes |
| HuGaDB | 364 | Physical Medicine, Rehabilitation | IMU | Motion Analysis, Classification | Yes |
| Expression Atlas | 4.5K | Molecular Biology, Genetics | Gene Expression | Expression Analysis, Classification | Yes |
| Geo | 126K | Molecular Biology, Genetics | Gene Expression | Expression Analysis, Classification | Yes |
| Vital | 210K | Multiple | Multimodal | Diagnostics | Both |
| MIMIC-IV | 800K | Cardiology,Radiology | EHR, ECG, X-ray | Diagnostics, Prognostics, Severity Grading | Both |

Table 8 shows a breakdown of the data sources in CLIMB. In this section, we provide details for each dataset. We describe the source and structure of the datasets, split used to evaluate the models, QA prompts, access restrictions, and licenses. A list of classes we constructed is included in Table 9. We also include a list of medical institutions and locations where the data was collected, as well as the demographic info available from each dataset in Table 10.

The definitions of each column are as follows:

**Clinical Domain and Modality.** Clinical Domain involves the clinical specialty and the exact diseases described by the data. Our dataset aims to cover as many clinical domains as possible. The final dataset spans 13 domains, including radiology, cardiology, pathology, dermatology, oncology, ophthalmology, molecular biology, sleep cardiology and neurological disorders. Within radiology, our dataset spans diseases such as breast cancer, kidney cancer, and pneumonia. Modality involves the type of data at play. Our data ranges from time series (1D), vision (2D & 3D), text records and graphs, making it one of the most diverse datasets up to date.

**Task** describes the applicable tasks for a multi-modal model. CLIMB spans diagnostics (disease labels), attribute recognition (recognition of inherent structures in the data), abnormality detection (localized abnomality labels), segmentation, prognostics (labels related improvement of condition over time), severity grading, and plain classification (non-disease related labels).

**Granularity** specifies the level of localization in the task involved. For example, tasks such as segmentation and abnormality

detection requiring localized reasoning and reference to a specific object in an image are classified as fine-grained. In contrast, tasks that make a prediction based on the entire image (e.g. diagnostics) are considered coarse-grained.

Here, we first introduce the list of modalities in CLIMB, followed by a detailed description of each individual dataset.

## A.1. List of Modalities

**ECG (Electrocardiogram)**   Electrocardiogram is a cardiac diagnostic tool that records the electrical activity of the heart over time. The datasets in this modality include {PTB-XL, Chapman-Shaoxing, Georgia, CPSC}, primarily focusing on cardiac diagnostics and abnormality detection. The collective classes across these datasets encompass Normal, Conduction Delay (CD), Hypertrophy (HYP), Myocardial Infarction (MI), Sinus Tachycardia/Bradycardia/Conduction (STTC), Atrial Fibrillation/Atrial Flutter (A. Fib/Aflutter), and Other conditions.

**EEG (Electroencephalogram)**   EEG is a neurological monitoring method that records brain's electrical activity through electrodes placed on the scalp. The datasets in this category include {IIIC, TUAB, TUEV}, used for diagnosing neurological disorders and detecting abnormalities. The classes across these datasets include Seizure (SZ), Lateralized Periodic Discharges (LPD), Generalized Periodic Discharges (GPD), Lateralized Rhythmic Delta Activity (LRDA), Generalized Rhythmic Delta Activity (GRDA), Spike and Slow Wave (SPSW), Generalized Periodic Epileptiform Discharge (GPED), Periodic Lateralized Epileptiform Discharge (PLED), Eye Movement (EYEM), Artifact (ARTF), Background (BCKG), and simple Normal/Abnormal classifications.

**Chest X-ray**   Chest X-ray is a radiological imaging technique used to examine the chest cavity, including the heart, lungs, and surrounding structures. The datasets in this modality include {MIMIC-CXR, CheXpert, VinDr-CXR, COVID-19, CoronaHack}. The comprehensive set of classes across these datasets includes Atelectasis, Cardiomegaly, Consolidation, Edema, Enlarged Cardiomediastinum, Fracture, Lung Lesion, Lung Opacity, Pleural Effusion, Pneumonia (including Bacterial and Viral), Pneumothorax, Pleural Other, Support Devices, Lung tumor, Tuberculosis, COPD, COVID-19, and No Finding.

**Mammography**   Mammography is a specialized medical imaging that uses low-dose X-rays to examine breast tissue for early detection of breast cancer. The datasets in this category include {VinDr-Mammo, CBIS-DDSM, CMMD}. The classes are primarily based on the BI-RADS scoring system (ranging from 0-5) and binary classification of Benign versus Malignant lesions.

**Dermoscopy**   Dermoscopy is a non-invasive skin imaging technique used for examining skin lesions and early detection of skin cancer. The datasets in this modality include {ISIC-2020, HAM10000, PAD-UFES-20}. The classes across these datasets include Melanoma (MEL), Nevus (NV), Basal Cell Carcinoma (BCC), Actinic Keratosis/Intraepithelial Carcinoma (AKIEC), and Other categories, with some datasets using simple Malignant/Benign classification.

**Fundus**   Fundus photography is a specialized technique for capturing detailed images of the interior surface of the eye, particularly useful in diagnosing retinal conditions. The datasets include {Messidor-2, APTOS 2019, Jichi}. The classes focus on different stages of Diabetic Retinopathy (DR), including None/No DR, Mild DR, Moderate DR, Severe DR, PDR (Proliferative DR), as well as SDR (simple diabetic retinopathy) and PPDR (pre-proliferative diabetic retinopathy).

**CT (Computed Tomography)**   CT is an advanced imaging technique that produces detailed cross-sectional images of the body. The datasets in this modality include {LNDb, INSPECT, KiTS23, Hemorrhage, RSPECT}. The classes across these datasets cover various conditions including nodule classification ($\geq$3mm, <3mm, non-nodule), Pulmonary Embolism (PE) categories (No PE, Acute PE, Chronic PE, Subsegmental PE), Hemorrhage detection, and tumor classification (Benign, Malignant).

**Ultrasound**   Ultrasound imaging uses high-frequency sound waves to produce real-time images of the inside of the body. The datasets include {EchoNet-Dynamic, BUSI, COVID-BLUES, COVID-US}. The classes across these datasets include Normal, Malignant, Benign, COVID-19, and Pneumonia, with some datasets focused on segmentation tasks rather than classification.

**MRI (Magnetic Resonance Imaging)**   MRI uses magnetic fields and radio waves to create detailed images of organs and tissues. The datasets in this category include {Brain Tumor, Brain MRI}. The classes focus on tumor detection and

classification, including No Tumor, Pituitary Tumor, Glioma Tumor, Meningioma Tumor, and simple presence/absence of tumors.

**BrainNet**   Brain Network represents brain connectivity networks derived from neuroimaging data. The datasets include {ABCD, ABIDE, PPMI}. The classes focus on binary classifications including Normal/Abnormal, ASD/Typical controls, and Control/PD patients.

**Molecule**   Molecular data represents structural and functional information about biological molecules. The datasets include {PROTEINS, PPI}, with classification focusing on enzyme/non-enzyme categorization for proteins and molecule property prediction.

**Tissues**   Tissue imaging from pathology involves microscopic examination of biological tissue samples. The datasets include {LC25000, BCSS}. The classes include various types of adenocarcinomas (Colon, Lung), Benign tissue (Colon, Lung), Lung squamous cell carcinomas, and tissue components (Tumor, Stroma, Lymphocytic infiltrate, Necrosis/debris).

**Video**   Medical video data captures dynamic medical procedures. The dataset in this category is {Cholec80}, which focuses on surgery phase annotations and tool labels rather than traditional classification tasks.

**IMU (Inertial Measurement Unit)**   IMU data captures motion and orientation information. The dataset {HuGaDB} includes classes for basic physical activities: Sitting, Standing, Sitting down, and Standing up.

**Gene Expression**   Gene expression data measures the activity levels of genes. The datasets {Expression Atlas, Geo} are not primarily used for classification tasks but rather for expression analysis.

**Multimodal**   Multimodal datasets combine multiple types of medical data. The datasets include {Vital, MIMIC-IV}, with MIMIC-IV specifically focusing on 48-hour In-Hospital-Mortality prediction (Yes/No) while combining EHR, ECG, and X-ray data.

## A.2. List of Datasets

1. **PTB-XL** (Wagner et al., 2020) is a dataset of 12-lead ECGs from 18,869 patients of 10 second length. The raw waveform data was annotated by two cardiologists, who assigned and validated diagnostic classification, form, and rhythm statements to each record. We provide a grouped label of 7 classes: Normal, CD, HYP, MI, STTC, A. Fib/ Aflutter and Other, following conventions from (Wantlin et al., 2023). For out-of-domain transfer learning, we utilize the subclass diagnostic labels from the PTB-XL dataset, which provides a more challenging 24-label classification task.
   **Split**: For multitask training, we use the BenchMD split, which includes label remapping to 7 diagnostic categories. This split consists of 17,476 records in the training set and 4,361 records in the test set, totaling 21,837 records.
   **Access restrictions**: The dataset is available to download from https://physionet.org/files/ptb-xl/1.0.3/
   **Licenses**: ECG records under this dataset are available in Creative Commons Attribution 4.0 International Public License https://creativecommons.org/licenses/by/4.0/legalcode
   **Ethical considerations**: No personally identifiable information or offensive content is present in the dataset.

2. **Chapman Shaoxing** (Zheng et al., 2020) consists of 12-lead ECGs from 10,646 patients, created under the auspices of Chapman University and Shaoxing People's Hospital. We provide a grouped label of 7 classes: Normal, CD, HYP, MI, STTC, A. Fib/ Aflutter and Other, following conventions from (Wantlin et al., 2023).
   **Split**: For multitask training, we use the BenchMD split, which includes label remapping to 7 diagnostic categories. The split consists of 38,207 records in the training set and 2,051 records in the test set, totaling 40,258 records.
   **Access restrictions**: The dataset is available to download from https://www.kaggle.com/datasets/erarayamorenzomuten/chapmanshaoxing-12lead-ecg-database.

   **Licenses**: ECG records under this dataset are available in Creative Commons Attribution 4.0 International Public License https://creativecommons.org/licenses/by/4.0/
   **Ethical considerations**: No personally identifiable information or offensive content is present in the dataset.

3. **Georgia** (Alday et al., 2020) is a database from the 2020 Physionet Computing in Cardiology Challenge, curated by Emory University. It consists of 12-lead ECGs from 15,742 patients of 10 second lengths and 500 Hz frequency, representing a unique demographic of the Southeastern United States. We provide a grouped label of 7 classes: Normal,

CD, HYP, MI, STTC, A. Fib/ Aflutter and Other, following conventions from (Wantlin et al., 2023).

**Split**: For multitask training, we use the BenchMD split, which includes label remapping to 7 diagnostic categories. The split consists of 18,622 records in the training set and 2,067 records in the test set, totaling 20,689 records.

**Access restrictions**: The dataset is available to download from https://www.kaggle.com/datasets/bjoernjostein/georgia-12lead-ecg-challenge-database. **Licenses**: ECG records under this dataset are available in Creative Commons Public Domain License https://creativecommons.org/publicdomain/zero/1.0/

**Ethical considerations**: No personally identifiable information or offensive content is present in the dataset.

4. **CPSC** (Liu et al., 2018) is a database from the 2021 China Physiological Signal Challenge. It consists of 12-lead Holter and 3-lead wearable ECG monitoring device recordings of variable lengths, each sampled at 200 Hz. We provide a grouped label of 7 classes: Normal, CD, HYP, MI, STTC, A. Fib/ Aflutter and Other, following conventions from (Wantlin et al., 2023).

    **Split**: For multitask training, we use the BenchMD split, which includes label remapping to 7 diagnostic categories. The split consists of 4,815 records in the training set and 1,377 records in the test set, totaling 6,192 records.

    **Access restrictions**: The dataset is available to download from https://physionet.org/files/cpsc2021/1.0.0/.

    **Licenses**: ECG records under this dataset are available in Creative Commons Attribution 4.0 International Public License https://creativecommons.org/licenses/by/4.0/legalcode

    **Ethical considerations**: No personally identifiable information, hospital identification, or offensive content is present in the dataset.

5. **IIIC** (Jing et al., 2023b) consists of EEG samples collected from 2,711 patients at Massachusetts General Hospital, annotated by 124 raters. The publicly available version includes 134,450 EEG segments from 1,950 patients, each segment lasting 10 seconds. The test set is non-public. Our evaluation focuses on the 6 diagnostic categories: seizure (SZ), lateralized periodic discharges (LPD), generalized periodic discharges (GPD), lateralized rhythmic delta activity (LRDA), generalized rhythmic delta activity (GRDA), and "Other" if none of those patterns was present.

    **Split**: Since the test dataset is not publicly available, we divide patient groups into training/validation/test sets by 60%:20%:20%.

    **Access restrictions**: The dataset is available to download from https://bdsp.io/content/bdsp-sparcnet/1.1/.

    **Licenses**: This dataset is available in the BDSP Restricted Health Data License 1.0.0. https://bdsp.io/content/bdsp-sparcnet/view-license/1.1/.

    **Ethical considerations**: No personally identifiable information or offensive content is present in the dataset.

6. **TUAB** (Lopez et al., 2015) is a dataset from the Temple University EEG Corpus. The dataset consists of 276 EEG recording sessions from 253 subjects. Each session is segmented into 10-second samples using event markers. We evaluate the models based on the binary classification labels "Normal" and "Abnormal."

    **Split**: The training and test separation is provided by the dataset.

    **Access restrictions**: The dataset is available to download from https://isip.piconepress.com/projects/nedc/html/tuh_eeg/.

    **Licenses**: Users must apply using the form described on https://isip.piconepress.com/projects/nedc/html/tuh_eeg/.

    **Ethical considerations**: No personally identifiable information or offensive content is present in the dataset.

7. **TUEV** (Lopez et al., 2016) is a dataset from the Temple University EEG Corpus. The dataset consists of 518 EEG recording sessions from 368 subjects. Each session is segmented into 5-second samples using event markers. We evaluate the models based on the 6 event categories: spike and slow wave (SPSW), generalized periodic epileptiform discharge (GPED), periodic lateralized epileptiform discharge (PLED), eye movement (EYEM), artifact (ARTF), and background (BCKG).

    **Split**: The training and test separation is provided by the dataset.

    **Access restrictions**: The dataset is available to download from https://isip.piconepress.com/projects/nedc/html/tuh_eeg/.

    **Licenses**: Users must apply using the form described on https://isip.piconepress.com/projects/nedc/html/tuh_eeg/.

    **Ethical considerations**: No personally identifiable information or offensive content is present in the dataset.

8. **MIMIC-CXR** (Johnson et al., 2019) is a dataset of chest X-rays in JPG format. Our evaluation utilizes the 14 disease labels: "Atelectasis", "Cardiomegaly", "Consolidation", "Edema", "Enlarged Cardiomediastinum", "Fracture", "Lung Lesion", "Lung Opacity", "Pleural Effusion", "Pneumonia", "Pneumothorax", "Pleural Other", "Support Devices", and "No Finding".

    **Split**: We use a training set of 348,516 records, test set of 7,709 records, and total size of 356,225 records.

    **Access restrictions**: The dataset is available to download from https://physionet.org/content/mimic-cxr-jpg/2.0.0/ with a credentialed account and CITI Data or Specimens Only Research training

**Licenses**: Radiology images under this dataset are available in PhysioNet Credentialed Health Data License 1.5.0 https://physionet.org/content/vindr-cxr/view-license/1.0.0/

**Ethical considerations**: No personally identifiable information or offensive content is present in the dataset.

9. **CheXpert** (Irvin et al., 2019b) is a chest radiology dataset collected from Stanford Hospital, covering 65,240 patients and 224,316 radiographs. The original dataset labels each record with a uncertainty level for 14 diagnostic observations including Atelectasis, Cardiomegaly, Consolidation, Edema, Enlarged Cardiomediastinum, Fracture, Lung Lesion, Lung Opacity, Pleural Effusion, Pneumonia, Pneumothorax, Pleural Other, Support Device and No Finding. Our evaluation focuses on predicting the labels that are "positive".

   **Split**: We use a training set of 212,243 records, a test set 225 records, and a total size of 212,498 records.

   **Access restrictions**: The dataset is available to download from https://stanfordaimi.azurewebsites.net/datasets/8cbd9ed4-2eb9-4565-affc-111cf4f7ebe2 with registration

   **Licenses**: Radiology images under this dataset are available under the Stanford University Dataset Research Use Agreement https://stanfordaimi.azurewebsites.net/datasets/8cbd9ed4-2eb9-4565-affc-111cf4f7ebe2

   **Ethical considerations**: No personally identifiable information or offensive content is present in the dataset.

10. **VinDr-CXR** (Nguyen et al., 2021) consists of adult chest X-rays collected from Hospital 108 and Hanoi Medical University Hospital in Vietnam. The dataset contains local labels for bounding boxes, however we evaluate our models based on the 6 global labels: "Lung tumor", "Pneumonia", "Tuberculosis", "COPD", "Other diseases", and "No finding", all annotated by 17 radiologists with at least 8 years of experience.

    **Split**: We use a training set of 15,000 records, a test set of 3,000 records, and a total size of 18,000 records.

    **Access restrictions**: The dataset is available to download from https://physionet.org/content/vindr-cxr/1.0.0/ with a credentialed account and CITI Data or Specimens Only Research training

    **Licenses**: Radiology images under this dataset are available in PhysioNet Credentialed Health Data License 1.5.0 https://physionet.org/content/vindr-cxr/view-license/1.0.0/

    **Ethical considerations**: No personally identifiable information or offensive content is present in the dataset.

11. **COVID-19** is a chest X-ray dataset for COVID-19 related diseases. We evaluate the models based on the diagnostic labels: "Normal", "Bacterial Pneumonia", "COVID-19", and "Viral Pneumonia".

    **Split**: We use a training set of 2,002 records, a test set of 988 records, and a total size of 2,990 records.

    **Access restrictions**: The dataset is available to download from https://www.kaggle.com/datasets/darshan1504/covid19-detection-xray-dataset.

    **Licenses**: This dataset is available in the Creative Commons Attribution 4.0 International License https://creativecommons.org/licenses/by/4.0/

    **Ethical considerations**: No personally identifiable information or offensive content is present in the dataset.

12. **CoronaHack** (Cohen et al., 2020) is a chest X-ray dataset compiled at the University of Montreal. Our evaluation utilizes the diagnosis labels: "Normal", "Bacterial Pneumonia", and "Viral Pneumonia".

    **Split**: We use a training set of 5,284 records, a test set of 624 records, and a total size of 5,908 records.

    **Access restrictions**: The dataset is available to download from https://www.kaggle.com/datasets/praveengovi/coronahack-chest-xraydataset.

    **Licenses**: This dataset is available in the Creative Commons Attribution 4.0 International License https://creativecommons.org/licenses/by/4.0/

    **Ethical considerations**: No personally identifiable information or offensive content is present in the dataset.

13. **VinDr-Mammo** (Pham et al., 2022) consists of mammography collected from Hospital 108 and Hanoi Medical University Hospital in Vietnam. The dataset contains local labels for bounding boxes, however we evaluate our models based on the 5 global labels for BI-RAD 1-5.

    **Split**: We use a training set of 16,000 records, a test set of 4,000 records, and a total size of 20,000 records.

    **Access restrictions**: The dataset is available to download from https://www.physionet.org/content/vindr-mammo/1.0.0/ with a data use agreement

    **Licenses**: Images under this dataset are available in PhysioNet Restricted Health Data License 1.5.0 https://www.physionet.org/content/vindr-mammo/view-license/1.0.0/

    **Ethical considerations**: No personally identifiable information or offensive content is present in the dataset.

14. **CBIS-DDSM** (R. et al., 2016) is a curated subset of the Digital Database for Screening Mammography (DDSM). Our evaluation focuses on the BI-RAD labels (0-5).

**Split**: We use a training set of 2230 records, a test set of 595 records, and a total of 2,825 records.

**Access restrictions**: The dataset is available to download from https://www.cancerimagingarchive.net/collection/cbis-ddsm/

**Licenses**: Images under this dataset are available in Creative Commons Attribution 3.0 Unported License https://creativecommons.org/licenses/by/3.0/

**Ethical considerations**: No personally identifiable information or offensive content is present in the dataset.

15. **CMMD** (Cui et al., 2021) is a breast mammography dataset for 1,775 patients from China. Our evaluation utilizes the diagnostic labels, "Benign" and "Malignant", which are confirmed through biopsy. However, due to issues found in the provided labels, while this dataset is a part of CLIMB, it is not a part of the experiment while pending expert verifications.

    **Split**: We use a training set of 1,404 records, a test set of 468 records, and a total size of 1,872 records.

    **Access restrictions**: The dataset is available to download from https://www.cancerimagingarchive.net/collection/cmmd/.

    **Licenses**: This dataset is available in the Creative Commons Attribution 4.0 International License https://creativecommons.org/licenses/by/4.0/

    **Ethical considerations**: No personally identifiable information or offensive content is present in the dataset.

16. **ISIC-2020** (Rotemberg et al., 2021) consists of dermoscopy of skin lesions from over 2000 patients, generated by the International Skin Imaging Collaboration (ISIC). We evaluate the models on the binary classification ("Malignant" or "Benign") for each image, where all malignant diagnoses have been confirmed through histopathology, and benign diagnoses have been confirmed using either expert agreement, longitudinal follow-up, or histopathology.

    **Split**: We use a training set of 26,501 records, a test set of 6,625 records, and a total size of 33,126 records.

    **Access restrictions**: The dataset is available to download from https://challenge2020.isic-archive.com

    **Licenses**: Images under this dataset are available in Creative Commons Attribution-Noncommercial 4.0 International License https://creativecommons.org/licenses/by-nc/4.0/

    **Ethical considerations**: No personally identifiable information or offensive content is present in the dataset.

17. **HAM10000** (Tschandl et al., 2018) is a dataset from the ISIC 2018 classification challenge, comprising dermoscopy images of pigmented lesions from from the ISIC archive. Our evaluation focuses on the 5 diagnostic categories: Melanoma (MEL), Nevus (NV), Basal Cell Carcinoma (BCC), Actinic Keratosis/Intraepithelial Carcinoma (AKIEC), Other (OTHER)

    **Split**: We use a training set of 8,012 records, a test set of 2,003 records, and a total size of 10,015 records.

    **Access restrictions**: The dataset is available to download from https://www.kaggle.com/datasets/kmader/skin-cancer-mnist-ham10000

    **Licenses**: Images under this dataset are available in Creative Commons Attribution-Noncomercial-Sharealike 4.0 International License https://creativecommons.org/licenses/by-nc-sa/4.0/

    **Ethical considerations**: No personally identifiable information or offensive content is present in the dataset.

18. **PAD-UFES-20** (Pacheco et al., 2020) consists of dermoscopy images of 1641 skin lesions from 1373 patients. We evaluate the models on the 5 skin diagnostics, three of which are skin disease and three of which are skin cancers: Melanoma (MEL), Nevus (NV), Basal Cell Carcinoma (BCC), Actinic Keratosis/Intraepithelial Carcinoma (AKIEC), Other (OTHER). All of the skin cancers are biopsy-proven, and more than half of the skin disease are biopsy-proven as well.

    **Split**: We use a training set of 1,839 records, a test set of 459 records, and a total size of 2,298 records.

    **Access restrictions**: The dataset is available to download from https://www.kaggle.com/datasets/mahdavi1202/skin-cancer

    **Licenses**: Images under this dataset are available in Creative Commons Attribution 4.0 International License https://creativecommons.org/licenses/by/4.0/

    **Ethical considerations**: No personally identifiable information or offensive content is present in the dataset.

19. **Messidor-2** (Abràmoff et al., 2013) (Decencière et al., 2014) is a dataset of Diabetic Retinopathy (DR) examinations, where each record consists of two macula-centered eye fundus images. The dataset is kindly provided by the Messidor program partners (see https://www.adcis.net/en/third-party/messidor/). We utilize the 5 point ICDR grades: "None", "Mild DR", "Moderate DR", "Severe DR", and "PDR".

    **Split**: We use a training set of 1,394 records, a test set of 350 records, and a total size of 1,744 records.

    **Access restrictions**: The dataset is available to download from https://www.kaggle.com/datasets/google-brain/messidor2-dr-grades.

**Licenses**: Images under this dataset are available in Creative Commons Public Domain License https://creativecommons.org/publicdomain/zero/1.0/
**Ethical considerations**: No personally identifiable information or offensive content is present in the dataset.

20. **APTOS 2019** (Asia Pacific Tele-Ophthalmology Society, 2019) is a dataset from the 4th Asia Pacific Tele-Ophthalmology Society Symposium, collected from rural India. The dataset consists of fundus images under a variety of imaging conditions. Our evaluation focuses on the 5 diabetic retinopathy ratings: "No DR", "Mild", "Moderate", "Severe", and "Proliferative DR".
    **Split**: We use a training set of 2,929 records, a test set of 733 records, and a total size of 3,662 records.
    **Access restrictions**: The dataset is available to download from https://www.kaggle.com/competitions/aptos2019-blindness-detection/data.
    **Licenses**: Images under this dataset are available under the Kaggle Competition Rules https://www.kaggle.com/competitions/aptos2019-blindness-detection/rules#7-competition-data
    **Ethical considerations**: No personally identifiable information or offensive content is present in the dataset.

21. **Jichi** (Takahashi et al., 2017) is a posterior pole fundus photography dataset collected at the Jichi Medical University in Japan, covering a total of 2740 patients. We evaluate the models based on the David Grading for each image: SDR (simple diabetic retinopathy), PPDR (pre-proliferative diabetic retinopathy), and PDR (proliferative diabetic retinopathy).
    **Split**: We use a training set of 7,950 records, a test set of 1,989 records, and a total size of 9,939 records.
    **Access restrictions**: The dataset is available to download from https://pmc.ncbi.nlm.nih.gov/articles/PMC5480986/#notes1.
    **Licenses**: Images under this dataset are available under the Creative Commons Attribution 4.0 International License. https://creativecommons.org/licenses/by/4.0/
    **Ethical considerations**: No personally identifiable information or offensive content is present in the dataset.

22. **LNDb** (Pedrosa et al., 2023) is lung cancer CT scan dataset collected at the Centro Hispitalar e Universitario de Sao Joao in Portugal between 2016 and 2018. Our evaluation focuses on the pulmonary nodule labels created by radiologists, including "nodule ≥3mm", "nodule <3mm", and "non-nodule".
    **Split**: We use a training set of 4,130 records, a test set of 1,431 records, and a total size of 5,561 records.
    **Access restrictions**: The dataset is available to download from https://zenodo.org/records/8348419.
    **Licenses**: Images under this dataset are available under the Creative Commons Attribution 4.0 International License. https://creativecommons.org/licenses/by/4.0/
    **Ethical considerations**: No personally identifiable information or offensive content is present in the dataset.

23. **INSPECT** (Huang et al., 2023) is multi-modal dataset containing CT images, radiology report impression sections, and structured electronic health records (EHR) from 19,438 patients. We focus on the pulmonary embolism (PE) labels which include "No PE", "Acute Subsegmental-only PE", "Acute PE", "Subsegmental-only PE", and "Chronic PE".
    **Split**: We use a training set of 17,434 records, a test set of 5,806 records, and a total size of 23,240 records.
    **Access restrictions**: The dataset is available to download from https://stanfordaimi.azurewebsites.net/datasets/318f3464-c4b6-4006-9856-6f48ba40ad67 with registration.
    **Licenses**: This dataset is available under the Stanford University Dataset Research Use Agreement https://stanfordaimi.azurewebsites.net/datasets/318f3464-c4b6-4006-9856-6f48ba40ad67
    **Ethical considerations**: No personally identifiable information or offensive content is present in the dataset.

24. **KiTS23** (Heller et al., 2023) is a dataset from the 2023 Kidney Tumor Segmentation Challenge. The dataset consists of CT videos showing kidney tumors. Although the data contains metrics and labels for segmentation tasks, we evaluate the models based on the "Benign" and "Malignant" key of each patient.
    **Split**: We use a training set of 361 records, a test set of 117 records, and a total size of 478 records.
    **Access restrictions**: The dataset is available to download from https://github.com/neheller/kits23.
    **Licenses**: This dataset is available in the Creative Commons Attribution-NonCommercial-ShareAlike 4.0 International License https://creativecommons.org/licenses/by-nc-sa/4.0/
    **Ethical considerations**: No personally identifiable information or offensive content is present in the dataset.

25. **Hemorrhage** (Hssayeni et al., 2020) consists of Intracranial hemorrhage CT images for 82 patients at Al Hilla Teaching Hospital, Iraq, each with brain and bone window images and approximately 30 image slices in total. We evaluate the models on the diagnosis labels "No Hemorrhage" and "Has Hemorrhage".
    **Split**: We use a training set of 1,986 records, a test set of 515 patient records, and a total size of 2,501 records.

**Access restrictions**: The dataset is available to download from https://www.kaggle.com/datasets/vbookshelf/computed-tomography-ct-imagesc .

**Licenses**: This dataset is available in the Creative Commons Attribution 4.0 International Public License https://physionet.org/content/ct-ich/view-license/1.0.0/

**Ethical considerations**: No personally identifiable information or offensive content is present in the dataset.

26. **RSPECT** (Colak et al., 2021) consists of CT scans for patients from five different countries suspected of Pulmonary Embolism (PE), created by the Radiological Society of North America (RSNA) and the Society of Thoracic Radiology (STR). We evaluate the models on the diagnosis labels "No PE", "Chronic PE", and "Acute PE", which are annotated by expert thoracic radiologists.

    **Split**: We use a training set of 1,342,945 records, a test set of 447,649 records, and a total size of 1,790,594 records.

    **Access restrictions**: The dataset is available to download from https://www.kaggle.com/c/rsna-str-pulmonary-embolism-detection/data.

    **Licenses**: This dataset is available in under the Kaggle competition rules https://www.kaggle.com/competitions/rsna-str-pulmonary-embolism-detection/data

    **Ethical considerations**: No personally identifiable information or offensive content is present in the dataset.

27. **EchoNet-Dynamic** (Ouyang et al., 2020) consists of 10,030 apical-4-chamber echocardiography videos from patients who underwent imaging between 2016 and 2018 as part of routine clinical care at Stanford University Hospital. Each video comes with two pairs of human tracings used to estimate ventricular volume—the first pair representing the left ventricle, and subsequent coordinate pairs representing short axis linear distances starting from the apex of the heart to the mitral apparatus.

    **Split**: We use a training set of 8,196 records, a test of 2,051 records, and a total size of 10,247 records.

    **Access restrictions**: The dataset is available to download from https://stanfordaimi.azurewebsites.net/datasets/834e1cd1-92f7-4268-9daa-d359198b310a with registration.

    **Licenses**: This dataset is available under the Stanford University Dataset Research Use Agreement https://stanfordaimi.azurewebsites.net/datasets/834e1cd1-92f7-4268-9daa-d359198b310a

    **Ethical considerations**: No personally identifiable information or offensive content is present in the dataset.

28. **BUSI** (Al-Dhabyani et al., 2020) is a breast cancer ultrasound image dataset from 600 female patients between 25 and 75 years old in 2018. We utilize the labels "Normal", "Malignant", and "Benign".

    **Split**: We use a training set of 583 records, a test set of 197 records, and a total size of 780 records.

    **Access restrictions**: The dataset is available to download from https://www.kaggle.com/datasets/aryashah2k/breast-ultrasound-images-dataset.

    **Licenses**: This dataset is available in the Creative Commons Public Domain License https://creativecommons.org/publicdomain/zero/1.0/

    **Ethical considerations**: No personally identifiable information or offensive content is present in the dataset.

29. **COVID-BLUES** (Wie, 2021) consists of bluepoint-specific lung ultrasound videos for 63 patients at the Maastricht University Medical Center in the Netherlands, each with 6 recordings. Our evaluation focuses on two labels: the diagnostic label ("Has COVID", "No COVID"), and the patient age label.

    **Split**: We use a training set of 266 records, a test set of 96 records, and a total size of 362 records.

    **Access restrictions**: The dataset is available to download from https://github.com/NinaWie/COVID-BLUES?tab=readme-ov-file.

    **Licenses**: This dataset is available in the Creative Commons Attribution-Noncommercial-NoDerivatives 4.0 International License https://creativecommons.org/licenses/by-nc-nd/4.0/

    **Ethical considerations**: No personally identifiable information or offensive content is present in the dataset.

30. **COVID-US** (Ebadi et al., 2022) consists of 150 COVID-related lung ultrasound videos. We evaluate the models based on the diagnostic labels: "Covid", "Pneumonia", and "Normal".

    **Split**: We use a training set of 74 records, a test set of 25 records, and a total size of 99 records.

    **Access restrictions**: The dataset is available to download from https://github.com/nrc-cnrc/COVID-US.

    **Licenses**: This dataset is available in the GNU Affero General Public License 3.0 https://www.gnu.org/licenses/agpl-3.0.en.html

    **Ethical considerations**: No personally identifiable information or offensive content is present in the dataset.

31. **Brain Tumor** (Bhuvaji et al., 2020) consists of brain MRI images. Each image is labeled as either "No Tumor",

"Pituitary Tumor", "Glioma Tumor", or "Meningioma Tumor".
**Split**: We use a training set of 2,870 records, a test set of 394 records, and a total size of 3,264 records.
**Access restrictions**: The dataset is available to download from https://www.kaggle.com/datasets/sartajbhuvaji/brain-tumor-classification-mri?select=Testing.
**Licenses**: This dataset is available in the MIT License https://www.mit.edu/ amini/LICENSE.md
**Ethical considerations**: No personally identifiable information or offensive content is present in the dataset.

32. **Brain MRI** is a brain MRI image dataset where each image is labeled with the presence of tumors, either as "Yes" or "No".
    **Split**: We use a training set of 202 records, a test set of 51 records, and a total size of 253 records.
    **Access restrictions**: The dataset is available to download from https://www.kaggle.com/datasets/jjprotube/brain-mri-images-for-brain-tumor-detection.
    **Licenses**: This dataset does not have a license.
    **Ethical considerations**: No personally identifiable information or offensive content is present in the dataset.

33. **ABCD** (Cui et al., 2022) is a study supported by the NIH on adolescent brain cognitive development on nearly 12,000 youths of ages 9-10, who were studied for 10 years. The dataset contains MRI images, behavioral and cognitive assessments, mental health, and other environmental factors data, with labels such as mental health diagnosis. In our dataset, we do a mental health diagnosis that classifies the sample into normal and abnormal.
    **Split**: The dataset has a total of 9563 records. We randomly split the dataset into a train set of 7650 and a test dataset of 1613 records.
    **Access restrictions**: The dataset is available to download from Link.
    **Licenses**: The license is available at Link.
    **Ethical considerations**: No personally identifiable information or offensive content is present in the dataset.

34. **ABIDE** (Cui et al., 2022) is a autism brain MRI diagnosis dataset with 1112 samples, including 539 from individuals with ASD and 573 from typical controls.
    **Split**: We use a random split to build a training set of 807 records, a test set of 202 records, and a total size of 1009 records.
    **Access restrictions**: The dataset is available to download from https://fcon_1000.projects.nitrc.org/indi/abide/ with account registration.
    **Licenses**: The dataset is available in the Creative Commons Attribution-NonCommercial-Share Alike License https://creativecommons.org/licenses/by-nc-sa/3.0/.
    **Ethical considerations**: No personally identifiable information or offensive content is present in the dataset.

35. **PPMI** (Cui et al., 2022) is a multi-center, longitudinal study dedicated to understanding the progression of Parkinson's disease. This data is derived from a cohort of 1694 subjects, broken down into 309 controls and 1385 PD patients.
    **Split**: After curation, we use random split to build a training set of 572 records, a test set of 143 records, and a total size of 718 records.
    **Access restrictions**: The dataset is available to download from Link.
    **Licenses**: The license is available in the Link.
    **Ethical considerations**: No personally identifiable information or offensive content is present in the dataset.

36. **PROTEINS** (Borgwardt et al., 2005) consists of 1,113 graphs where the nodes represent amino acids, and two nodes are connected by an edge if they are less than 6 Angstroms apart. We evaluate our models on the binary classification labels of whether a protein is an enzyme or not.
    **Split**: We use random split to build a training set of 890 records, a test set of 223 records, and a total size of 1,113 records.
    **Access restrictions**: The dataset is available to download from https://paperswithcode.com/dataset/proteins.
    **Licenses**: The dataset does not have a license.
    **Ethical considerations**: No personally identifiable information or offensive content is present in the dataset.

37. **PPI** (Stark et al., 2006) is a protein dataset from BioGRID covering physical and genetic interaction of proteins.
    **Split**: We use a training set of 45555 records, a test set of 11389 records, and a total size of 56944 records.
    **Access restrictions**: The dataset is available to download from https://snap.stanford.edu/graphsage/#datasets.
    **Licenses**: The dataset is available in the MIT License https://biogrid-downloads.nyc3.digitaloceanspaces.com/LICENSE.txt.

**Ethical considerations**: No personally identifiable information or offensive content is present in the dataset.

38. **LC25000** (Borkowski et al., 2019) is a lung and colon histopathological image dataset containg labels for colon adenocarcinomas, benign colon, lung adenocarcinomas, lung squamous cell carcinomas, and benign lung.
   **Split**: We use a training set of 15,000 records, a test set of 3,750 records, and a total size of 18,750 records.
   **Access restrictions**: The dataset is available to download from https://github.com/tampapath/lung_colon_image_set.
   **Licenses**: The dataset does not have a license.
   **Ethical considerations**: No personally identifiable information or offensive content is present in the dataset.

39. **BCSS** contains 151 breast cancer slides from 25 participants. We curate 5264 non-overlapping samples, with labels tumor, stroma, lymphocytic infiltrate, and necrosis/debris
   **Split**: We use a training set of 4211 records, a test set of 1053 records, and a total size of 5264 records.
   **Access restrictions**: The dataset is available to download from Link.
   **Licenses**: This dataset is licensed under a CC0 1.0 Universal (CC0 1.0) license.
   **Ethical considerations**: No personally identifiable information or offensive content is present in the dataset.

40. **Cholec80** (Twinanda et al., 2016) consists of 80 videos of cholecystectomy surgeries performed by 13 surgeons. We evaluate the models based on the surgery phase annotations (at 25 fps) and the surgery tool labels (at 1 fps).
   **Split**: We use a training set of 5,760 records, a test set of 1,440, and a total size of 7,200 records.
   **Access restrictions**: The dataset is available to download from http://camma.u-strasbg.fr/datasets/ through a request form.
   **Licenses**: This dataset is available in the Creative Commons Attribution-Noncomercial-Sharealike 4.0 International License https://creativecommons.org/licenses/by-nc-sa/4.0/legalcode.en
   **Ethical considerations**: No personally identifiable information or offensive content is present in the dataset.

41. **HuGaDB** (Chereshnev & Kertész-Farkas, 2018) is a dataset for human gait analysis collected from 18 healthy young adults using six wearable inertial sensors and two EMG sensors. It contains labels "sitting", "standing", "sitting down", and "standing up". However, due to the limitation of the number of samples, this gait dataset is not included in the experiments.
   **Split**: We use a training set of 291 records, a test set of 73 records, and a total size of 364 records.
   **Access restrictions**: The dataset is available to download from https://github.com/romanchereshnev/HuGaDB.
   **Licenses**: The dataset does not have a license.
   **Ethical considerations**: No personally identifiable information or offensive content is present in the dataset.

42. **Expression Atlas** (Papatheodorou et al., 2018) consists of RNA gene expression data across species and biological conditions.
   **Split**: We use a training set of 3605 records, a test set of 901 records, and a total size of 4,506 records.
   **Access restrictions**: The dataset is available to download from https://www.ebi.ac.uk/gxa/download.
   **Licenses**: This dataset is available in the Creative Commons Attribution-Noncommercial-NoDerivatives 4.0 International License https://creativecommons.org/licenses/by-nc-nd/4.0/.
   **Ethical considerations**: No personally identifiable information or offensive content is present in the dataset.

43. **Geo** is a functional genomics dataset supporting MIAME-compliant data submissions.
   **Split**: We use a training set of 101162 records, a test set of 25290 records, and a total size of 126,452 records.
   **Access restrictions**: The dataset is available to download from https://www.ncbi.nlm.nih.gov/geo/.
   **Licenses**: The license is available at Link.
   **Ethical considerations**: No personally identifiable information or offensive content is present in the dataset.

44. **Vital** (Cui et al., 2024) is a medical image-language dataset based on PMC-15, where instructional data is generated using the gpt-4-vision-preview API.
   **Split**: We use a training set of 42000 records, a test set of 168000 records, and a total size of 210,000 records.
   **Access restrictions**: The dataset is available to download from https://huggingface.co/datasets/mao1207/BioMed-VITAL-instructions.
   **Licenses**: The dataset is available in Apache License 2.0 https://www.apache.org/licenses/LICENSE-2.0.
   **Ethical considerations**: No personally identifiable information or offensive content is present in the dataset.

45. **MIMIC-IV** (Johnson et al., 2023) (Johnson et al., 2016) (Goldberger et al., 2000) is a multimodal medical dataset on patients admitted to the emergency department or intesnive care unit at the Beth Israel Deaconess Medical Center in Boston, MA. We train and evaluate our models using the EHR, vital sign, and chest-xray modalities.

**Split**: CLIMB provide a training set of 640,000 records, a test set of 160,000 records, and a total size of 800,000 records. In the fusion experiment, however, we focus on the multimodal subset of MIMIC-IV, which contains 166,215 training samples, 41,554 test samples, and 207769 total samples.

**Access restrictions**: The dataset is available to download from https://physionet.org/content/mimiciv/3.1/ with credentialed account and CITI training.

**Licenses**: The dataset is available in PhysioNet Credentialed Health Data License 1.5.0 https://physionet.org/content/mimiciv/view-license/3.1/.

**Ethical considerations**: No personally identifiable information or offensive content is present in the dataset.

## Dataset Selection Methodology

Our dataset collection and curation process follows a systematic two-stage approach to ensure both comprehensive coverage and accessibility while maintaining data quality and diversity.

### Stage 1: Initial Dataset Identification

We first conducted a comprehensive literature review of publicly available clinical datasets across different modalities. Our inclusion criteria at this stage focused on accessibility:

- Datasets with direct download access through public repositories

- Datasets requiring application but with clear, less restrictive licensing terms

- Datasets with well-documented data collection protocols and annotation procedures

We explicitly excluded datasets that:

- Require institutional review board (IRB) approval

- Have restrictive licensing terms limiting research use

### Stage 2: Selection and Prioritization

In the second stage, we applied a two-tier selection process:

- **Tier 1:** We first included widely-cited benchmark datasets that serve as standard evaluation metrics in their respective domains. These datasets were identified based on citation count and frequency of use in published literature. Examples include CheXpert and MIMIC-IV.

- **Tier 2:** We then systematically identified and included datasets that addressed our three key criteria:

  – Novel tasks: Datasets covering emerging clinical challenges (e.g., COVID-19 diagnosis)
  – Understudied modalities: Datasets from underrepresented data types (e.g., EEG, endoscopic videos)
  – Underrepresented regions: Datasets from developing regions with limited representation

### Selection Criteria

Here, we elaborate on our methodology for identifying understudied modalities and underrepresented regions during the stage 2 selection.

**Understudied Modalities.** We evaluated modalities from two complementary perspectives:

1. **Research Attention:** We quantified research activity by aggregating Google Scholar search results using standardized queries (e.g., '[Modality] classification', '[Modality] machine learning'). Our analysis revealed significant disparities in research attention across modalities:

   - High attention (>1M articles): Pathology (1.38M), X-ray (1.08M), CT Scan (2.83M), Endoscopic (1.74M)
   - Medium attention (300K-1M): MRI (931K), Dermoscopy (530K), Ultrasound (636K), Fundus (848K), EEG (450K)
   - Low attention (<500K): Mammography (56.1K), ECG (265K)

2. **Data Availability:** We analyzed the total number of publicly available samples per modality:

   - High availability (>500K): X-ray (595,264), CT Scan (1,810,256), EEG (655,786)

- Medium availability (50K-500K): ECG (84,172), Genomic (130,958)
- Low availability (5K-50K): Mammography (24,697), Dermoscopy (45,439), Fundus (18,067), Endoscopic (14,400), Pathology (24,014), MRI (14,807)
- Very low availability (<5k): Ultrasound (1,633)

Based on this analysis, we identified several critically understudied modalities. Mammography and ECG emerged as understudied based on research attention, while ultrasound was identified as understudied due to extremely limited public data availability (1,633 samples). These findings guided our focused efforts to collect additional datasets in these modalities.

**Underrepresented Regions.** We also employed a two-step approach to identify geographic gaps in dataset coverage:

1. **Geographic Distribution:** We created a global heatmap of dataset origins, revealing significant underrepresentation in:

    - Africa
    - South America
    - Parts of South and Southeast Asia

2. **Economic Development:** We mapped datasets to their countries of origin, specifically identifying datasets from developing nations. This analysis highlighted the importance of including datasets from:

    - Asia: India, Vietnam
    - Middle East: Iraq
    - South America: Brazil

This analysis informed our targeted efforts to include datasets from these underrepresented regions, aiming to improve the geographic and demographic diversity of our benchmark. The complete list of included datasets and their geographic distribution is provided in Table X.

**Data Preprocessing and Standardization**
To preserve data fidelity while ensuring usability, we implemented minimal preprocessing steps:

- **Time Series Data (ECG/EEG):**

    - Standardized sampling rates across datasets
    - Normalized amplitude ranges
    - Preserved original waveform characteristics

- **Imaging Data:**

    - Maintained original image resolution and quality
    - Created standardized JSON metadata files linking:
        * Clinical labels
        * Demographic information
        * Multi-view relationships
        * Additional annotations (where available)

No preprocessing was done on graph data, as they are already processed and ready to use.

All additional metadata and multi-view images are preserved and made available, though our benchmark experiments utilize only the primary labels to ensure fair comparison across models. The complete preprocessing scripts and documentation are available in our code repository.

## B. CLIMB-QA Construction

To enable standardized evaluation of large vision-language models (VLMs), we construct CLIMB-QA, a question-answering version of our dataset. For each sample $x_i \in D_k$ with label set $y_i \subseteq V_k$, we generate a question-answer pair $(q_i, a_i)$ where:

- $q_i$ is constructed as a natural language question incorporating:

- Task description (e.g., "grade the diabetic retinopathy")
- Input modality context (e.g., "in the retinal image")
- Available choices from the label vocabulary $V_k$

- $a_i$ is the ground truth answer derived from $y_i$, formatted as either:

  - Single-label: $a_i \in V_k$ for mutually exclusive classes
  - Multi-label: $a_i \subseteq V_k$ for compatible conditions

Formally, we define a mapping function $\psi : (x_i, y_i, V_k) \to (q_i, a_i)$ that generates question-answer pairs while preserving the original classification task structure. For example, in the APTOS dataset for diabetic retinopathy grading:

$q_i =$ "Above is a retinal image of a patient. Grade the diabetic retinopathy on the Davis Scale, choosing from: No DR, Mild DR, Moderate DR, Severe DR, Proliferative DR."

$a_i =$ "Moderate DR"

For multi-label classification tasks, we evaluate predictions using order-agnostic matching: given a predicted answer set $\hat{a}_i$ and ground truth $a_i$, we consider the prediction correct if $\hat{a}_i = a_i$ regardless of the order in which the labels are listed. This ensures fair evaluation when multiple conditions are present and can be enumerated in any order.

## C. Taxonomy Standardization

The standardization of label taxonomies across diverse medical imaging modalities represents a significant methodological challenge. Our standardization efforts required careful balancing of two competing objectives:

1. Merging semantically similar terms to facilitate effective learning and enable cross-modality knowledge transfer

2. Minimizing information loss and avoiding the introduction of inaccuracies when modifying existing labels

Our standardization efforts concentrated particularly on ECG and chest X-ray modalities, as these domains offer fine-grained labels with highly variable terminologies across different datasets. For ECG standardization, we adopted the methodology established in BenchMD (Wantlin et al., 2023).

For chest X-ray standardization, we developed a novel mapping framework with input from radiologists, building upon established practices from recent literature (Nasser & Akhloufi, 2023; Tayebi Arasteh et al., 2023; Jang et al., 2024). We consolidated all general chest X-ray labels into the CheXpert 14 categories, as shown in Table 11:

We acknowledge that this standardization process necessarily affected label granularity, particularly for the lung opacity and lung lesion classes where multiple distinct pathologies were consolidated. To accommodate diverse research needs, CLIMB includes both standardized and raw labels, enabling researchers to prioritize either cross-dataset consistency or fine-grained granularity based on their specific requirements. All experiments reported in our main manuscript utilized the standardized labels.

### C.1. Empirical Evaluation of Standardization Impact

To quantitatively assess the effects of label standardization, we conducted a comparative evaluation using identical vision encoders trained on raw versus standardized labels. The results, presented in Table 12, demonstrate the impact across multiple imaging modalities:

The standardized labels yielded superior overall performance with a 3.6% improvement in AUC, 10.3% improvement in sensitivity, and 4.0% improvement in specificity. Interestingly, the benefits of standardization were more pronounced in modalities other than chest X-ray, despite chest X-ray being the modality where most relabeling occurred. This suggests that standardized labels enable the model to more effectively establish conceptual connections across different imaging modalities, compensating for the inevitable loss of some fine-grained distinctions. We believe these findings underscore the importance of continued efforts toward terminology standardization and systematic relabeling of public clinical imaging datasets to facilitate more effective multi-modal learning approaches.

# D. Detailed Experimental Setup

## D.1. Problem Definition

**RQ1/RQ2:** We use the definition of the dataset as well as the vocabularies as defined in Sec. 3.2. For each input sample $x_i \in \mathcal{X}$, which can be an image, video, or multi-channel time series, we extract $n$ sequential elements. The input processing varies by modality:

1. Images: $\phi_{img} : \mathbb{R}^{H \times W \times C} \to \mathbb{R}^{n \times \sigma \times \sigma \times C}$

2. Videos: $\phi_{vid} : \mathbb{R}^{T \times H \times W \times C} \to \mathbb{R}^{n \times \sigma \times \sigma \times C}$

3. Time Series: $\phi_{ts} : \mathbb{R}^{T \times C} \to \mathbb{R}^{n \times C}$

where $\sigma$ denotes the model-specific input size, and $H, W, C, T$ represent spatial dimensions, channels, and temporal length, respectively. The objective is to learn a function $f : \mathcal{X} \to \{0, 1\}^{|\mathcal{V}|}$ that maps each input to a binary vector over $\mathcal{V}$.

**RQ3:** Given multimodal inputs $\mathcal{M} = m_{vis}, m_{lang}, m_{ts}$, where each modality-specific input $x_i^m \in \mathcal{X}^m$ corresponds to visual, language, and time series data respectively, the model aims to learn two prediction functions:

1. In-hospital Mortality in 48 hours (IHM 48): Binary prediction $f_{ihm} : \prod_{m \in \mathcal{M}} \mathcal{X}^m \to \{0, 1\}$ predicting 48-hour mortality

2. Length of Stay: Regression function $f_{los} : \prod_{m \in \mathcal{M}} \mathcal{X}^m \to \mathbb{R}^+$ estimating the expected duration of hospitalization.

## D.2. Experimental Procedures

To answer the above research questions, we design our experiments as follows:

### D.2.1. RQ1: UNIVERSAL ENCODERS PERFORMANCE UNDER MULTITASK LEARNING.

We investigate how can we build a universal encoder for each input type across all clinical tasks. Specifically, we train vision, graph and time series encoders on the complete $\mathcal{D}$ to assess general diagnostic capabilities. For each sample $x_i \in D_k$, we evaluate performance using a dataset-specific vocabulary mask $\mathbb{1}_{V_k} \in \{0, 1\}^{|\mathcal{V}|}$, where only predictions corresponding to labels in $V_k$ are considered in the evaluation metrics. Performance is measured using Area Under the ROC Curve (AUC) and binary classification metrics (specificity and sensitivity) with a decision threshold of 0.5 over the masked label space $\mathcal{V} \odot \mathbb{1}_{V_k}$. In addition, we compare the results pre-trained on our entire datasets against those just pre-trained on each target dataset, exploring the effect of large-scale pre-training on the model's performance.

### D.2.2. RQ2: TRANSFER LEARNING UNDER RESOURCE CONSTRAINTS.

Second, we evaluate few-shot generalization on out-of-distribution (OOD) datasets $\mathcal{D}_{ood} \not\subset \mathcal{D}_{train}$, where $\mathcal{D}_{ood}$ contains novel label sets $V_{ood}$ such that $V_{ood} \cap \mathcal{V} = \emptyset$ within the same modality. This setup simulates the practical scenario where models must adapt to novel diagnostic tasks with limited labeled examples while leveraging pre-trained representations from related but distinct tasks. A detailed description of the dataset composition, as well as the experimental procedure, are included in App. D.7.1.

### D.2.3. RQ3: SINGLE MODALITY TO MULTIMODALITY TRANSFER VIA ROBUST FUSION STRATEGY.

Finally, we investigate optimal fusion mechanisms for integrating heterogeneous modalities as defined in App. D.1. We evaluate three fusion architectures $g_\theta : \prod_{m \in \mathcal{M}} \mathbb{R}^{d_m} \to \mathbb{R}^d$:

1. late fusion $g_{late}$ that averages predictions from modality-specific classifiers: $g_{late}(h_m) = \frac{1}{|\mathcal{M}|} \sum_{m \in \mathcal{M}} \text{MLP}_m(h_m)$,

2. simple concatenation followed by two-layer MLP $g_{mlp}(h_m) = \text{MLP}(\text{Concat}([h_{vis}, h_{lang}, h_{ts}]))$, and

3. cross-modal attention $g_{attn}$ that uses text features to form queries and concatenated vision/time-series features for keys and values, computing $g_{attn}(h_m) = \text{FFN}(\text{softmax}(\frac{QK^T}{\sqrt{d}})V)$ where $Q = h_{lang}W_Q$, $K = \text{Concat}([h_{vis}, h_{ts}])W_K$, $V = \text{Concat}([h_{vis}, h_{ts}])W_V$

For controlled comparison, we fix the backbone encoders across all fusion strategies: ConvNextv2 (Liu et al., 2022) for visual inputs, ClinicalBERT (Liu et al., 2025) for text, and ECG-JEPA (Kim, 2024) for time series data.

## D.3. Evaluation Metrics

For binary classification tasks $f : \mathcal{X} \to \{0, 1\}$, we employ three complementary metrics to assess model performance:

1. Area Under the ROC Curve (AUC): Given predicted probabilities $\hat{y}_i \in [0, 1]$ and true labels $y_i \in \{0, 1\}$, AUC measures the model's ability to discriminate between classes across all possible decision thresholds:

$$\text{AUC} = \int_0^1 \text{TPR}(t)\text{FPR}'(t)dt$$

   where TPR(t) and FPR(t) are the true positive and false positive rates at threshold t.

2. Sensitivity (also known as recall): Measures the model's ability to correctly identify positive cases:

$$\text{Sensitivity} = \frac{\text{TP}}{\text{TP} + \text{FN}}$$

   where TP and FN denote true positives and false negatives respectively.

3. Specificity: Quantifies the model's ability to correctly identify negative cases:

$$\text{Specificity} = \frac{\text{TN}}{\text{TN} + \text{FP}}$$

   where TN and FP denote true negatives and false positives respectively.

The choice of these three metrics is particularly motivated by clinical considerations. In medical diagnosis, there is often an inherent trade-off between sensitivity and specificity, where improving one typically comes at the cost of the other. Sensitivity is crucial in cases where missing a positive diagnosis (false negative) could have severe consequences for patient outcomes, such as failing to detect a life-threatening condition. Conversely, specificity is vital when false positives could lead to unnecessary interventions, psychological distress, or resource waste.

AUC provides a threshold-independent measure of discriminative ability, making it particularly valuable for comparing models across different operating points and clinical contexts. This is especially relevant in our multi-task setting where different clinical applications may require different sensitivity-specificity trade-offs.

For regression tasks such as length of stay prediction ($f_{los}$), we use Mean Absolute Error (MAE):

$$\text{MAE} = \frac{1}{n}\sum_{i=1}^{n}|y_i - \hat{y}_i|$$

where $y_i$ and $\hat{y}_i$ represent the true and predicted values respectively. All metrics are computed using the dataset-specific vocabulary masks $\mathbb{1}_{V_k}$ as defined in Sec. D.1.

Below, we describe in detail about the setup and procedures of vision, time series and graph experiments.

### D.4. Vision Model Experiments
D.4.1. VISION MODEL DETAILS

**MedViT** (Manzari et al., 2023) is a Vision Transformer variant specifically designed for medical imaging tasks. It incorporates a hierarchical structure with varying token sizes across different stages and employs medical-specific attention mechanisms optimized for capturing fine-grained anatomical details.

**PMC-CLIP** (Lin et al., 2023) adapts the CLIP architecture for medical imaging by pretraining on PubMed Central articles and their associated figures. It maintains the original dual-encoder structure but incorporates medical domain knowledge through specialized text-image contrastive learning.

**RAD-DINO** (Pérez-García et al., 2025) extends the DINO self-supervised learning framework to radiology images. It employs specialized augmentation strategies and anatomical consistency constraints during the self-supervised pretraining process to better capture medical imaging characteristics.

**SBB2** (Radford et al., 2021) is an enhanced vision backbone that builds upon the clip architecture. It introduces improved spatial mixing operations and hierarchical feature representations while maintaining computational efficiency.

**Swin Transformer** (Liu et al., 2021) is a hierarchical vision transformer that computes self-attention within shifted windows. It introduces a hierarchical architecture with varying window sizes across different stages, enabling efficient modeling of both local and global dependencies.

**EVA-2** (Fang et al., 2024) is a large-scale vision foundation model that extends the original EVA architecture. It utilizes masked image modeling and contrastive learning objectives, incorporating improvements in model scaling and training strategies.

**InternViT** (Chen et al., 2024b) is a vision transformer model that introduces internalized attention mechanisms. It optimizes the traditional transformer architecture for improved efficiency while maintaining performance across diverse visual tasks.

**ConvNeXTv2** (Liu et al., 2022) is a pure convolutional architecture that modernizes traditional CNN design principles. It incorporates fully convolutional design, global response normalization, and gradient checkpointing, achieving strong performance across various vision tasks.

### D.4.2. VISION HYPERPARAMETERS AND EXPERIMENTAL PROCEDURES

All experiments are ran on a GPU server with 8xH200 141GB GPUs. We used the SOAP optimizer (Vyas et al., 2024) as it offers the best performance. Depending on the model sizes, we use a parameter search to identify the optimal learning rate from $1 \times 10^{-5}$ to $1 \times 10^{-3}$ for all experiments. The weight decay was set to $1 \times 10^{-3}$. We use a parameter search to find the largest GPU that could fit on a single server. All experiments were conducted using the PyTorch framework. We saved the model with the lowest $CrossEntropy$ loss over 5 epochs for evaluation on the test split. We report the AUROC, Sensitivity, Specificity, F1 Score and accuracy of our experiments in App. E.

### D.5. EEG Model Experiments

We investigate the performance of five baseline models and four variants of foundational time series models for EEG classifications. Our EEG experiment is built upon the repository sheared by (Yang et al., 2024), and the pre-trained weights were downloaded from `https://github.com/ycq091044/BIOT`.

### D.5.1. EEG DATASETS AND PREPROCESSING

We evaluated the models on IIIC (Jing et al., 2023b), TUAB (Lopez et al., 2015), and TUEV (Lopez et al., 2016). All EEG channels, $S[i]$, in each individual sample were resampled to 200 Hz and normalized using the 95th percentile of the absolute amplitude:

$$\frac{\mathbf{S}^{[i]}}{\text{percentile}([|\mathbf{S}[i, 1]|, |\mathbf{S}[i, 2]|, \ldots, |\mathbf{S}[i, J]|], 95\%)}$$

For details on dataset access and data splitting, please refer to App. A.

### D.5.2. EEG HYPERPARAMETERS AND EXPERIMENTAL PROCEDURES

We used the Adam optimizer (Diederik, 2014) with a learning rate of $1 \times 10^{-3}$ for all experiments. The weight decay was set to $1 \times 10^{-5}$. For most experiments, the batch size was set to 512, while for few-shot experiments, it was set to 4. All experiments were conducted using the PyTorch framework. We saved the model with the lowest $CrossEntropy$ loss over 20 epochs for evaluation on the test split. We report the AUROC, Sensitivity, Specificity, and F1 Score of our experiments in App. E.

### D.5.3. EEG MODEL DETAILS

**SPaRCNet** (Jing et al., 2023b) is a 1D-CNN designed for EEG classification. It employs a hierarchical feature extraction process, beginning with an initial convolutional layer followed by multiple densely connected blocks and transition layers. The model integrates ELU activation functions, batch normalization, and dropout for improved generalization.

**CNNTransformer** (Peh et al., 2022) is a hybrid deep learning model that combines convolutional neural networks (CNNs) with a Transformer encoder for EEG classification. The model first applies short-time Fourier transform (STFT) to extract spectral representations, which are then processed through a deep residual CNN with four stacked *ResBlocks* for hierarchical feature extraction. The CNN embeddings are segmented and passed through a Transformer encoder with positional encoding, allowing the model to capture long-range temporal dependencies.

**ContraWR** (Yang et al., 2023) is a EEG classification model that integrates STFT with a 2D-CNN for sleep staging. The model first converts raw EEG signals into spectrograms using STFT, which are then processed through a deep residual CNN

with four stacked *ResBlocks*, each employing batch normalization, dropout, and max pooling for feature extraction.

**FFCL** (Li et al., 2022) is a hybrid CNN-LSTM model designed for EEG classification, integrating both spectral and temporal feature representations. The model first apply STFT to extract frequency-domain features, which are then processed through a deep residual CNN with four stacked *ResBlocks*. In parallel, the raw EEG signals undergo temporal compression using a downsampling operation before being fed into a bidirectional LSTM for sequential feature extraction. The final representation is obtained by concatenating CNN and LSTM embeddings, which are passed through a fully connected classification layer.

**STTransformer** (Song et al., 2021) is a spatiotemporal Transformer model for EEG classification that integrates channel-wise attention with Transformer-based sequence modeling. The model first applies a *ChannelAttention* mechanism to capture inter-channel dependencies, followed by a *PatchSTEmbedding* module that encodes local temporal structures. These embeddings are then processed through a deep Transformer encoder, which models long-range dependencies and contextual information. The v2.0 of STTransformer is known as EEG-Conformer (Song et al., 2022).

**BIOT** (Yang et al., 2024) is a biosignal transformer model designed for cross-data learning in the wild, enabling robust representation learning across diverse biosignal modalities such as EEG and ECG. The model utilizes a frequency-based tokenization approach, where biosignals are first transformed into spectrogram representations via STFT. Channel-specific positional embedding and temporal positional embedding are added to the tokens to enhance both temporal and spatial representations. These spectral embeddings are then processed using a Linear Attention Transformer.

## D.6. ECG Model Experiments

We compare the performance of ECG-JEPA, a time series specific model, and UniTS, a generalized time series model for ECG classifciations. Our ECG experiment and pretrained encoder weights were adopted from the following repositories: `https://github.com/sehunfromdaegu/ecg_jepa` and `https://github.com/mims-harvard/UniTS`.

### D.6.1. ECG DATASETS AND PREPROCESSING

We evaluated the models on PTB-XL (Wagner et al., 2020), CPSC (Liu et al., 2018), Chapman-Shaoxing (Zheng et al., 2020), and Ga (Alday et al., 2020). All ECG signals were resampled to 500 Hz and standardized to a length of 2500 timesteps. Samples where the first 15 timesteps contained only zeros across all channels were removed. For ECG-JEPA, the number of ECG channels was reduced from 12 to 8, as the remaining 4 channels can be derived using linear combinations of the selected leads.

For details on dataset access and data splitting, please refer to App. A.

### D.6.2. HYPERPARAMETERS AND EXPERIMENTAL PROCEDURES

For the ECG-JEPA model, we use the Adam optimizer (Diederik, 2014) with a learning rate of $1 \times 10^{-3}$ and a weight decay of $1 \times 10^{-2}$. For the UniTS model, we use the Adam optimizer with a learning rate of $1 \times 10^{-4}$ and a weight decay of $5 \times 10^{-6}$. For all experiments, the batch size was set to 32. All experiments were conducted using the PyTorch framework. We report the AUCROC, Sensitivity, and Specificity Score of our experiments in App. E.

### D.6.3. ECG MODEL DETAILS

**ECG-JEPA** (Kim, 2024) is a self-supervised ECG representation learning model that predicts in the latent space rather than reconstructing raw signals. It introduces Cross-Pattern Attention (CroPA), a masked attention mechanism that prioritizes critical ECG features across multiple leads, enhancing performance on downstream tasks.

**UniTS** (Gao et al., 2024) is a unified multitask time series model that integrates predictive and generative tasks using task tokenization within a single framework. It employs a modified transformer block to learn transferable time series representations across diverse domains, handling variations in sampling rates and temporal patterns.

## D.7. Out-of-distribution Experiment Details

In this section, we describe the details on how we ran the OOD transfer experiment, which addresses RQ2. We first define the list of datasets we selected for the transfer experiments, and then explain the detailed procedure of the experiment.

### D.7.1. DEFINITION OF OUT-OF-DISTRIBUTION DATASETS

The list of datasets we selected for out-of-distribution (OOD) transfer experiments is described in Table 13. In general, we select the datasets such that they reflect a different task within a modality it was trained on. The OOD may be a novel task

(like COVID-19), a new task (cancer vs pulmonary embolism), or a different granularity of the same task (6-way BI-RADS classification instead of 5-way).

### D.7.2. EXPERIMENTAL PROCEDURES

**Vision Encoders.** For vision encoders, we run a mixed training on the full dataset, with the OOD dataset filtered out.

## E. Full Experimental Results

### E.1. Full Dataset Multitask Training Results

In this section, we report the detailed model's performance for each dataset in Table 3.

## F. Validation of Out-of-Distribution Dataset Selection

We provide additional empirical validation of the distinctiveness of out-of-distribution (OOD) datasets. Our OOD dataset selection was primarily guided by fundamental task differences rather than superficial variations. As detailed in Table 13, we assembled 10 OOD datasets spanning diverse clinical domains and imaging modalities. Notably, 7 out of these 10 datasets (COVID-19, CoronaHack, ISIC-2020, BCSS, BUSI, LNDb, and PTB-XL-Finegrained) share zero label overlap with any other datasets within their respective clinical domains, while the remaining datasets exhibit distinct task granularities that fundamentally alter the nature of the prediction problem.

To quantitatively validate the heterogeneity of our designated OOD datasets, we conducted a comprehensive membership inference experiment. Using ConvNeXT-v2-base as the backbone architecture, we trained a classifier to predict dataset membership—essentially testing whether the model could distinguish which dataset a given sample originated from. High performance on this task would indicate that the datasets possess distinct characteristics that make them easily distinguishable, thus supporting our designation of these datasets as out-of-distribution.

The results, presented in Table 32, demonstrate that our selected OOD datasets exhibit substantial distinctiveness:

The consistently high balanced accuracy and AUC scores across most datasets confirm that these datasets possess unique characteristics that make them readily distinguishable from one another. This empirical evidence strongly supports our methodological choice to designate these datasets as out-of-distribution, as they represent genuinely distinct data distributions rather than minor variations of the same underlying distribution.

## G. Exploration on Self-Supervised Learning

Self-supervised pretraining represents a fundamental component of modern representation learning. We conducted comprehensive experiments comparing unsupervised pretraining with supervised multitask learning across different modalities to understand their relative contributions to model performance.

### G.1. Time Series Modalities: ECG and EEG

Our experiments revealed that time series modalities benefited substantially from unsupervised pretraining. In the ECG domain, we implemented masked autoencoder (MAE) pretraining on the CLIMB dataset and compared it with pretraining exclusively on the target dataset. As shown in Table 33, pretraining on the diverse CLIMB dataset yielded consistent improvements:

Similarly, for EEG datasets, we compared three training strategies: no pretraining, multitask learning (MTL) only, and combined pretraining with multitask learning (PT+MTL). The results in Table 34 show that pretraining on diverse data improved performance for two out of three datasets:

### G.2. Vision Modalities

While our main manuscript (Figure 4) demonstrated that multitask learning effectively improved vision encoder performance, we found surprisingly different results for self-supervised pretraining in vision modalities. Table 35 presents our comprehensive comparison of different pretraining strategies:

We hypothesize that this phenomenon occurs because these vision models have already undergone extensive pretraining on massive unlabeled natural image corpora. Consequently, an additional phase of masked image modeling or contrastive learning on clinical data may not substantially shift or enrich their learned representations. This finding suggests that for vision models with strong natural image pretraining, supervised multitask learning on diverse labeled clinical data provides

more effective domain adaptation than additional self-supervised pretraining phases. We encourage the community to explore alternative approaches for better leveraging the diverse labeled data available in CLIMB to further improve vision model performance in clinical domains.

Table 9. **Dataset Classes for Multilabel Classification with CLIMB.**

| Dataset | # Classes | Classes |
|---|---|---|
| PTB-XL | 7 | Normal, Conduction Delay (CD), Hypertrophy (HYP), Myocardial Infarction (MI), Sinus Tachycardia/Bradycardia/Conduction (STTC), Atrial Fibrillation/Atrial Flutter (A. Fib/Aflutter), Other |
| Chapman-Shaoxing | 7 | Same as PTB-XL |
| Georgia | 7 | Same as PTB-XL |
| CPSC | 7 | Same as PTB-XL |
| IIIC | 6 | Seizure (SZ), Lateralized Periodic Discharges (LPD), Generalized Periodic Discharges (GPD), Lateralized Rhythmic Delta Activity (LRDA), Generalized Rhythmic Delta Activity (GRDA), Other |
| TUAB | 2 | Normal, Abnormal |
| TUEV | 6 | Spike and Slow Wave (SPSW), Generalized Periodic Epileptiform Discharge (GPED), Periodic Lateralized Epileptiform Discharge (PLED), Eye Movement (EYEM), Artifact (ARTF), Background (BCKG) |
| CheXpert | 14 | Atelectasis, Cardiomegaly, Consolidation, Edema, Enlarged Cardiomediastinum, Fracture, Lung Lesion, Lung Opacity, Pleural Effusion, Pneumonia, Pneumothorax, Pleural Other, Support Devices, No Finding |
| MIMIC-CXR | 14 | Same as CheXpert |
| VinDr-CXR | 6 | Lung tumor, Pneumonia, Tuberculosis, COPD, Other diseases, No finding |
| COVID-19 | 4 | Normal, Bacterial Pneumonia, COVID-19, Viral Pneumonia |
| CoronaHack | 3 | Normal, Bacterial Pneumonia, Viral Pneumonia |
| VinDr-Mammo | 5 | BI-RAD 1-5 |
| CBIS-DDSM | 6 | BI-RAD 0-5 |
| CMMD | 2 | Benign, Malignant |
| ISIC-2020 | 2 | Malignant, Benign |
| HAM10000 | 5 | Melanoma (MEL), Nevus (NV), Basal Cell Carcinoma (BCC), Actinic Keratosis/Intraepithelial Carcinoma (AKIEC), Other (OTHER) |
| PAD-UFES-20 | 5 | Melanoma (MEL), Nevus (NV), Basal Cell Carcinoma (BCC), Actinic Keratosis/Intraepithelial Carcinoma (AKIEC), Other (OTHER) |
| Messidor-2 | 5 | None, Mild DR, Moderate DR, Severe DR, PDR |
| APTOS 2019 | 5 | No DR, Mild, Moderate, Severe, Proliferative DR |
| Jichi | 3 | SDR (simple diabetic retinopathy), PPDR (pre-proliferative diabetic retinopathy), PDR (proliferative diabetic retinopathy) |
| LNDb | 3 | nodule $\geq$ 3mm, nodule ¡3mm, non-nodule |
| INSPECT | 5 | No PE, Acute Subsegmental-only PE, Acute PE, Subsegmental-only PE, Chronic PE |
| KiTS23 | 2 | Benign, Malignant |
| Hemorrhage | 2 | No Hemorrhage, Has Hemorrhage |
| RSPECT | 3 | No PE, Chronic PE, Acute PE |
| EchoNet-Dynamic | - | Not classification |
| BUSI | 3 | Normal, Malignant, Benign |
| COVID-BLUES | 2 | Has COVID, No COVID |
| COVID-US | 3 | Covid, Pneumonia, Normal |
| Brain Tumor | 4 | No Tumor, Pituitary Tumor, Glioma Tumor, Meningioma Tumor |
| Brain MRI | 2 | Yes, No (presence of tumors) |
| ABCD | 2 | Normal, Abnormal |
| ABIDE | 2 | ASD, Typical controls |
| PPMI | 2 | Control, PD patients |
| PROTEINS | 2 | Enzyme, Not enzyme |
| PPI | 2 | |
| LC25000 | 5 | Colon adenocarcinomas, Benign colon, Lung adenocarcinomas, Lung squamous cell carcinomas, Benign lung |
| BCSS | 4 | Tumor, Stroma, Lymphocytic infiltrate, Necrosis/debris |
| Cholec80 | | Surgery phase annotations and surgery tool labels |
| HuGaDB | 4 | Sitting, Standing, Sitting down, Standing up |
| Expression Atlas | - | Not classification |
| Geo | - | Not classification |
| Vital | - | Not classification |
| MIMIC-IV | 2 | 48 Hour In-Hospital-Mortality (48 IHM) (Yes/No) |

*Table 10.* **Dataset Demographics and Location Information.**

| Dataset | Locations | Demographic Information |
|---|---|---|
| PTB-XL | Multiple | sex: 52% male, 48% female; age range: 0-95 (median: 62, IQR: 22); height; weight |
| Chapman-Shaoxing | Shaoxing, Zhejiang, China | sex: male: 22,599 (56%); female: 17,659 (44%); age groups: 51–60 (19.8%), 61–70 (24%), and 71–80 (17.3%) |
| Georgia | Emory University, Atlanta, Georgia, USA | age; sex |
| CPSC | China | unknown |
| MIMIC-CXR | Beth Israel Deaconess Medical Center in Boston, MA | unknown |
| CheXpert | Stanford, California, US | sex; age |
| VinDr-CXR | The Hospital 108 and the Hanoi Medical University Hospital in Vietnam | Training set: median age: 43.77; sex: 52.21% male, 47.79% female; Test set: median age: 31.80; sex: 55.90% male, 44.10% female |
| VinDr-Mammo | The Institutional Review Board of Hanoi Medical University Hospital (HMUH) and Hospital 108 (H108) | age; imaging device's model |
| CBIS-DDSM | Stanford, California, US | Unknown |
| ISIC-2020 | Hospital Clínic de Barcelona, Medical University of Vienna, Memorial Sloan Kettering Cancer Center, Melanoma Institute Australia, University of Queensland, and the University of Athens Medical School | sex: female: 15981 (48%), male: 17080 (52%); age range: 0-90 (median: 48.87) |
| HAM10000 | unknown | unknown |
| PAD-UFES-20 | Federal University of Espírito Santo (UFES), Espírito Santo, Brazil | country of parents; age; gender; access to piped water; access to sewage system; region |
| Messidor-2 | Brest University Hospital | unknown |
| APTOS 2019 | Aravind Eye Hospital, India | unknown |
| Jinchi | Jichi Medical University | unknown |
| LNDb | Centro Hospitalar e Universitário de São João (CHUSJ) in Porto, Portugal | unknown |
| ABIDE | California Institute of Technology, Carnegie Mellon University, Kennedy Krieger Institute, and more | age; sex; handedness; full-scale IQ |
| ABCD | unknown | gender identity; environmental factors |
| PPMI | Multiple | sex: 54.5% male, 45.5% female; race; age |
| HuGaDB | unknown | sex: 4 females, 14 males; age: average: 23.67; height: average: 179.06 cm; weight: average: 73.44 kg |
| INSPECT | Stanford Medicine (2000-2021) | gender: female: 10,733, male: 8,666, unknown: 3; age: 18-39: 2,912, 39-69: 9,974, 69-89: 5,859, ¿89: 657; race: white: 10,704, asian: 2,976, black: 1,103, native: 415, unknown: 2,404; ethnicity: hispanic: 3,018, not hispanic: 15,628, unknown: 756 |
| EchoNet-Dynamic | Stanford University | unknown |
| BUSI | unknown | unknown |
| COVID-19 | University of Montreal, Canada | unknown |
| Brain Tumor | Eindhoven University of Technology, Netherlands | unknown |
| KiTS23 | Minnesota, US | unknown |
| Hemorrhage | Al Hilla Teaching Hospital, Iraq | age: mean: 27.8; gender: male: 46, female: 36 |
| CMMD | China | age |
| CoronaHack | Mila, University of Montreal | unknown |
| COVID-BLUES | Maastricht University Medical Center (UMC+) in the Netherlands | weight; sex; height; bmi; age |
| COVID-US | unknown | gender; age; alcoholic; drug use |
| IIIC | Massachusetts General Hospital, Harvard Medical School, Boston, USA | 1950 patients; labeled by 124 raters; 20 of the raters are physician experts |
| TUAB | The Temple University Hospital, Philadelphia, Pennsylvania, USA | Evaluation Dataset: Total: 276 files, 253 subjects; Abnormal Female: 63 files, 51 subjects; Abnormal Male: 63 files, 54 subjects; Normal Female: 85 files, 84 subjects; Normal Male: 65 files, 64 subjects; Train Dataset: Total: 2,717 files, 2,130 subjects |
| TUEV | The Temple University Hospital, Philadelphia, Pennsylvania, USA | 290 patients in train; 78 patients in eval |
| MIMIC-IV | Beth Israel Deaconess Medical Center, Massachusetts Institute of Technology | age; gender; insurance type (medicaid, medicare, other) |

*Table 11.* Chest X-ray label standardization mapping

| Raw Label | Standardized Label |
|---|---|
| Aortic enlargement, Enlarged PA | Enlarged Cardiomediastinum |
| Cardiomegaly | Cardiomegaly |
| Atelectasis | Atelectasis |
| Consolidation | Consolidation |
| Edema | Edema |
| Infiltration, Lung Opacity, ILD, Pulmonary fibrosis | Lung Opacity |
| Nodule/Mass, Other lesion, Lung cavity, Lung cyst, Lung tumor | Lung Lesion |
| Pleural effusion | Pleural Effusion |
| Pleural thickening | Pleural Other |
| Pneumothorax | Pneumothorax |
| Rib fracture, Clavicle fracture | Fracture |
| No finding | No Finding |
| Support Devices | Support Devices |
| Pneumonia | Pneumonia |

*Table 12.* Performance comparison between raw and standardized label training

| Model | CXR | | | Mammo | | | Derm | | | CT | | | Fundus | | | US | | | Overall | | |
|---|---|---|---|---|---|---|---|---|---|---|---|---|---|---|---|---|---|---|---|---|---|
| | AUC | Sen | Spe | AUC | Sen | Spe | AUC | Sen | Spe | AUC | Sen | Spe | AUC | Sen | Spe | AUC | Sen | Spe | AUC | Sen | Spe |
| ConvNextV2-RawLabel | **.820** | .358 | .935 | .543 | .293 | .693 | .853 | .492 | .757 | **.690** | .442 | .624 | .794 | .351 | .841 | .689 | .519 | .713 | .751 | .434 | .766 |
| ConvNextV2-StandardLabel | .817 | **.436** | **.939** | **.558** | **.330** | **.706** | **.901** | **.568** | **.777** | .671 | **.466** | **.641** | **.873** | **.563** | **.888** | **.774** | **.641** | **.770** | **.787** | **.537** | **.806** |

*Table 13.* **Out-of-Distribution Datasets**

| Dataset | Classes | Out-of-Distribution Characteristics |
|---|---|---|
| COVID-19 | Normal, Bacterial Pneumonia, COVID-19, Viral Pneumonia | Novel disease class (COVID-19) |
| CoronaHack | Normal, Bacterial Pneumonia, Viral Pneumonia, COVID-19 | Novel disease class (COVID-19) |
| CBIS-DDSM | BI-RAD 0-5 | More fine-grained classification (6 BI-RADS categories) compared to VinDr-Mammo (5 categories) and CMMD (binary classification) |
| ISIC-2020 | Malignant, Benign | Different task granularity (binary classification) |
| Jichi | SDR, PPDR, PDR | Different classification scheme for diabetic retinopathy progression compared to Messidor-2 and APTOS 2019's five-stage classification |
| BCSS | Tumor, Stroma, Lymphocytic infiltrate, Necrosis/debris | Different task type (tissue component classification) compared to LC25000's focus on cancer type classification |
| BUSI | Normal, Malignant, Benign | Different task focus (breast lesion classification) compared to other ultrasound datasets (COVID-BLUES, COVID-US) which focus on lung pathology |
| LNDb | nodule $\geq$ 3mm, nodule $<$ 3mm, non-nodule | Different task focus (nodule size classification) compared to other CT datasets like INSPECT and RSPECT which focus on pulmonary embolism |
| PTB-XL-Finegrained | Normal ECG (NORM), Ischemic in inferior leads (ISCI), Non-specific ST changes (NST_), Ischemic in anterior leads (ISCA), Non-specific ischemic (ISC_), ST-T changes (STTC), Right ventricular hypertrophy (RVH), Right atrial overload/enlargement (RAO/RAE), Septal hypertrophy (SEHYP), Left atrial overload/enlargement (LAO/LAE), Anterior myocardial infarction (AMI), Inferior myocardial infarction (IMI), Lateral myocardial infarction (LMI), Posterior myocardial infarction (PMI), Left anterior/left posterior fascicular block (LAFB/LPFB), Incomplete right bundle branch block (IRBBB), AV block (_AVB), Non-specific intraventricular conduction disturbance (IVCD), Complete right bundle branch block (CRBBB), Complete left bundle branch block (CLBBB), Wolff-Parkinson-White syndrome (WPW), Incomplete left bundle branch block (ILBBB) | More fine-grained classification (24 categories) compared to PTB-XL superclass (7 categories in BenchMD). |
| TUEV | Spike and slow wave (SPSW), Generalized periodic epileptiform discharge (GPED), Periodic lateralized epileptiform discharge (PLED), Eye movement (EYEM), artifact (ARTF), and Background (BCKG) | Data consists of pathological patterns, human artifacts, and normal background activity. |

*Table 14.* **Performance metrics of MedVit across different medical imaging datasets.**
**MedVit**

| Dataset | AUC | Sensitivity | Specificity | F1 Score | Accuracy |
|---|---|---|---|---|---|
| CT RSPECT | 0.7609 | 0.3333 | 0.6667 | 0.3248 | 0.9667 |
| CT INSPECT | 0.5495 | 0.2000 | 0.8000 | 0.1764 | 0.9156 |
| MIMIC-CXR | 0.6743 | 0.1447 | 0.9432 | 0.1444 | 0.8737 |
| Fundus JICHI | 0.5464 | 0.2500 | 0.7625 | 0.1951 | 0.8077 |
| Fundus APTOS | 0.4141 | 0.2000 | 0.8000 | 0.0750 | 0.6923 |
| CT LNDB | 0.7009 | 0.4960 | 0.4960 | 0.4395 | 0.7841 |
| ISIC 2020 | 0.2676 | 0.5000 | 0.5000 | 0.4931 | 0.9726 |
| CBIS-DDSM | 0.8333 | 0.5000 | 0.5000 | 0.3000 | 0.4286 |
| BUSI | 0.1111 | 0.5000 | 0.5000 | 0.3333 | 0.5000 |
| LC25000 | 0.9448 | 0.4444 | 0.8631 | 0.3218 | 0.7836 |
| HAM10000 | 0.6446 | 0.2500 | 0.7500 | 0.2143 | 0.8750 |
| VinDr CXR | 0.6546 | 0.1607 | 0.9005 | 0.1278 | 0.8587 |
| CoronaHack | 0.6907 | 0.4583 | 0.7526 | 0.4242 | 0.7333 |
| BCSS | 0.5833 | 0.3333 | 0.6667 | 0.2727 | 0.7949 |
| VinDr Mammo | 0.4205 | 0.3333 | 0.6667 | 0.2716 | 0.7917 |
| COVID-BLUES | 0.3556 | 0.5000 | 0.5000 | 0.3448 | 0.5263 |
| Brain Tumor | 0.8927 | 0.4375 | 0.8229 | 0.3167 | 0.7500 |
| KiTS23 | 0.4063 | 0.5000 | 0.5000 | 0.4706 | 0.8889 |
| CheXpert | 0.6333 | 0.1686 | 0.9021 | 0.1693 | 0.8455 |
| PAD-UFES-20 | 0.6528 | 0.3333 | 0.6667 | 0.0667 | 0.4074 |
| COVID-19 CXR | 0.6972 | 0.3333 | 0.6667 | 0.2222 | 0.6667 |
| COVID-US | 0.8889 | 0.5000 | 0.7500 | 0.3556 | 0.6000 |
| Messidor-2 | 0.0000 | 0.5000 | 0.5000 | 0.4286 | 0.7500 |
| Overall | 0.5793 | 0.3642 | 0.6903 | 0.2821 | 0.7484 |

*Table 15.* **Performance metrics of PMC CLIP across different medical imaging datasets.**
**PMC CLIP**

| Dataset | AUC | Sensitivity | Specificity | F1 Score | Accuracy |
|---|---|---|---|---|---|
| CT RSPECT | 0.7928 | 0.3333 | 0.6667 | 0.3241 | 0.9639 |
| CT INSPECT | 0.5484 | 0.2000 | 0.8000 | 0.1779 | 0.9205 |
| MIMIC-CXR | 0.6613 | 0.1261 | 0.9411 | 0.1234 | 0.8438 |
| Fundus JINCHI | 0.5492 | 0.2560 | 0.7513 | 0.2110 | 0.8308 |
| Fundus APTOS | 0.5863 | 0.2033 | 0.8023 | 0.0928 | 0.7124 |
| CT LNDB | 0.7012 | 0.5000 | 0.5000 | 0.4515 | 0.8232 |
| ISIC 2020 | 0.7351 | 0.5000 | 0.5000 | 0.4955 | 0.9823 |
| CBIS-DDSM | 0.6475 | 0.2069 | 0.8064 | 0.1798 | 0.8024 |
| BUSI | 0.5371 | 0.3233 | 0.6596 | 0.2847 | 0.6616 |
| LC25000 | 0.9842 | 0.7328 | 0.9332 | 0.7058 | 0.8931 |
| HAM10000 | 0.7439 | 0.2191 | 0.8050 | 0.1921 | 0.8670 |
| VinDr CXR | 0.6233 | 0.0935 | 0.9235 | 0.0720 | 0.9110 |
| CoronaHack | 0.8480 | 0.5062 | 0.7729 | 0.4458 | 0.7212 |
| BCSS | 0.6432 | 0.2555 | 0.7534 | 0.1855 | 0.7289 |
| VinDr Mammo | 0.5807 | 0.2062 | 0.8007 | 0.1685 | 0.8651 |
| COVID-BLUES | 0.4175 | 0.4941 | 0.4941 | 0.2304 | 0.2708 |
| Brain Tumor | 0.6160 | 0.3210 | 0.7771 | 0.2656 | 0.6802 |
| KiTS23 | 0.4712 | 0.5000 | 0.5000 | 0.4658 | 0.8718 |
| CheXpert | 0.6680 | 0.1170 | 0.9191 | 0.1287 | 0.7665 |
| PAD-UFES-20 | 0.5427 | 0.2552 | 0.8121 | 0.1436 | 0.7089 |
| COVID-19 CXR | 0.8240 | 0.4106 | 0.8579 | 0.3550 | 0.8138 |
| CMMD | 0.6140 | 0.5226 | 0.5226 | 0.0499 | 0.0513 |
| COVID-US | 0.6091 | 0.3333 | 0.6730 | 0.1961 | 0.6000 |
| Messidor-2 | 0.3898 | 0.2000 | 0.8000 | 0.1473 | 0.8331 |
| CT Hemorrhage | 0.5805 | 0.5000 | 0.5000 | 0.4550 | 0.8350 |
| Brain Tumor 2 | 0.5848 | 0.5441 | 0.5441 | 0.5433 | 0.5882 |
| Overall | 0.6346 | 0.3408 | 0.7237 | 0.2727 | 0.7518 |

*Table 16.* **Performance metrics of RAD-DINO across different medical imaging datasets.**
**RAD-DINO**

| Dataset | AUC | Sensitivity | Specificity | F1 Score | Accuracy |
|---|---|---|---|---|---|
| CT RSPECT | 0.8357 | 0.3333 | 0.6667 | 0.3241 | 0.9639 |
| CT INSPECT | 0.5752 | 0.2000 | 0.8000 | 0.1779 | 0.9205 |
| MIMIC-CXR | 0.7281 | 0.1873 | 0.9404 | 0.2034 | 0.8583 |
| Fundus JICHI | 0.6093 | 0.2500 | 0.7500 | 0.1988 | 0.8301 |
| Fundus APTOS | 0.6589 | 0.2000 | 0.8000 | 0.1320 | 0.7970 |
| CT LNDB | 0.6950 | 0.5000 | 0.5000 | 0.4515 | 0.8232 |
| ISIC 2020 | 0.7753 | 0.4999 | 0.4999 | 0.4955 | 0.9822 |
| CBIS-DDSM | 0.5935 | 0.1988 | 0.8015 | 0.1438 | 0.8225 |
| BUSI | 0.6570 | 0.3333 | 0.6667 | 0.2389 | 0.7056 |
| LC25000 | 0.9809 | 0.6853 | 0.9213 | 0.6288 | 0.8741 |
| HAM10000 | 0.8128 | 0.2392 | 0.8221 | 0.2337 | 0.8752 |
| VinDr CXR | 0.7605 | 0.1179 | 0.9376 | 0.1003 | 0.9152 |
| CoronaHack | 0.9173 | 0.6078 | 0.7995 | 0.5739 | 0.7436 |
| BCSS | 0.6502 | 0.2622 | 0.7535 | 0.1526 | 0.6768 |
| VinDr Mammo | 0.5756 | 0.2000 | 0.8000 | 0.1606 | 0.8682 |
| COVID-BLUES | 0.5775 | 0.5000 | 0.5000 | 0.2066 | 0.2604 |
| Brain Tumor | 0.6725 | 0.3334 | 0.7760 | 0.2437 | 0.6459 |
| KiTS23 | 0.4565 | 0.5000 | 0.5000 | 0.4658 | 0.8718 |
| CheXpert | 0.7516 | 0.2124 | 0.9264 | 0.2347 | 0.8068 |
| PAD-UFES-20 | 0.6125 | 0.2666 | 0.8217 | 0.1722 | 0.7028 |
| COVID-19 CXR | 0.8946 | 0.4199 | 0.8357 | 0.3952 | 0.7753 |
| CMMD | 0.5065 | 0.5000 | 0.5000 | 0.0064 | 0.0064 |
| COVID-US | 0.6750 | 0.5803 | 0.7397 | 0.5056 | 0.6800 |
| Messidor-2 | 0.5311 | 0.2000 | 0.8000 | 0.1473 | 0.8331 |
| CT Hemorrhage | 0.7054 | 0.5000 | 0.5000 | 0.4550 | 0.8350 |
| Brain Tumor 2 | 0.5519 | 0.5000 | 0.5000 | 0.2500 | 0.3333 |
| Overall | 0.6831 | 0.3588 | 0.7253 | 0.2807 | 0.7464 |

*Table 17.* **Performance metrics of SBB2 across different medical imaging datasets.**
**SBB2**

| Dataset | AUC | Sensitivity | Specificity | F1 Score | Accuracy |
|---|---|---|---|---|---|
| CT RSPECT | 0.8556 | 0.3343 | 0.6676 | 0.3260 | 0.9640 |
| CT INSPECT | 0.5624 | 0.2000 | 0.8000 | 0.1779 | 0.9204 |
| MIMIC-CXR | 0.7227 | 0.1714 | 0.9429 | 0.1875 | 0.8561 |
| Fundus JINCHI | 0.7215 | 0.3261 | 0.7697 | 0.3071 | 0.8389 |
| Fundus APTOS | 0.8212 | 0.3514 | 0.8927 | 0.3292 | 0.8647 |
| CT LNDB | 0.6932 | 0.5000 | 0.5000 | 0.4515 | 0.8232 |
| ISIC 2020 | 0.8026 | 0.5000 | 0.5000 | 0.4955 | 0.9823 |
| CBIS-DDSM | 0.6366 | 0.1994 | 0.8007 | 0.1437 | 0.8232 |
| BUSI | 0.6649 | 0.3561 | 0.6773 | 0.3040 | 0.7056 |
| LC25000 | 0.9977 | 0.9622 | 0.9906 | 0.9622 | 0.9849 |
| HAM10000 | 0.8641 | 0.2797 | 0.8438 | 0.2902 | 0.8824 |
| VinDr CXR | 0.6377 | 0.0845 | 0.9188 | 0.0717 | 0.9261 |
| CoronaHack | 0.9127 | 0.7521 | 0.8830 | 0.7532 | 0.8462 |
| BCSS | 0.7249 | 0.4504 | 0.8171 | 0.3938 | 0.7550 |
| VinDr Mammo | 0.5981 | 0.2003 | 0.8013 | 0.1668 | 0.8653 |
| COVID-BLUES | 0.6890 | 0.5977 | 0.5977 | 0.6063 | 0.7500 |
| Brain Tumor | 0.7723 | 0.5411 | 0.8434 | 0.4945 | 0.7640 |
| KiTS23 | 0.6137 | 0.5000 | 0.5000 | 0.4658 | 0.8718 |
| CheXpert | 0.7325 | 0.2204 | 0.9238 | 0.2520 | 0.8027 |
| PAD-UFES-20 | 0.6844 | 0.3048 | 0.8280 | 0.2064 | 0.7264 |
| COVID-19 CXR | 0.9487 | 0.7784 | 0.9393 | 0.7951 | 0.9216 |
| CMMD | 0.3803 | 0.3871 | 0.3871 | 0.4348 | 0.7692 |
| COVID-US | 0.7789 | 0.5318 | 0.7921 | 0.5258 | 0.7333 |
| Messidor-2 | 0.6529 | 0.2000 | 0.8000 | 0.1473 | 0.8331 |
| CT Hemorrhage | 0.7307 | 0.4815 | 0.4815 | 0.4583 | 0.7883 |
| Brain Tumor 2 | 0.7734 | 0.7059 | 0.7059 | 0.7230 | 0.7843 |
| Overall | 0.7297 | 0.4199 | 0.7540 | 0.4027 | 0.8378 |

*Table 18.* **Performance metrics of Swin Transformer across different medical imaging datasets.**
**Swin Transformer**

| Dataset | AUC | Sensitivity | Specificity | F1 Score | Accuracy |
|---|---|---|---|---|---|
| CT RSPECT | 0.8962 | 0.4318 | 0.7605 | 0.4613 | 0.9683 |
| CT INSPECT | 0.6468 | 0.2049 | 0.8035 | 0.1878 | 0.9219 |
| MIMIC-CXR | 0.7620 | 0.1748 | 0.9459 | 0.2113 | 0.8571 |
| Fundus JICHI | 0.7821 | 0.3663 | 0.7967 | 0.3392 | 0.8396 |
| Fundus APTOS | 0.8636 | 0.3632 | 0.9089 | 0.3123 | 0.8849 |
| CT LNDB | 0.6660 | 0.5113 | 0.5113 | 0.4778 | 0.8239 |
| ISIC 2020 | 0.7654 | 0.5313 | 0.5313 | 0.5438 | 0.9780 |
| CBIS-DDSM | 0.7078 | 0.2019 | 0.7973 | 0.1688 | 0.8071 |
| BUSI | 0.6264 | 0.3627 | 0.6742 | 0.1955 | 0.5228 |
| LC25000 | 0.9987 | 0.9573 | 0.9893 | 0.9572 | 0.9829 |
| HAM10000 | 0.8764 | 0.3743 | 0.8557 | 0.3842 | 0.8860 |
| VinDr CXR | 0.6206 | 0.0920 | 0.9210 | 0.0811 | 0.9248 |
| CoronaHack | 0.9231 | 0.7725 | 0.8909 | 0.7580 | 0.8419 |
| BCSS | 0.7141 | 0.3573 | 0.7713 | 0.2427 | 0.6434 |
| VinDr Mammo | 0.6330 | 0.2000 | 0.8000 | 0.1606 | 0.8682 |
| COVID-BLUES | 0.6175 | 0.5989 | 0.5989 | 0.6069 | 0.7708 |
| Brain Tumor | 0.8555 | 0.6215 | 0.8729 | 0.6311 | 0.8160 |
| KiTS23 | 0.4621 | 0.5000 | 0.5000 | 0.4658 | 0.8718 |
| CheXpert | 0.7395 | 0.2009 | 0.9438 | 0.2395 | 0.8140 |
| PAD-UFES-20 | 0.8042 | 0.3996 | 0.8542 | 0.3654 | 0.8100 |
| COVID-19 CXR | 0.9319 | 0.7029 | 0.9268 | 0.7396 | 0.8978 |
| CT Hemorrhage | 0.7562 | 0.5000 | 0.5000 | 0.4550 | 0.8350 |
| COVID-US | 0.8721 | 0.6727 | 0.8619 | 0.6672 | 0.8133 |
| Messidor-2 | 0.6645 | 0.2524 | 0.8072 | 0.2183 | 0.8366 |
| Brain Tumor 2 | 0.9360 | 0.5588 | 0.5588 | 0.5149 | 0.7059 |
| Overall | 0.7649 | 0.4364 | 0.7753 | 0.4154 | 0.8369 |

*Table 19.* **Performance metrics of EVA-2 across different medical imaging datasets.**
**EVA-2**

| Dataset | AUC | Sensitivity | Specificity | F1 Score | Accuracy |
|---|---|---|---|---|---|
| Fundus JINCHI | 0.6808 | 0.2500 | 0.7500 | 0.5250 | 0.8301 |
| COVID-19 CXR | 0.9007 | 0.4372 | 0.8751 | 0.6081 | 0.8335 |
| ISIC 2020 | 0.7291 | 0.5000 | 0.5000 | 0.9736 | 0.9823 |
| HAM10000 | 0.8003 | 0.2625 | 0.8204 | 0.5575 | 0.8684 |
| MIMIC-CXR | 0.7636 | 0.2018 | 0.9291 | 0.3616 | 0.8546 |
| Brain Tumor | 0.6221 | 0.3531 | 0.7756 | 0.2297 | 0.6523 |
| CT LNDB | 0.6405 | 0.5000 | 0.5000 | 0.7434 | 0.8232 |
| CBIS-DDSM | 0.5590 | 0.2916 | 0.8046 | 0.4000 | 0.8151 |
| VinDr Mammo | 0.5240 | 0.2013 | 0.8002 | 0.5388 | 0.8683 |
| CT Hemorrhage | 0.6970 | 0.5000 | 0.5000 | 0.7599 | 0.8350 |
| KiTS23 | 0.6337 | 0.5196 | 0.5196 | 0.0958 | 0.1624 |
| VinDr CXR | 0.6986 | 0.1136 | 0.9326 | 0.4718 | 0.9223 |
| Messidor-2 | 0.5972 | 0.2000 | 0.8000 | 0.4293 | 0.8331 |
| CoronaHack | 0.8826 | 0.4654 | 0.7472 | 0.4578 | 0.6902 |
| Fundus APTOS | 0.8674 | 0.2773 | 0.8562 | 0.4195 | 0.8117 |
| PAD-UFES-20 | 0.5795 | 0.2386 | 0.8073 | 0.2656 | 0.7673 |
| CheXpert | 0.8056 | 0.2596 | 0.8940 | 0.3897 | 0.8034 |
| CMMD | 0.4932 | 0.5000 | 0.5000 | 0.0001 | 0.0064 |
| BUSI | 0.5925 | 0.3333 | 0.6667 | 0.4001 | 0.7056 |
| Brain Tumor 2 | 0.6739 | 0.5000 | 0.5000 | 0.5333 | 0.6667 |
| Overall | 0.6871 | 0.3452 | 0.7239 | 0.4580 | 0.7366 |

*Table 20.* **Performance metrics of ConvNextv2 across different medical imaging datasets.**
**ConvNextv2**

| Dataset | AUC | Sensitivity | Specificity | F1 Score | Accuracy |
|---------|-----|-------------|-------------|----------|----------|
| CT RSPECT | 0.9400 | 0.5598 | 0.8337 | 0.6080 | 0.9748 |
| CT INSPECT | 0.5867 | 0.2098 | 0.8084 | 0.1983 | 0.9217 |
| MIMIC-CXR | 0.7997 | 0.2400 | 0.9480 | 0.2740 | 0.8721 |
| Fundus JICHI | 0.8568 | 0.5949 | 0.8492 | 0.5997 | 0.8798 |
| Fundus APTOS | 0.9380 | 0.5835 | 0.9455 | 0.6105 | 0.9214 |
| CT LNDB | 0.6670 | 0.5311 | 0.5311 | 0.5322 | 0.7722 |
| ISIC 2020 | 0.8529 | 0.5042 | 0.5042 | 0.5039 | 0.9823 |
| CBIS-DDSM | 0.7070 | 0.2435 | 0.8170 | 0.2214 | 0.8299 |
| BUSI | 0.7653 | 0.4862 | 0.7431 | 0.4623 | 0.6853 |
| LC25000 | 0.9999 | 0.9926 | 0.9982 | 0.9926 | 0.9971 |
| HAM10000 | 0.9423 | 0.6485 | 0.9298 | 0.6511 | 0.9251 |
| VinDr CXR | 0.5609 | 0.0852 | 0.9175 | 0.0726 | 0.9270 |
| CoronaHack | 0.9573 | 0.8369 | 0.9260 | 0.8381 | 0.9017 |
| BCSS | 0.8098 | 0.5076 | 0.8180 | 0.4981 | 0.7773 |
| VinDr Mammo | 0.6732 | 0.2536 | 0.8069 | 0.2414 | 0.8693 |
| COVID-BLUES | 0.7318 | 0.6814 | 0.6814 | 0.6742 | 0.7396 |
| Brain Tumor | 0.9293 | 0.7257 | 0.9069 | 0.7136 | 0.8655 |
| KiTS23 | 0.4108 | 0.5000 | 0.5000 | 0.4658 | 0.8718 |
| CheXpert | 0.8037 | 0.2722 | 0.9521 | 0.3160 | 0.8236 |
| PAD-UFES-20 | 0.9068 | 0.5511 | 0.8978 | 0.5873 | 0.8693 |
| COVID-19 CXR | 0.9639 | 0.7474 | 0.9495 | 0.7545 | 0.9337 |
| CMMD | - | 0.4935 | 0.4935 | 0.4951 | 0.9808 |
| COVID-US | 0.8254 | 0.7561 | 0.8841 | 0.7313 | 0.8400 |
| Messidor-2 | 0.8245 | 0.5116 | 0.8684 | 0.5116 | 0.8606 |
| CT Hemorrhage | 0.7485 | 0.5306 | 0.5306 | 0.5188 | 0.8388 |
| Brain Tumor 2 | 0.9585 | 0.9265 | 0.9265 | 0.9328 | 0.9412 |
| Overall | 0.7867 | 0.5374 | 0.8064 | 0.5387 | 0.8770 |

*Table 21.* **Performance metrics of InternViT across different medical imaging datasets.**

**InternViT**

| Dataset | AUC | Sensitivity | Specificity | F1 Score | Accuracy |
|---|---|---|---|---|---|
| CT RSPECT | 0.9005 | 0.4407 | 0.7597 | 0.4791 | 0.9679 |
| CT INSPECT | 0.5954 | 0.2013 | 0.8008 | 0.1808 | 0.9206 |
| MIMIC-CXR | 0.7780 | 0.2196 | 0.9426 | 0.2397 | 0.8667 |
| Fundus JINCHI | 0.8258 | 0.5438 | 0.8224 | 0.4625 | 0.7293 |
| Fundus APTOS | 0.8786 | 0.4505 | 0.9260 | 0.4115 | 0.8941 |
| CT LNDB | 0.6810 | 0.5448 | 0.5448 | 0.5470 | 0.8050 |
| ISIC 2020 | 0.8427 | 0.5040 | 0.5040 | 0.5037 | 0.9820 |
| CBIS-DDSM | 0.6701 | 0.2714 | 0.8217 | 0.2488 | 0.8205 |
| BUSI | 0.7310 | 0.5016 | 0.7391 | 0.4475 | 0.6311 |
| LC25000 | 0.9990 | 0.9738 | 0.9934 | 0.9737 | 0.9895 |
| HAM10000 | 0.9012 | 0.6092 | 0.9210 | 0.5381 | 0.8820 |
| VinDr CXR | 0.6309 | 0.0899 | 0.9184 | 0.0808 | 0.9272 |
| CoronaHack | 0.9303 | 0.7680 | 0.8910 | 0.7515 | 0.8419 |
| BCSS | 0.7856 | 0.4696 | 0.8235 | 0.3876 | 0.7584 |
| VinDr Mammo | 0.6450 | 0.2473 | 0.8176 | 0.2423 | 0.7962 |
| COVID-BLUES | 0.6772 | 0.6592 | 0.6592 | 0.6364 | 0.6875 |
| Brain Tumor | 0.8236 | 0.6134 | 0.8734 | 0.5869 | 0.8135 |
| KiTS23 | 0.5722 | 0.5000 | 0.5000 | 0.4658 | 0.8718 |
| CheXpert | 0.7830 | 0.2333 | 0.9502 | 0.2757 | 0.8168 |
| PAD-UFES-20 | 0.8600 | 0.5162 | 0.8844 | 0.5476 | 0.8501 |
| COVID-19 CXR | 0.9526 | 0.7525 | 0.9484 | 0.7360 | 0.9281 |
| CMMD | 0.2817 | 0.5000 | 0.5000 | 0.4984 | 0.9936 |
| COVID-US | 0.7978 | 0.4864 | 0.7556 | 0.4680 | 0.7067 |
| Messidor-2 | 0.8119 | 0.2977 | 0.8050 | 0.1526 | 0.6720 |
| CT Hemorrhage | 0.7790 | 0.6562 | 0.6562 | 0.6398 | 0.7806 |
| Brain Tumor 2 | 0.9377 | 0.7500 | 0.7500 | 0.7733 | 0.8235 |
| Overall | 0.7720 | 0.4923 | 0.7888 | 0.4721 | 0.8368 |

*Table 22.* **Performance evaluation of GCN across different modalities and datasets.**

**GCN**

| Modality | Dataset | Performance Metrics | | |
|---|---|---|---|---|
| | | AUC | Sensitivity | Specificity |
| Brain Networks | PPMI | 0.973 | 0.922 | 0.897 |
| | ABIDE | 0.626 | 0.596 | 0.586 |
| | ABCD | 0.814 | 0.570 | 0.916 |
| | Average | 0.804 | 0.696 | 0.800 |
| Molecular | PPI | 0.807 | 0.496 | 0.716 |
| | PROTEINS | 0.718 | 0.568 | 0.803 |
| | Average | 0.763 | 0.532 | 0.760 |
| Overall | Average | 0.783 | 0.614 | 0.780 |

*Table 23.* **Performance evaluation of GAT across different modalities and datasets.**

GAT

| Modality | Dataset | Performance Metrics | | |
|---|---|---|---|---|
| | | AUC | Sensitivity | Specificity |
| Brain Networks | PPMI | 0.927 | 0.931 | 0.828 |
| | ABIDE | 0.688 | 0.818 | 0.385 |
| | ABCD | 0.500 | 1.000 | 0.000 |
| | Average | 0.705 | 0.916 | 0.404 |
| Molecular | PPI | 0.926 | 0.572 | 0.798 |
| | PROTEINS | 0.719 | 0.529 | 0.803 |
| | Average | 0.823 | 0.551 | 0.801 |
| Overall | Average | 0.764 | 0.733 | 0.602 |

*Table 24.* **Performance evaluation of Graph Transformers across different modalities and datasets.**

Graph Transformers

| Modality | Dataset | Performance Metrics | | |
|---|---|---|---|---|
| | | AUC | Sensitivity | Specificity |
| Brain Networks | PPMI | 0.950 | 0.862 | 0.957 |
| | ABIDE | 0.743 | 0.707 | 0.683 |
| | ABCD | 0.864 | 0.860 | 0.837 |
| | Average | 0.852 | 0.810 | 0.826 |
| Molecular | PPI | 0.997 | 0.606 | 0.873 |
| | PROTEINS | 0.580 | 0.156 | 0.967 |
| | Average | 0.789 | 0.381 | 0.920 |
| Overall | Average | 0.820 | 0.595 | 0.873 |

*Table 25.* **Performance metrics across different medical imaging datasets.**
**ConvNextv2 (Single Task Training)**

| Dataset | AUC | Sensitivity | Specificity | F1 Score | Accuracy |
|---|---|---|---|---|---|
| CT RSPECT | 0.9680 | 0.7388 | 0.9096 | 0.7472 | 0.9675 |
| CT INSPECT | 0.5524 | 0.2170 | 0.8103 | 0.2070 | 0.7930 |
| MIMIC-CXR | 0.8036 | 0.8100 | 0.6526 | 0.3735 | 0.6805 |
| Fundus JINCHI | 0.8994 | 0.6689 | 0.8794 | 0.6465 | 0.7763 |
| Fundus APTOS | 0.9162 | 0.4880 | 0.9309 | 0.4969 | 0.7613 |
| CT LNDB | 0.6750 | 0.4983 | 0.4983 | 0.4507 | 0.8204 |
| ISIC 2020 | 0.8936 | 0.4957 | 0.9559 | 0.6120 | 0.9478 |
| CBIS-DDSM | 0.6375 | 0.2173 | 0.8070 | 0.1837 | 0.5563 |
| BUSI | 0.6370 | 0.4359 | 0.7013 | 0.4064 | 0.5381 |
| LC25000 | 1.0000 | 0.9971 | 0.9993 | 0.9971 | 0.9971 |
| HAM10000 | 0.9531 | 0.7154 | 0.9416 | 0.6850 | 0.8018 |
| VinDr CXR | - | 0.3353 | 0.5809 | 0.1281 | 0.7099 |
| CoronaHack | 0.9461 | 0.8404 | 0.9225 | 0.8400 | 0.8462 |
| BCSS | 0.8431 | 0.6276 | 0.8571 | 0.6053 | 0.6219 |
| VinDr Mammo | 0.6864 | 0.3138 | 0.8179 | 0.2802 | 0.6747 |
| COVID-BLUES | 0.5282 | 0.5000 | 0.5000 | 0.2066 | 0.2604 |
| Brain Tumor | 0.8291 | 0.4801 | 0.8188 | 0.4555 | 0.4721 |
| KiTS23 | 0.3235 | 0.5000 | 0.5000 | 0.4658 | 0.8718 |
| CheXpert | - | 0.6351 | 0.7669 | 0.4516 | 0.7603 |
| PAD-UFES-20 | 0.8112 | 0.3295 | 0.8531 | 0.3430 | 0.5425 |
| COVID-19 CXR | 0.9168 | 0.5689 | 0.8947 | 0.6013 | 0.7419 |
| CMMD | 0.5204 | 0.4860 | 0.4860 | 0.4913 | 0.9658 |
| COVID-US | 0.5000 | 0.3333 | 0.6667 | 0.0920 | 0.1600 |
| Messidor-2 | 0.8301 | 0.2590 | 0.8073 | 0.2117 | 0.5886 |
| CT Hemorrhage | 0.7666 | 0.8884 | 0.2471 | 0.5724 | 0.7825 |
| Brain Tumor 2 | 0.7734 | 0.0000 | 1.0000 | 0.4000 | 0.6667 |

*Table 26.* **Performance metrics of Llava-Med across different medical imaging datasets.** Note the overall row is averaged across dataset, which is different from the modality-wise average in Table 6.

**Llava-Med**

| Dataset | Zero-Shot | | | | Fine-Tuned | | | |
|---|---|---|---|---|---|---|---|---|
| | Accuracy | Sensitivity | Specificity | F1 Score | Accuracy | Sensitivity | Specificity | F1 Score |
| MIMIC-CXR | 0.001 | 0.214 | 0.786 | 0.063 | 0.001 | 0.214 | 0.794 | 0.090 |
| CheXpert | 0.000 | 0.077 | 0.924 | 0.036 | 0.000 | 0.154 | 0.847 | 0.093 |
| VinDr-CXR | 0.045 | 0.083 | 0.916 | 0.016 | 0.702 | 0.083 | 0.917 | 0.069 |
| COVID-19 | 0.009 | 0.250 | 0.750 | 0.005 | 0.455 | 0.250 | 0.750 | 0.156 |
| CoronaHack | 0.388 | 0.333 | 0.667 | 0.186 | 0.388 | 0.333 | 0.667 | 0.186 |
| Brain Tumor | 0.393 | 0.445 | 0.799 | 0.300 | 0.292 | 0.250 | 0.750 | 0.113 |
| Brain Tumor 2 | 0.333 | 0.500 | 0.500 | 0.250 | 0.667 | 0.500 | 0.500 | 0.400 |
| BUSI | 0.579 | 0.391 | 0.697 | 0.357 | 0.558 | 0.333 | 0.667 | 0.239 |
| COVID-BLUES | 0.365 | 0.557 | 0.557 | 0.353 | 0.740 | 0.500 | 0.500 | 0.425 |
| COVID-US | 0.400 | 0.333 | 0.667 | 0.190 | 0.440 | 0.333 | 0.667 | 0.204 |
| CBIS | 0.008 | 0.202 | 0.799 | 0.005 | 0.560 | 0.200 | 0.800 | 0.144 |
| VinDr-Mammo | 0.047 | 0.200 | 0.800 | 0.018 | 0.670 | 0.200 | 0.800 | 0.161 |
| CMMD | 0.091 | 0.206 | 0.800 | 0.045 | 0.994 | 0.500 | 0.500 | 0.498 |
| ISIC 2020 | 0.982 | 0.500 | 0.500 | 0.496 | 0.982 | 0.333 | 0.667 | 0.330 |
| HAM10000 | 0.110 | 0.201 | 0.800 | 0.047 | 0.670 | 0.201 | 0.800 | 0.162 |
| PAD-UFES-20 | 0.307 | 0.186 | 0.800 | 0.107 | 0.368 | 0.200 | 0.800 | 0.108 |
| Messidor-2 | 0.151 | 0.157 | 0.832 | 0.048 | 0.583 | 0.200 | 0.800 | 0.147 |
| APTOS | 0.492 | 0.200 | 0.800 | 0.132 | 0.492 | 0.200 | 0.800 | 0.132 |
| Jichi | 0.660 | 0.250 | 0.750 | 0.199 | 0.660 | 0.250 | 0.750 | 0.199 |
| LNDb | 0.823 | 0.500 | 0.500 | 0.452 | 0.645 | 0.541 | 0.541 | 0.520 |
| Kits23 | 0.128 | 0.500 | 0.500 | 0.114 | 0.872 | 0.500 | 0.500 | 0.466 |
| Brain CT | 0.835 | 0.500 | 0.500 | 0.455 | 0.835 | 0.500 | 0.500 | 0.455 |
| INSPECT | 0.004 | 0.194 | 0.800 | 0.006 | 0.801 | 0.200 | 0.800 | 0.178 |
| Cholec 80 | 0.000 | 1.000 | 0.000 | 0.243 | 0.296 | 0.143 | 0.857 | 0.069 |
| Overall | 0.298 | 0.333 | 0.685 | 0.172 | 0.570 | 0.297 | 0.707 | 0.231 |

*Table 27.* **Model Performance of BIOT Variants with TUEV Finetuning**

| Pretrain Encoder | Num Shots | Finetune | AUC | Sens | Spec |
|---|---|---|---|---|---|
| BIOT | 1 | TUEV | .589 | .232 | .841 |
| | 8 | | .688 | .298 | .859 |
| | 32 | | .740 | .359 | .873 |
| | full | | .856 | .466 | .908 |
| BIOT-pretrain-PREST | 1 | TUEV | .609 | .239 | .835 |
| | 8 | | .754 | .372 | .871 |
| | 32 | | .781 | .363 | .899 |
| | full | | .898 | .580 | .918 |
| BIOT-pretrain-SHHS+PREST | 1 | TUEV | .630 | .239 | .836 |
| | 8 | | .723 | .305 | .861 |
| | 32 | | .777 | .382 | .879 |
| | full | | .880 | .586 | .914 |
| BIOT-pretrain-IIIC+TUAB | 1 | TUEV | .553 | .179 | .836 |
| | 8 | | .759 | .332 | .873 |
| | 32 | | .807 | .410 | .894 |
| | full | | .869 | .510 | .905 |

*Table 28.* **Model Performance on Different ECG Datasets**

| Model Name | PTB-XL | | | ChapmanShao | | | CPSC | | | Ga | | | Overall | | |
|---|---|---|---|---|---|---|---|---|---|---|---|---|---|---|---|
| | AUC | Sens | Spe | AUC | Sens | Spe | AUC | Sens | Spe | AUC | Sens | Spe | AUC | Sens | Spe |
| Transformer | .785 | .239 | .885 | .797 | .312 | .919 | .579 | .213 | .883 | .671 | .251 | .891 | .708 | .253 | .895 |
| ECG-JEPA | .906 | .591 | .918 | .858 | .392 | .935 | .979 | .797 | .980 | .767 | .294 | .899 | .877 | .518 | .877 |
| UniTS | .669 | .150 | .861 | .656 | .146 | .859 | .641 | .143 | .857 | .598 | .158 | .863 | .641 | .149 | .860 |

*Table 29.* **Model Performance of ECG-JEPA and UniTS Variants with PTB-XL Finetuning.** Ours is the model pretrained on CLIMB dataset with PTB-XL removed.

| Model | Pretrain Encoder | Num Shots | Finetune | AUC | Sens | Spe |
|---|---|---|---|---|---|---|
| ECG-JEPA | Ours | 1 | PTB-XL | .633 | .048 | .956 |
| | | 8 | | .760 | .113 | .966 |
| | | full | | .895 | .210 | .980 |
| | PTB-XL | 1 | PTB-XL | .512 | .043 | .956 |
| | | 8 | | .472 | .043 | .956 |
| | | full | | .868 | .195 | .979 |
| UniTS | Ours | 1 | PTB-XL | .527 | .430 | .537 |
| | | 8 | | .549 | .641 | .370 |
| | | full | | .673 | .025 | .993 |
| | PTB-XL | 1 | PTB-XL | .512 | .674 | .371 |
| | | 8 | | .470 | .322 | .658 |
| | | full | | .688 | .053 | .984 |

*Table 30.* **Model Performance of ECG-JEPA Variants with Different Pretraining.** Ours is the model pretrained on CLIMB dataset.

| Pretrain | Finetuned | AUC | Sens | Spe |
|---|---|---|---|---|
| PTB-XL | PTB-XL | .776 | .237 | .883 |
| CPSC | CPSC | .682 | .144 | .857 |
| ChapmanShao | ChapmanShao | .771 | .248 | .899 |
| Ga | Ga | .474 | .143 | .857 |
| Ours | PTB-XL | .805 | .329 | .902 |
| | CPSC | .843 | .343 | .918 |
| | ChapmanShao | .831 | .355 | .932 |
| | Ga | .704 | .273 | .894 |

*Table 31.* **Model Performance of UniTS Variants with Different Pretraining.** Ours is the model pretrained on CLIMB dataset and Original is the pretrained encoder provided by the UniTS model.

| Pretrain | Train | Eval | AUC | Sens | Spe |
|---|---|---|---|---|---|
| Original | PTB-XL | PTB-XL | .669 | .150 | .861 |
| | CPSC | CPSC | .641 | .143 | .857 |
| | ChapmanShao | ChapmanShao | .656 | .146 | .859 |
| | Ga | Ga | .598 | .158 | .863 |
| Original | Ours | PTB-XL | .772 | .242 | .886 |
| | | CPSC | .871 | .336 | .919 |
| | | ChapmanShao | .812 | .333 | .927 |
| | | Ga | .742 | .271 | .894 |

*Table 32.* Dataset membership prediction results validating OOD dataset distinctiveness

| OOD Dataset | Balanced Accuracy | AUC | F1 Score |
|---|---|---|---|
| BCSS | 99.8 | 1.000 | 0.928 |
| CBIS-DDSM | 99.9 | 1.000 | 0.801 |
| CoronaHack | 99.7 | 0.999 | 0.371 |
| COVID-19 | 53.2 | 0.953 | 0.112 |
| BUSI | 99.9 | 1.000 | 0.997 |
| Jichi | 99.8 | 0.999 | 0.758 |
| ISIC-2020 | 99.7 | 0.999 | 0.841 |

*Table 33.* **ECG unsupervised pretraining results comparing dataset-specific vs. CLIMB pretraining.** The results demonstrate that pretraining on a diverse multi-dataset collection effectively improved downstream performance on the target dataset.

| Model | Pretrain Dataset | Evaluation Dataset | AUC | Sensitivity | Specificity |
|---|---|---|---|---|---|
| ECG-JEPA | PTB-XL | PTB-XL | .868 | .195 | .979 |
| ECG-JEPA | CLIMB | PTB-XL | **.895** | **.210** | **.980** |

*Table 34.* **EEG pretraining comparison across different training strategies.** The combination of pretraining and multitask learning achieved the best overall results, with pretraining appearing to play a more significant role than multitask learning in EEG model performance.

| Model Name | IIIC | | | | TUEV | | | | TUAB | | | | Overall | | | |
|---|---|---|---|---|---|---|---|---|---|---|---|---|---|---|---|---|
| | AUC | Sens | Spe | F1 | AUC | Sens | Spe | F1 | AUC | Sens | Spe | F1 | AUC | Sens | Spe | F1 |
| No Pretrain | .854 | .510 | .905 | .499 | .856 | .466 | .908 | .371 | **.879** | **.798** | **.798** | **.799** | .863 | .591 | .870 | .556 |
| MTL Only | .848 | .484 | .901 | .475 | .903 | .386 | .932 | .387 | .844 | .764 | .764 | .761 | .865 | .545 | .866 | .541 |
| PT+MTL | **.862** | **.546** | **.911** | **.531** | .878 | **.549** | **.917** | **.397** | .869 | .794 | .794 | .795 | **.870** | **.630** | **.874** | **.574** |

*Table 35.* **Vision model performance comparison across different pretraining strategies.** MAE: Masked Autoencoder following ConvNeXTv2 (Liu et al., 2022), CL: Contrastive Learning using CLIP-style approach from InternVL (Chen et al., 2024b), MTL: MultiTask Learning. Contrary to expectations, neither MAE nor contrastive learning pretraining improved model performance. The multitask learning approach alone achieved the best results across all metrics.

| PT Method | Model | CXR | | | Mammo | | | Derm | | | CT | | | Fundus | | | US | | | Overall | | |
|---|---|---|---|---|---|---|---|---|---|---|---|---|---|---|---|---|---|---|---|---|---|---|
| | | AUC | Sen | Spe | AUC | Sen | Spe | AUC | Sen | Spe | AUC | Sen | Spe | AUC | Sen | Spe | AUC | Sen | Spe | AUC | Sen | Spe |
| MAE + MTL on CLIMB | ConvNeXTv2 | .801 | .379 | .923 | .489 | .276 | .671 | .795 | .414 | .738 | **.699** | .430 | .614 | .757 | .325 | .835 | .705 | .484 | .687 | .733 | .433 | .766 |
| MAE + CL + MTL on CLIMB | InternViT | .753 | .338 | .906 | .500 | .287 | .689 | .767 | .353 | .715 | .678 | .409 | .595 | .683 | .298 | .825 | .683 | .532 | .689 | .697 | .394 | .743 |
| **Only MTL on CLIMB** | ConvNeXTv2 | **.817** | **.436** | **.939** | **.558** | **.330** | **.706** | **.901** | **.568** | **.777** | .671 | **.466** | **.641** | **.873** | **.563** | **.888** | **.774** | **.641** | **.770** | **.787** | **.537** | **.806** |

*Table 36.* **Performance Comparison with Dataset Specific Encoders.** Ours is the result of multitask universal encoders trained on CLIMB, whereas Dataset SoTA is the state-of-the-art encoder specially tuned and optimized for that particular task. Note we aim to preserve as much information as possible from each datasets and avoids binning classes, so the task for some datasets, like VinDr-CXR and CBIS-DDSM, may be different from the ones used in other works. The results are taken directly from the papers.

| Dataset | Ours (AUC) | Dataset SoTA (AUC) | Source/Method Name | Reference |
|---|---|---|---|---|
| PTB-XL | 0.895 | 0.896 | ECG JEPA | Link |
| Chapman-Shaoxing | 0.858 | 0.979 | X3ECG w/ HC + DDI | – |
| Georgia | 0.767 | – | – | – |
| CPSC | 0.978 | 0.974 | ECG JEPA | Link |
| MIMIC-CXR | 0.800 | 0.834 | ChexClusion | Link |
| CheXpert | 0.783 | 0.933 | CFT | Link |
| VinDr-CXR | 0.631 (14 classes) | 0.961 (6 classes) | Paper | Link |
| VinDr-Mammo | 0.673 (5 classes) | 0.840 (2 classes) | MaMT4 | Link |
| CBIS-DDSM | 0.707 (5 classes) | 0.900 (2 classes) | MEWOA | Link |
| ISIC 2020 | 0.853 | 0.943 | Kaggle Leaderboard | Link |
| HAM10000 | 0.942 | 0.943 | Paper | Link |
| PAD-UFES-20 | 0.907 | 0.920 | EfficientNetB3 | Link |
| Messidor-2 | 0.825 (5 classes) | 0.971 (2 classes) | Paper | Link |
| APTOS 2019 | 0.938 | 0.920 | Survey | Link |
| Jichi | 0.857 | – | – | – |
| LNDb | 0.681 | 0.831 (Fleischner_kw) | Leaderboard | Link |
| INSPECT | 0.595 | 0.771 (Accuracy) | Paper | Link |
| BUSI | 0.765 | – | – | – |
| COVID-19 | 0.964 | – | – | – |
| Brain Tumor | 0.929 | – | – | – |
| Brain MRI | 0.959 | – | – | – |
| Kits23 | 0.572 | 0.835 (DICE) | Leaderboard | Link |
| Hemorrhage | 0.779 | – | – | – |
| CoronaHack | 0.957 | – | – | – |
| COVID-BLUES | 0.732 | – | – | – |
| COVID-US | 0.825 | 0.94 | Review | Link |
| IIIC | 0.862 | 0.580 (Balanced Acc.) | – | – |
| TUAB | 0.882 | 0.882 | – | – |
| TUEV | 0.898 | 0.528 (Balanced Acc.) | – | – |
| PROTEINS | 0.719 | 0.849 (Acc) | HGP-SL | Link |
| PPI | 0.997 | 0.997 (Acc) | g2-MLP | Link |
| RSPECT | 0.94 | – | – | – |
| LC25000 | 1.000 | 1.000 | SE Networks | Link |
| BCSS | 0.810 | 0.710 (mIoU) | MLP-MF | Link |

