# OpenReview forum: "CLIMB: Data Foundations for Large Scale Multimodal Clinical Foundation Models"
_ICML.cc/2025/Conference — ICML 2025 poster_

### Official Review · Reviewer_ovwX · 2025-02-27

**Overall Recommendation:** 3

**Summary:**

This paper introduces the Clinical Large-scale Integrative Multi-modal Benchmark (CLIMB), a benchmark unifying diverse clinical data across imaging, language, temporal, and graph modalities. The dataset comprises 4.51 million patients distributed across multiple modalities. The authors conduct extensive empirical evaluations and demonstrate three key findings: (1) Multitask pretraining significantly improves performance on understudied domains. (2) Models pretrained on CLIMB demonstrate improved few-shot transfer learning capabilities, and (3) Unimodal encoder performance translates well to multimodal tasks when paired with appropriate fusion strategies.

**Claims And Evidence:**

Generally, the claims made in this submission are well-supported.
1. Multitask pretraining improving performance on understudied domains: the authors provide evidence in Figure 4, showing substantial AUC improvements for understudied modalities.
2. The second claim regarding few-shot transfer is supported by experiments in Figure 7, demonstrating improvements across various modalities including ultrasound, CT, and ECG domains when using CLIMB pretraining versus standard pertaining.
3. The third claim about fusion strategies is supported by results in Table 5, showing that different fusion methods (late fusion, MLP, cross-attention) exhibit varying effectiveness depending on the task complexity.
However,

**Essential References Not Discussed:**

A few references that should be included like recent work on multimodal foundation models in healthcare like BiomedCLIP published in NEJM AI (https://ai.nejm.org/doi/full/10.1056/AIoa2400640), and works on self-supervised learning for medical imaging.

**Experimental Designs Or Analyses:**

I reviewed the experimental designs and analyses in detail and found them to be generally sound. The authors use appropriate statistical measures and the multitask training setup is well-designed.

**Methods And Evaluation Criteria:**

The proposed methods and evaluation criteria are well-suited for the problem.

**Other Comments Or Suggestions:**

1. The authors should discuss why multitask pre-training instead of self-supervised methods (like masked image modeling or contrastive learning), which might better leverage the large-scale nature of CLIMB.
2. A more thorough analysis separating the effects of increased data quantity from the benefits of multitask learning would strengthen the paper. For example, comparing with models trained on equivalent amounts of data but without task sharing.
3. The paper would benefit from a more detailed analysis of when multitask learning helps versus when it potentially causes negative transfer, as Figure 4 shows varying impacts across datasets.

**Other Strengths And Weaknesses:**

**Strengths**
1. The scale and diversity of the CLIMB benchmark is impressive.
2. The focus on understudied modalities and underrepresented regions is valuable for addressing biases in clinical AI.


**Weaknesses**
1. The paper focuses on supervised multitask pretraining rather than exploring self-supervised approaches, which might be more data-efficient and better leverage the unlabeled portions of clinical data, also no bias problem.
2. The improvement in understudied domains might be primarily attributable to increased data exposure rather than true cross-task knowledge transfer.

**Questions For Authors:**

1. Have you explored self-supervised pretraining approaches as an alternative to supervised multitask learning?
2. To what extent are the improvements in understudied domains attributable to multitask learning specifically versus simply having access to more training data? Have you conducted ablation studies with equivalent data quantities in single-task settings?
3. Figure 4 shows that some datasets experience minimal gains or even slight performance decreases with multitask pretraining. What factors determine whether dataset benefits from multitask learning, and could you elaborate on potential cases of negative transfer?
4. How did you address potential biases in the datasets, particularly for underrepresented regions?

**Relation To Broader Scientific Literature:**

The paper positions itself well within the broader scientific literature on clinical AI and multimodal learning. The authors compare CLIMB to existing medical benchmarks like BenchMD, PMC-VQA, GMAI-MMBench, and CARES.

**Theoretical Claims:**

The paper does not contain formal proofs for theoretical claims.

---

> ### Author Rebuttal · Authors · 2025-04-01
>
> W1-2, Q1-2, C1-2: In our previous experiments, both pretraining and multitask learning are performed in Exp. 1. We added experiments below comparing unsupervised pretraining vs supervised multitask learning.
>
> First, we found time series models benefited substantially from unsupervised pretraining. As illustrated in App. Tab. 26, we conducted unsupervised masked autoencoder (MAE) pretraining and compared it with pretraining on target dataset only:
>
> | Model | PT Dataset | Eval. Dataset | AUC | Sens. | Spec. |
> | --- | --- | --- | --- | --- | --- |
> | ECG-JEPA | PTB-XL | PTB-XL | .868 | .195 | .979 |
> | ECG-JEPA | CLIMB | PTB-XL | .895 | .210 | .980 |
>
> We found pretraining on diverse data effectively improved the downstream outcome on the target dataset. Similarly, for EEG datasets, as shown in App. Table 24, pretraining on a diverse range of data improved the performance for 2/3 datasets and overall:
>
> | Model Name | IIIC |  |  |  | TUEV |  |  |  | TUAB |  |  |  | Overall |  |  |  |
> |------------|------|------|------|------|------|------|------|------|------|------|------|------|------|------|------|------|
> |  | AUC | Sens | Spe | F1 | AUC | Sens | Spe | F1 | AUC | Sens | Spe | F1 | AUC | Sens | Spe | F1 |
> | Single Task | .854 | .510 | .905 | .499 | .856 | .466 | .908 | .371 | **.879** | **.798** | **.798** | **.799** | .863 | .591 | .870 | .556 |
> | MTL Only | .848 | .484 | .901 | .475 | .903 | .386 | .932 | .387 | .844 | .764 | .764 | .761 | .865 | .545 | .866 | .541 |
> | Pretrain+MTL | **.862** | **.546** | **.911** | **.531** | **.878** | **.549** | **.917** | **.397** | .869 | .794 | .794 | .795 | **.870** | **.630** | **.874** | **.574** |
>
> While combining pretrain and multitask learning achieves the best results, pretraining seems to play a larger role than multitask learning in EEG models.
>
> On the other hand, we showed in Fig. 4 that multitask learning effectively improved the vision encoder’s performance. Surprisingly, vision models did not benefit from further pretraining, and only multitask learning helps:
>
> | PT Method | Model | CXR | | | Mammo | | | Derm | | | CT | | | Fundus | | | US | | | Overall | | |
> |---|---|---|---|---|---|---|---|---|---|---|---|---|---|---|---|---|---|---|---|---|---|---|
> | | | AUC | Sen | Spe | AUC | Sen | Spe | AUC | Sen | Spe | AUC | Sen | Spe | AUC | Sen | Spe | AUC | Sen | Spe | AUC | Sen | Spe |
> | MAE + MTL on CLIMB | ConvNeXTv2 | 0.801 | 0.379 | 0.923 | 0.489 | 0.276 | 0.671 | 0.795 | 0.414 | 0.738 | **0.699** | 0.430 | 0.614 | 0.757 | 0.325 | 0.835 | 0.705 | 0.484 | 0.687 | 0.733 | 0.433 | 0.766 |
> | MAE + CL + MTL on CLIMB | InternViT | 0.753 | 0.338 | 0.906 | 0.500 | 0.287 | 0.689 | 0.767 | 0.353 | 0.715 | 0.678 | 0.409 | 0.595 | 0.683 | 0.298 | 0.825 | 0.683 | 0.532 | 0.689 | 0.697 | 0.394 | 0.743 |
> | **Only MTL on CLIMB** | ConvNeXTv2 | **0.817** | **0.436** | **0.939** | **0.558** | **0.330** | **0.706** | **0.901** | **0.568** | **0.777** | 0.671 | **0.466** | **0.641** | **0.873** | **0.563** | **0.888** | **0.774** | **0.641** | **0.770** | **0.787** | **0.537** | **0.806** |
>
> Here, MAE = Masked Autoencoder, CL = Contrastive Learning and MTL = Multitask Learning. For MAE, we followed the same approach as in [ConvNeXTv2](https://arxiv.org/pdf/2301.00808). For contrastive learning, we followed the CLIP-style approach as outlined in [InternVL](https://arxiv.org/abs/2312.14238). In the above vision experiment, neither MAE nor CL pretraining improved the model’s performance for downstream tasks. One hypothesis is that these models are already heavily pre-trained on massive unlabeled natural image corpora, so an additional masked image or contrastive-style phase on clinical data doesn’t substantially shift or enrich their learned representation. We welcome the community’s contribution to better leveraging the diverse labeled data available in CLIMB.
>
> Q3, C3: We found that datasets with novel tasks, Understudied modalities or from Underrepresented regions, as defined in Lines 1219-1290, benefit the most from multitask learning. We believe this is due to the scarcity of data in these categories, which makes data from other modalities or tasks, albeit not closely related, beneficial. On the other hand, if a dataset has a large number of samples or fairly saturated performance on a specialized diagnostic task, then multitask learning may not add much beyond what single-task training already learns.
>
> Q4: During the construction of CLIMB, we aim to include as many datasets from underrepresented regions as possible. This includes datasets from geographically underrepresented regions, as well as data from developing countries where the data has been historically scarce, as defined in Lines 1274-1290. We included a list of dataset collection locations in the App. Table 8. In summary, 8 out of 30 (26.7%) of the datasets with known source locations come from underrepresented regions, a percentage significantly higher than all datasets available from public sources.

---

### Official Review · Reviewer_ZboR · 2025-03-09

**Overall Recommendation:** 3

**Summary:**

This paper introduces a large-scale clinical multimodal benchmark. The authors conduct multitask pretraining, few-shot transfer, and multimodal fusion. Based on the constructed data, they provide extensive experiment results to answer the proposed research questions.

**Claims And Evidence:**

Yes

**Essential References Not Discussed:**

They cover most of the recent works. But lack some of them, such as Wang, Xiaochen, et al. "Unity in diversity: Collaborative pre-training across multimodal medical sources." Proceedings of the 62nd Annual Meeting of the Association for Computational Linguistics (Volume 1: Long Papers). 2024.

**Experimental Designs Or Analyses:**

Yes

**Methods And Evaluation Criteria:**

Yes

**Other Comments Or Suggestions:**

No

**Other Strengths And Weaknesses:**

Strengths:

1, This is a comprehensive multimodal clinical benchmark covering different modalities.

2, This paper introduces the data construction, experiment, evaluation, and discussion.

3, The writing is easy to follow.

Weakness:

1, Figure 3.a and 3.b seem not that clear to understand.

2, I am wondering what computation resources do the users need to implement this approach.

**Questions For Authors:**

Please see the weakness.

Also, is all the data used in this work public? Are the data collection and cleaning scripts provided?

**Relation To Broader Scientific Literature:**

This work provides contribution to the medical and healthcare domain.

**Theoretical Claims:**

There are no proofs or theoretical claims.

---

> ### Author Rebuttal · Authors · 2025-04-01
>
> Thanks for your feedback regarding Figures 3.a and 3.b. Both figures illustrate two different experiments we conducted on CLIMB: multitask learning and transfer learning. The figures display example data from CLIMB (such as x-rays, CT scans, etc.) as inputs to the model and display evaluation on the right. For Figure 3.a, we train a single model on multiple clinical tasks across different medical modalities and evaluate it on each individual dataset. The goal is to evaluate whether a multitask pre-train model can generalize across diverse tasks, including understudied ones. This setup helps assess if shared representations from CLIMB improve generalization within each dataset. In Figure 3.b, a multitask pre-trained model is applied to a new task with limited data to see whether the model adapts its learned representations to the new task, despite having few samples available. The goal is to determine if exposure to a broader range of tasks in CLIMB helps compensate for data scarcity in specific datasets.
>
> To improve clarity, we will implement the following revisions: 1) provide a more detailed figure description 2) move labels for "understudied" and “few samples" to the side to avoid implying they apply to the entire column. We will also cite the recent works mentioned in the review.
>
> Regarding the computational resources, as described in Appendix C.4.2, all experiments are performed on a server with 8xH200 GPUs for the best performance. At least 20TB of storage is needed if you would like to train a model on the entire dataset. With that said, our model is small enough to fit on one GPU with 24GB of VRAM, and the entire training on one GPU would take less than a week.
>
> All data used in this work is public and can be accessed easily using our framework, as outlined in our response to Reviewer pTYB. In summary, a user only needs to complete one-time registrations of two accounts and complete one CITI training, which takes less than 4 hours. Our framework will then prompt the user to agree to agreements, and downloading each dataset will take less than 10 seconds of human labor. Our framework will also handle the data cleaning, processing and standardization of formats.

---

### Official Review · Reviewer_SVNB · 2025-03-12

**Overall Recommendation:** 3

**Summary:**

This paper introduces the Clinical Large-scale Integrative Multimodal Benchmark (CLIMB), which integrates diverse clinical data across imaging, language, time-series, and graph modalities. CLIMB consists of 4.51 million patient samples (19.01 terabytes), covering 2D imaging, 3D video, and multimodal data. Empirical evaluations demonstrate that multitask pretraining significantly enhances performance, with improvements of 29% in ultrasound analysis and 23% in ECG analysis over single-task learning. Additionally, pretraining on CLIMB improves generalization to new tasks, while strong unimodal encoders effectively contribute to multimodal models when combined with appropriate fusion strategies.


**update after rebuttal**

In my initial review, I had some concerns about the novelty of the proposed unified framework, and the results appeared fairly predictable.

In their response, the authors provided additional clarification on the taxonomy standardization, which I found helpful. Moreover, the improved performance on underrepresented regions and modalities (ultrasound, CT, EEG) is particularly meaningful in the healthcare domain.

Therefore, I am increasing my initial score to 3, leaning toward acceptance.

**Claims And Evidence:**

The primary contribution of this work is the standardization of multiple publicly available datasets to demonstrate that pretraining on a large-scale medical dataset enhances downstream task performance. However, the results appear somewhat predictable, as prior research has already established that large-scale pretraining generally improves model performance.

**Essential References Not Discussed:**

Regarding the primary research questions addressed with CLIMB, I believe that similar findings have already been demonstrated in prior work. Specifically, previous studies have shown that training a foundation model on large-scale medical datasets improves downstream task performance. The following papers provide strong evidence in support of this claim:
- Med Gemini (arXiv:2405.03162)
- BiomedGPT (arXiv:2305.17100)
- RadFM (arXiv:2308.02463)
I assume that the authors are already well aware of these works. Given this, a clearer differentiation of CLIMB’s unique contributions—beyond reaffirming known benefits of large-scale pretraining—would further strengthen the impact of this study.

**Experimental Designs Or Analyses:**

The experiments are designed to address the following three research questions:
Q1: Can multitask-pretrained clinical models perform consistently across multiple tasks, particularly for understudied tasks?
Q2: How well do multitask-pretrained clinical models transfer to new tasks within the same clinical modality, especially when data is limited?
Q3: Can multitask-pretrained unimodal models be effectively fused to tackle multimodal clinical tasks?

If the novelty and significance of these research questions are set aside, the experiments themselves are adequately designed to address them. However, regarding the first and third questions, the conclusion is fairly predictable—multitask learning tends to be particularly beneficial in scenarios where data or research focus has historically been limited.

For the second question, the paper concludes that the large-scale pretraining dataset enables efficient learning of novel tasks with limited samples, yielding consistent performance improvements across all modalities. While this finding aligns with the results presented, it largely re-claims well-established principles in machine learning.

**Methods And Evaluation Criteria:**

This paper does not propose a novel method, so evaluating a specific methodology is not applicable. However, in terms of evaluation criteria, this work makes contributions by introducing a unified framework for holistic training and benchmarking of clinical models through standardization of data loading and prediction tasks.
The key contributions include:

**Standardized Task Formulation**
- All tasks are framed as multi-label classification across different clinical modalities.
- Terminology variations are standardized, and similar concepts (e.g., Lung Opacity from CheXpert and Infiltration from VinDR-CXR) are merged to create a consistent and clinically meaningful vocabulary.

**Question-Answering (QA) Reformulation**

The dataset is also structured as a closed-choice question-answering (QA) task, named CLIMB-QA, to support comparative evaluation of large vision-language models (LVLMs).

**Unified Data Processing Interface**
- A standardized pipeline is provided for downloading and processing datasets into a unified format, ensuring compatibility for large-scale mixed-modality pretraining (subject to dataset-specific consent agreements).

Beyond the significance and novelty of these contributions, the rationale behind formulating multiple heterogeneous tasks in a unified manner is well-grounded and conceptually sound.

**Other Comments Or Suggestions:**

**Comment on Figure 4**

The authors define underrepresented regions and understudied modalities. For underrepresented regions, Brazil and China are mentioned. Could you clarify why these countries are considered underrepresented in the medical domain? Do the individual datasets listed in Table 6 lack sufficient samples from Asian and South American cohorts? Providing statistical evidence would make this argument more convincing.

Regarding understudied modalities, ultrasound and CT scans are included. However, this characterization may be somewhat misleading, as both modalities play a crucial role in daily clinical practice. I believe the authors intended to highlight the relative scarcity of publicly available datasets for these modalities compared to others. It would be helpful to clarify this point more explicitly.

**Comment on dataset construction**

I am particularly interested in how similar medical concepts are merged or how terminology is standardized, as this is a crucial preprocessing step when handling large-scale medical data for model training.

For chest X-ray datasets, did you merge lung opacity and infiltration into a single category, such as lung opacity? If so, could you provide the final list of combined classes used in the dataset?

Additionally, VinDR defines diseases (e.g., pneumonia, cancer, tuberculosis) based on findings such as nodules, fibrosis, consolidations, and etc. How did you establish the mapping between diseases and their corresponding findings? Understanding the rationale behind these classifications would provide better insight into the dataset's structure and its impact on model performance.

**Suggestion**

Combining and refining publicly available datasets is valuable and provides significant benefits to the research community. However, in my opinion, a more substantial revision of the dataset should be undertaken. For example, the relabeled version of ImageNet (Beyer et al., 2020) has been widely appreciated for improving data quality. Similarly, in image-text datasets, re-captioning existing datasets such as LAION or Conceptual Captions (CC) has proven helpful for researchers and practitioners.

Following this direction, this work would greatly benefit from further efforts to refine the existing datasets. One possible approach is to standardize terminology more rigorously in collaboration with domain experts, such as thoracic radiologists. Although this work has made some efforts toward standardization, I could not find detailed information on the extent of these efforts. Specifically, it remains unclear how many radiologists were involved and what process was followed for standardization. Additionally, rephrasing existing Chest X-ray reports using standardized medical terminology would be highly valuable for deep learning applications in radiology. These enhancements would improve the dataset’s usability and reliability, ultimately benefiting clinical AI research.

**Other Strengths And Weaknesses:**

None

**Questions For Authors:**

Please refer to the comment section above.

**Relation To Broader Scientific Literature:**

It’s related to ML4H.

**Theoretical Claims:**

N/A

---

> ### Author Rebuttal · Authors · 2025-04-01
>
> We appreciate reviewer SVNB's feedback. Besides our scale and focus on multimodal, our work distinguishes itself from related research in several ways:
>
> - Our focus extends beyond confirming general pretraining benefits by specifically targeting underrepresented regions and modalities (ultrasound, CT, EEG). These more rare, OOD tasks have demonstrated superior performance gain as compared to tasks that occur often in the datasets. This challenges the traditional idea that pretraining mainly helps with popular tasks while struggling to improve perf. on OOD tasks (https://arxiv.org/abs/2211.08411, https://arxiv.org/abs/2212.10511).
>
> - In addition, we show that traditional vision encoders still perform better than medical VLLMs by a large margin, whereas related works mentioned in the review mainly focus on improving VLLMs.
>
> - We release a unified framework that streamlines downloading, processing, and model training across vision, time series, and graph domains, enabling researchers to rapidly replicate and iterate on methods against diverse clinical tasks. Our work is of a much larger scale than related works, as shown in our response to Reviewer pTYB.
>
> Comment on Figure 4: We define underrepresented regions and understudied modalities in Lines 1219-1290 using a two-step approach that identified geographic and economic gaps in dataset coverage. While 8/30 (26.7%) datasets with known collection sites come from these regions, this representation still lags behind developed countries. We will clarify that our classification of understudied modalities reflects public dataset availability rather than clinical importance.
>
> On dataset construction: We put extensive efforts into standardizing the taxonomy. Standardization is difficult as it requires balancing two competing objectives:
>
> - Merging similar terms to facilitate learning and cross-modality transfer
> - Minimizing information loss and avoiding inaccuracies when modifying labels
>
> Our standardization efforts concentrated on ECG and chest X-ray, which offer fine-grained labels with varying terminologies. For ECG, we followed approaches from [arXiv:2304.08486](https://arxiv.org/abs/2304.08486) and formulated a mapping [here](https://anonymous.4open.science/r/climb_submission-5D0E/Label_Processing.md) in our anonymous repo.
>
> For Chest X-rays, we developed a new mapping with radiologist input. Following practices from [1](https://pmc.ncbi.nlm.nih.gov/articles/PMC10173935), [2](https://www.nature.com/articles/s41598-023-33303-y) and [3](https://pmc.ncbi.nlm.nih.gov/articles/PMC11455863/), we consolidated all Chest X-ray labels into the CheXpert 14 categories:
>
> | Raw Label | Standardized Label |
> |-----------|-------------------|
> | Aortic enlargement, Enlarged PA | Enlarged Cardiomediastinum |
> | Cardiomegaly | Cardiomegaly |
> | Atelectasis | Atelectasis |
> | Consolidation | Consolidation |
> | Edema | Edema |
> | Infiltration, Lung Opacity, ILD, Pulmonary fibrosis | Lung Opacity |
> | Nodule/Mass, Other lesion, Lung cavity, Lung cyst, Lung tumor | Lung Lesion |
> | Pleural effusion | Pleural Effusion |
> | Pleural thickening | Pleural Other |
> | Pneumothorax | Pneumothorax |
> | Rib fracture, Clavicle fracture | Fracture |
> | No finding | No Finding |
> | Support Devices | Support Devices |
> | Pneumonia | Pneumonia |
>
> Labels from other modalities with the same name are then merged with these classes. We acknowledge standardization may affect label granularity, particularly for lung opacity and lung lesion classes. CLIMB provides both standard and raw labels, allowing researchers to prioritize either granularity or standardization. All experiments in our work used standard labels.
>
> To evaluate standardization effects, we compared vision encoders trained on raw versus standardized labels:
>
> | Model | CXR | | | Mammo | | | Derm | | | CT | | | Fundus | | | US | | | Overall | | |
> |---|---|---|---|---|---|---|---|---|---|---|---|---|---|---|---|---|---|---|---|---|---|
> | | AUC | Sen | Spe | AUC | Sen | Spe | AUC | Sen | Spe | AUC | Sen | Spe | AUC | Sen | Spe | AUC | Sen | Spe | AUC | Sen | Spe |
> | ConvNextV2-RawLabel | **.820** | .358 | .935 | .543 | .293 | .693 | .853 | .492 | .757 | **.690** | .442 | .624 | .794 | .351 | .841 | .689 | .519 | .713 | .751 | .434 | .766 |
> | ConvNextV2-StandardLabel | .817 | **.436** | **.939** | **.558** | **.330** | **.706** | **.901** | **.568** | **.777** | .671 | **.466** | **.641** | **.873** | **.563** | **.888** | **.774** | **.641** | **.770** | **.787** | **.537** | **.806** |
>
> Standardized labels yielded better performance: 3.6% improvement in overall AUC, 10.3% in sensitivity, and 4.0% in specificity. Notably, standardization benefited other modalities more than Chest X-ray itself, where most relabeling occurred. We believe standardized labels help the model connect concepts across modalities more effectively. We hope this work motivates further efforts in terminology standardization and fine-grained relabeling of public clinical datasets.

---

### Official Review · Reviewer_pTYB · 2025-03-14

**Overall Recommendation:** 4

**Summary:**

The paper introduces CLIMB (Clinical Large-scale Integrative Multimodal Benchmark), a clinical benchmark that puts together a large number of existing datasets across different modalities with a strong focus on vision, including 1D, 2D, and 3D signals, as well as graph data. The authors conduct a thorough comparison of models trained on CLIMB improve performance across a number of different tasks compared to the best existing models in the literature for these tasks (often either pretrained within a single modality). CLIMB is very large scale (including over 4.5M patients and 19TB of data). To obtain CLIMB, one must go through the steps necessary for each individual dataset.

## Update after rebuttal
I have read the rebuttal and will keep my recommendation as is.

**Claims And Evidence:**

- Lines 237-239 (column 2): "To evaluate few-shot generalization, we test on out-of-distribution (OOD) datasets $D_\text{ood} \not\subset D_\text{train}$." => You are claiming that if a dataset is not part of the training datasets used, it is out-of-distribution, but that may not always be the case. Authors themselves explain in lines 185-188 (column 2) that they "balance the dataset such that each modality contains 3-5 datasets, providing multiple data sources per modality while maintaining diversity within each category." It is likely that another dataset will still have a lot in common with the training datasets. Please do not use OOD to describe these datasets unless you formally quantify their "OOD-ness", e.g., by training a membership model.
- In Table 5, you show results for the "SoTA" encoder compared to "Ours". Please include the citation of the paper(s) that introduce the SoTA encoder(s) instead of only "SoTA". Where can we find these papers with SoTA results for LOS and IHM?
- In your impact statement, lines 470-473, you state: "Our holistic evaluation metrics will also encourage the research community to quantify the tradeoffs between performance, complexity, robustness, fairness, and privacy in clinical AI." => I only see evaluation metrics related to standard model performance, and even in that case only simple metrics like AUC, accuracy, precision, and recall are used for classification, and MAE for regression. Please remove that from your impact statement.

**Essential References Not Discussed:**

No crucial references missing that I noticed to the best of my knowledge.

**Experimental Designs Or Analyses:**

The experimental design used in the paper is comprehensive. The set of baselines compared against is extensive, and it is really nice to see this large-scale, wide comparison being made. The main research questions looked into in this paper have to do with whether pretraining a model on CLIMB improves downstream performance across tasks, and to what extent do models transfer to unseen datasets.

**Methods And Evaluation Criteria:**

- Not sure where to mention this, but in my opinion an issue with CLIMB is not explicitly discussing how the user can obtain the data upfront. Anyone working with applications in healthcare know this is a major bottleneck. There are details in the github repository shared and in the appendices about the licences, but it would be nice to have a paragraph early on in the main paper where authors clearly state that users need to obtain approval to access specific datasets, that upon doing that they can use their codebase to download all datasets automatically, that it can take "X" weeks/months to obtain approval, etc.
- One part of the data I find the authors did not explore well (even in the appendices) is textual and EHR data. In fact, CLIMB is mostly an image-first benchmark, perhaps also a time-series-second benchmark. However, there is very little textual or EHR data in CLIMB, but this is not at all something clear from the paper. (For example, in Table 1 "Comparison of clinical benchmarks" it seems that CLIMB "fully covers" the text modality, which I find does not fully tell the truth.) Free-text clinical notes and EHR data are, to the best of my knowledge, only available from MIMIC-IV (one of the numerous datasets used in CLIMB). Even in Appendix A.2, where the "Understudied modalities" are discussed, only images are mentioned. These downsides, such as limited EHR/text data, should be more clearly highlighted to the reader in the main paper. => I am not sure what would be the alternative to showcase this more clearly, but perhaps you could have a plot like a bar plot or a radar chart, where in each dimension you have proxies to the amount of text in the dataset (e.g., "number of words", "number of documents", "size in GB", etc). That way you could compare CLIMB with existing benchmarks (like PMC-VQA, GMAI-MMBench, CARES) in more detail when it comes to the amount of text available in the benchmarks. For this, you should also include, for example, "comments" in the COVID-BLUE dataset, or any other free-text data available.
- For evaluation of in-hospital mortality in Table 5, you should probably include the F-1 score and the area under the precision-recall curve. (Perhaps in the appendix, if that does not fit the main paper.) I find that the answer to RQ3 is a bit lacking compared to the other 2 RQs, RQ1 and RQ3.

**Other Comments Or Suggestions:**

- In line 325, you refer to Table D.1 but it links to Table 25 in Appendix D.1.
- If you are going to focus on precision and recall (like in Tables 2, 3, etc), you should probably also include an F-score. The F-score helps quickly see the trade-off between both quantities (which is of course do-able in with only P and R, but considerably slower).
- In all your Tables, only embolden the overall best-performing results (using bold-face). In Table 2, for instance, you embolden InternViT Mammography Sen (.340), but MedViT obtained .417 Sen. You also say in Table 2's caption that "The best performance of each model in AUC is bolded." However, it is not just the AUC that is emboldened. In Table 5, SoTa MLP for 48 IHM (8-Shots) has recall of .536, whereas the entry that is emboldened is Ours MLP with a recall of 0.295. Please fix these typos/mistakes/inconsistencies throughout the paper.
- Figures and Tables in your paper do not appear in the order they are mentioned in the text. Please fix that.
- You often compare few-shot performance with "full dataset" performance, like for instance in Figure 6 or Table 5. I may have missed that, but in "full dataset" do you mean the model is fine-tuned on the full training data available for the task? If that is the case, please do not use "1 shot", "8 shots", and "full", but for the latter use "full FT" instead (and explain in the figure/table caption that FT means fine-tuning).

**Other Strengths And Weaknesses:**

Strengths worth mentioning:
- Very large scale clinical benchmark
- Extensive experiments showing improvements on downstream tasks

Weaknesses specific to each dimension of the review are discussed in the dimension's section.

**Questions For Authors:**

For MIMIC-IV experiments under RQ3: what exactly are the clinical notes you used (radiology reports, discharge summaries)? For modelling length of stay as a regression task, did you do any normalisation of the stay duration?

Is it fair to say that this is an image-first, time-series second benchmark? For instance, in Appendix C4, for your vision experiments you have 8 baseline models, for your EEG/ECG experiments 8 more models, that cover a number of different architectures and strategies. For text and EHR structured data, you have ClinicalBERT.

You mention you use "EHR data" from MIMIC-IV, but I could not find a clear explanation of what exact variables do you mean by that. What are the variables used in your experiments that you refer to as "EHR data"?

**Relation To Broader Scientific Literature:**

Good comparisons to existing benchmarks (Table 1) to the best of my knowledge.

**Theoretical Claims:**

No theoretical claims are made in this paper that I could verify.

---

> ### Author Rebuttal · Authors · 2025-04-01
>
> We thank Reviewer pTYB for the positive reviews and constructive feedback. We have addressed all typographical errors in our manuscript.
>
> Regarding the definition of OOD datasets (Claim 1), we selected OOD datasets primarily based on task differences and provided a comprehensive list with justifications in the App. Table 9. Of the 10 OOD datasets, 7 (COVID-19, CoronaHack, ISIC-2020, BCSS, BUSI, LNDb, PTB-XL-Finegrained) have zero label overlap with other datasets in the same clinical domain, while the remainder feature distinct task granularity. To quantify this heterogeneity, we conducted a membership experiment using ConvNeXT-v2-base as the backbone to predict dataset membership, following hyperparameter settings in Appendix C.4.2. Results confirm our designated OOD datasets exhibit substantial distinctiveness:
>
> |OOD Dataset|Balanced Accuracy|AUC|F1 Score|
> |-----------|----------------|---|--------|
> |BCSS|99.8|1.000|0.928|
> |CBIS-DDSM|99.9|1.000|0.801|
> |CoronaHack|99.7|0.999|0.371|
> |COVID-19|53.2|0.953|0.112|
> |BUSI|99.9|1.000|0.997|
> |Jichi|99.8|0.999|0.758|
> |ISIC-2020|99.7|0.999|0.841|
>
> Claim 2: In Table 5, "SoTA encoders" refers to the best pre-trained encoders tested across clinical-specific and general domain models. Specifically, as detailed in Sec. 4.1, we employed ConvNextV2, ECG JEPA, and ClinicalBERT. Our experimental protocol follows [FuseMoE](https://arxiv.org/abs/2402.03226), except we implemented regression without binning for LOS to increase granularity, and utilized AUC, sensitivity and specificity for 48 IHM to maintain consistency throughout the paper. We will incorporate a comprehensive results table for experiment 3 in the appendix.
>
> Claim 3: We will remove lines 470-473 from the impact statement. As typical of impact statements, this referred to potential future work. The dataset samples contain demographic information where available, which could facilitate future evaluation of robustness, fairness, and privacy in clinical AI.
>
> Methods 1: Data accessibility was a key consideration in dataset selection, as outlined on page 23 (Dataset Selection Methodology). We specifically selected datasets that do not require lengthy approval processes. Researchers can access 37 of 44 datasets in CLIMB after completing these steps, requiring less than 4 hours total:
>
> - Register for PhysioNet (15 mins) and Kaggle (10 mins) accounts
>
> - Complete CITI certification training (3 hours)
>
> - Acknowledge dataset agreements (10 seconds each)
>
> Our framework then manages all downloads and processing automatically.
>
> Methods 2: CLIMB integrates all metadata, labels and text reports with each sample. We did an analysis of the number of words, the number of QA pairs and the total size of the dataset. Comparing with other multimodal clinical QA datasets:
>
> | Dataset        | Number of Words | Num QA Pairs | Size of Dataset |
> |----------------|------------------|----------------|------------------|
> | CLIMB-QA      | 129.1M           | 4.51M          | 19.01 TB         |
> | PMC-VQA        | 10.2M            | 227K           | -                |
> | GMAI-MMBench   | 980K             | 26K            | 49 GB            |
> | CARES          | 1.74M            | 41K            | 21.61 GB         |
>
> Our dataset exceeds others by at least an order of magnitude in all three metrics.
>
> Methods 3: We have added accuracy and F-1 score to Table 5. We include a subset here due to space limit and will add the full table in Appendix of final manuscript:
>
> | | | LOS | 48 IHM (Full) | | | | | 48 IHM (8-Shots) | | | | |
> |---|---|---|---|---|---|---|---|---|---|---|---|---|
> | **Enc** | **Fusion** | **MAE** | **AUC** | **Sens** | **Spec** | **Accuracy** | **F1 Score** | **AUC** | **Sens** | **Spec** | **Accuracy** | **F1 Score** |
> | SoTA | CrossAtt | 2.77 | 0.786 | 0.628 | 0.814 | 0.792 | 0.416646562 | 0.58 | 0.286 | 0.766 | 0.763 | 0.014860231 |
> | Ours | MLP | 2.84 | 0.961 | 0.824 | 0.975 | 0.968 | 0.704789834 | 0.672 | 0.295 | 0.858 | 0.767 | 0.290421866 |
> | Ours | CrossAtt | 2.61 | 0.796 | 0.822 | 0.59 | 0.825 | 0.90490467 | 0.57 | 0.294 | 0.753 | 0.728 | 0.105340098 |
>
> Question 2: In this work, our focus is on vision, time series and graphs, where large pretraining efforts are particularly scarce. We focused less on textual modality since multiple textual LLMs trained on diverse medical data already exist, including [ClinicalBERT](https://huggingface.co/medicalai/ClinicalBERT) and [BioMistral](https://huggingface.co/BioMistral/BioMistral-7B).
>
> Question 1,3: The EHR data encompasses all textual data available in MIMIC-IV within 48 hours of admission: vital signs, lab measurements, treatments, medications, and demographics. We excluded data without timestamps (e.g., diagnoses) and omitted discharge summaries and radiology reports. We performed no normalization or binning on LOS regression. The text is JSON-formatted and fed directly into the text model for embedding.

---

### Decision · Program_Chairs · 2025-05-01

**Decision:**

Accept (poster)

**Comment:**

This manuscript introduces CLIMB (Clinical Large-scale Integrative Multimodal Benchmark), a comprehensive clinical benchmark integrating diverse data across imaging, language, temporal and graph modalities, comprising 4.51 million patient samples (19.01 TB). The work demonstrates that multitask pretraining on CLIMB significantly improves performance on understudied domains and modalities, with notable gains in ultrasound (29%) and ECG analysis (23%). The reviewers highlighted several key strengths, including the extensive scale and diversity of the benchmark, standardised taxonomy development with radiologist input, and improved performance on underrepresented regions and modalities. Through their rebuttal, the authors provided additional experimental evidence comparing supervised multitask learning with self-supervised approaches, demonstrating that vision models benefited primarily from multitask learning while time-series models showed substantial gains from unsupervised pretraining. Some limitations noted by reviewers included the predictability of certain findings regarding pretraining benefits and the need for more thorough analysis of negative transfer cases. The rebuttal effectively addressed concerns about data accessibility and standardisation approaches, leading two reviewers to increase their scores. Following the rebuttal, reviewers reached general consensus supporting acceptance, with scores ranging from weak to full accept, particularly appreciating the meaningful improvements demonstrated for understudied modalities and underrepresented regions.